# Machine Unlearning in Audio: Bridging The Modality Gap Via the Prune and Regrow Paradigm

## Abstract

The ubiquity and success of deep learning is primarily owed to large human datasets; however, increasing interest in personal data raises questions of how to satisfy privacy legislation in deep learning. Machine unlearning is a nascent discipline centred on satisfying user privacy demands, by enabling data removal requests on trained models. While machine unlearning has reached a good level of maturity in the vision and language domains, applications in audio are largely underexplored, despite it being a highly prevalent and widely used modality. We address this modality gap by providing the first systematic analysis of machine unlearning techniques covering multiple architectures trained on audio datasets. Our analysis highlights that in audio, existing methods fail to remove data for the most likely case of unlearning – Item Removal. We present a novel Prune and Regrow Paradigm that bolsters sparsity unlearning through Cosine and Post Optimal Pruning, achieving the best unlearning accuracy for 9/12 (75%) of Item Removal experiments and best, or joint best, for for 50% (6/12) of Class Removal Experiments. Furthermore, we run experiments showing performance as unlearning requests scale, and we shed light on the mechanisms underpinning the success of our Prune and Regrow Paradigm.

## 1 Introduction

Deep Neural Networks (DNNs) have achieved remarkable success across several applications and modalities, such as disease classification (Bondareva et al., 2023; Abbas et al., 2024), facial expression recognition (Canedo & Neves, 2019), and clinical advice (Singhal et al., 2023). Alongside the success of DNNs, several challenges have arisen, notably adherence to the *Right To Be Forgotten* (RTBF) (a key principle General Data Protection Regulation (GDPR) (European Parliament & Council of the European Union)) and other removal legislation that is gaining momentum worldwide (APP, 2003; IND, 2023; BUKATY, 2019).

The machine unlearning domain has emerged in response to the RTBF in DNNs, providing a structured and auditable way of removing data from models, enabling organisations to comply with GDPR. Naive Retraining, the approach of removing training instances and retraining a new model from scratch, is a largely impractical (Xu et al., 2023; He et al., 2021), but verifiable exact machine unlearning approach. While machine unlearning has verifiable implementations within statistical querying (Cao & Yang, 2015), it is a challenge in deep learning due to the stochastic and incremental nature of training (Nguyen et al., 2022; Bourtoule et al., 2021). As a result, machine unlearning focuses on developing unlearning mechanisms that can remove the influence of data in a computationally inexpensive and verifiable manner, overcoming the costs of Naive Retraining.

Despite the expanding use of audio DNNs in applications such as voice recognition (Hughes & Mierle, 2013), event classification (Dong et al., 2020), and health monitoring (Bondareva et al., 2023; Srivastava et al., 2021; Aptekarev et al., 2023; Barata et al., 2019), there exist no studies that address Item and Class Removal for machine unlearning in the audio domain, while there is a cumulative total of over 100 studies in other domains (Shaik et al., 2023; Zaman et al., 2023). Studying machine unlearning in audio is vital for safeguarding and maintaining data privacy, upholding the RTBF, and reducing the computational costs associated with Naive Retraining.

Our work bridges this modality gap in unlearning literature and systematically studies the effectiveness and adaptability of existing unlearning methods (previously applied to other domains) on audio data – specifically, AudioMNIST, Becker et al. (2023); SpeechCommands V2, Warden (2017) and UrbanSounds8K Salamon et al. (2014) – and across different architectures. Our findings show that, while current methods are effective for Class Removal, they are inadequate for Item Removal, regarded as the most important unlearning task Nguyen et al. (2022). Our proposed *Prune and Regrow Paradigm* fills this gap by leveraging dynamic sparsity unlearning for audio models that remove the requirement for extensive empirical studies and, we also show the transferability of this dynamic sparsity method on CIFAR10 Krizhevsky et al. (2009)(Appendix F) where it achieves the best Item Removal for all architectures. Additionally, our study into unlearning scaling shows that our method remains performant as Item Removal requests scale.

The contributions of this paper are threefold:

- An in-depth study and evaluation of five existing strong unlearning methods on three different audio datasets and core architecture classes under Item and Class Removal, revealing that the majority of current approaches are ineffective on Item Removal requests, necessitating the development of novel methods for audio data.

- A novel *Prune and Regrow Paradigm* that achieves the lowest unlearning accuracy gap 9/12 (75%) of the time for Item Removal across three audio datasets and three architectures and transfers to CIFAR10.

- An investigation into the scaling laws of unlearning in audio that uncovers the ability of existing and novel unlearning methods to scale for increased removal requests, showing greater applicability of methods in audio.

## 2 EXISTING MACHINE UNLEARNING AND EVALUATION METHODS

In this section, we formalise machine unlearning, types of unlearning requests, existing machine unlearning methods and evaluation metrics used in previous literature. $\mathcal{M}^-$ and $\mathcal{M}_r^\theta$ represent the **Unlearned** model and the **Naive** model respectively.

### 2.1 MACHINE UNLEARNING PRIMER

Strong machine unlearning represents a more practical version of unlearning that deviates from creating an unlearnt ($\mathcal{M}^-$) and retrained ($\mathcal{M}_r^\theta$) model that is indistinguishable to creating an $\mathcal{M}^-$ that approximates $\mathcal{M}_r^\theta$ (Xu et al., 2023). Strong unlearning can be represented as a mathematical problem in equation 1 - equation 4. Strong unlearning is described as a less strict formalisation of machine unlearning that enables a broader array of unlearning methods.

$$\text{Take a training dataset: } \mathcal{D}_{train} = \{(x_1, y_1), \ldots, (x_n, y_n)\} \tag{1}$$

$$\text{Apply Learning Algorithm: } \mathcal{M}^\theta \xleftarrow{\$} \mathcal{M}(A(\mathcal{D}_{train})) \tag{2}$$

Identify instances to be removed forming $\mathcal{D}_{forget}$ and apply an unlearning mechanism $\mathcal{U}$ to remove the influence of $\mathcal{D}_{forget}$ from the parameter distribution of $\mathcal{M}^\theta$:

$$\text{Apply Unlearning Mechanism: } \mathcal{M}^- = \mathcal{U}(\mathcal{M}^\theta, \mathcal{D}_{forget}) \tag{3}$$

Create a model with an internal distribution that *strongly* resembles the distribution of a model that is an instance of a possible model retrained on $\mathcal{D}_{forget}$.

$$\text{Strong Removal Goal: } \mathcal{U}(\mathcal{M}(A(\mathcal{D}_{train}), \mathcal{D}_{forget}) \approx \mathcal{M}(A(\mathcal{D}_{remain})) \tag{4}$$

**Item & Class Removal** The most common unlearning request is identified in **Item Removal** (Nguyen et al., 2022). A forget set ($\mathcal{D}_{forget}$) is to be removed from the parameter distribution of a model ($\mathcal{M}^\theta$). The task is to remove the influence of $\mathcal{D}_{forget}$ from $\mathcal{M}^\theta$ with an unlearning mechanism, $\mathcal{U}$, to create $\mathcal{M}^-$ that is approximately or absolutely equal to a parameter distribution of a retrained model ($\mathcal{M}_r^\theta$) trained on the remaining dataset ($\mathcal{D}_{remain}$). A challenging unlearning

request emerges in the form of a **Class Removal** request (Nguyen et al., 2022); the task is to remove the impact of all instances included within the class to unlearn contained in $\mathcal{M}^\theta$. Ultimately, Class Removal requires the destruction of a decision boundary from $\mathcal{M}^\theta$ ensuring $\mathcal{M}^-$ classifies the instances within $\mathcal{D}_{forget}$ as the remaining classes in $\mathcal{D}_{remain}$.

## 2.2 Unlearning Methods

Numerous machine unlearning methods have been devised in other modalities; this section presents the existing methods we use to evaluate current unlearning capacity for audio. In the Appendix, we describe the benefits and drawbacks of these approaches in Table 6 of Section A.

✶ **Gradient Ascent (GA):** Gradient Ascent (Graves et al., 2021; Thudi et al., 2022) is one of the simplest strong unlearning methods. When an unlearning request is made, gradient ascent subverts the training strategy and moves in gradient mini-batches in the opposing direction to make a gradient ascent step on $\mathcal{D}_{forget}$. Accuracy is then recovered through fine tuning on $\mathcal{D}_{remain}$.

✶ **Fine Tuning (FT):** Fine Tuning unlearning (Golatkar et al., 2020a; Liu et al., 2024; Choi & Na, 2023; Wang et al., 2022) leverages catastrophic forgetting (McCloskey & Cohen, 1989) to fulfil removal requests. The rudimentary approach employs fine-tuning on $\mathcal{D}_{remain}$ to get $\mathcal{M}^-$ and remove the influence of instances in $\mathcal{D}_{forget}$.

✶ **Stochastic Teacher (ST):** Stochastic Teacher unlearning (Zhang et al., 2023), also known as Incompetent Teacher unlearning (Chundawat et al., 2023a), leverages knowledge distillation (Hinton et al., 2015) for unlearning. The competent teacher is the original $\mathcal{M}^\theta$ and the stochastic teacher is a randomly initialised $\mathcal{M}^\theta$, $M_{init}$. The student starts as $\mathcal{M}^\theta$ trained on $\mathcal{D}_{train}$. During the unlearning process, for $\mathcal{D}_{remain}$, the student receives the logits of $\mathcal{M}^\theta$ but on instances from $\mathcal{D}_{forget}$, it receives the logits from $M_{init}$.

✶ **One-Shot Magnitude Prune (OMP):** Sparsity unlearning via OMP at 95% sparsity can significantly reduce the approximation gap between $\mathcal{M}^\theta_r$ and $\mathcal{M}^-$ fine-tuned on $\mathcal{D}_{remain}$ (Liu et al., 2024). OMP takes an $\mathcal{M}^\theta$ and prunes weights and biases to 0 with a mask that prevents weight updates when fine-tuning on $\mathcal{D}_{remain}$.

✶ **Amnesiac (AM):** Amnesiac unlearning (Graves et al., 2021; Golatkar et al., 2020b), seeks to remove $\mathcal{D}_{forget}$ from $\mathcal{M}^\theta$ by forcing a $\mathcal{M}^\theta$ to learn random class relationships for $\mathcal{D}_{forget}$. The operation is performed by taking $\mathcal{D}_{forget}$ and modifying it to add a random incorrect, $y_{ri}$, label to each instance. Following this, the $\mathcal{M}^-$ is fine-tuned on $\mathcal{D}_{remain}$.

## 2.3 Evaluation Metrics

Unlearning literature has devised several metrics to quantify the unlearning performed by an unlearning mechanism. The metrics employed are described below and formalised in the Appendix in Table 8 of Section B.

✶ **Unlearning Accuracy (UA):** The performance of $\mathcal{M}^-$ on $\mathcal{D}_{forget}$. Compared to $\mathcal{M}^\theta_r$.

✶ **Remain Accuracy (RA):** Performance of $\mathcal{M}^-$ the remain set $\mathcal{D}_{remain}$ compared to $\mathcal{M}^\theta_r$.

✶ **Test Accuracy (TA):** Accuracy on $\mathcal{D}_{test}$ of $\mathcal{M}^-$ compared to $\mathcal{M}^\theta_r$

✶ **Membership Inference Attack Efficacy (MIA Efficacy):** Membership Inference attacks (Shokri et al., 2017), established the goal of taking a machine learning model $\mathcal{M}^\theta$ and an instance $(x_i, y_i)$ and deducing whether $x_i, y_i \in \mathcal{D}_{train}$ or $x_i, y_i \notin \mathcal{D}_{train}$ (Shokri et al., 2017). For machine unlearning MIA Efficacy is the proportion of data points in $\mathcal{D}_{forget}$ classified as non-training instances, $y_1$ (Graves et al., 2021; Liu et al., 2024). If MIA Efficacy of $\mathcal{M}^- > \mathcal{M}^\theta_r$, the Streisand Effect is induced, which can undermine the privacy.

✶ **Disparity Average (D AVE):** The disparity of $\mathcal{M}^-$ and $\mathcal{M}^\theta_r$ on UA, RA, TA and MIA Efficacy.

✶ **Activation distance (A DIST):** The $\mathcal{L}_2$ distance of softmax outputs of $\mathcal{M}^\theta_r$ compared to $\mathcal{M}^-$ on $\mathcal{D}_{forget}$. It is proxy for the amount $\mathcal{D}_{forget}$ removed from $\mathcal{M}^-$.

✶ **Jensen-Shannon Divergence (JS DIST):** A weighted average of KL divergence (Lin, 1991) of the loss of $\mathcal{M}^-$ compared to $\mathcal{M}^\theta_r$ on $\mathcal{D}_{forget}$.

⋆ **Run-Time Efficiency (RTE):** The compute efficiency increase of creating $\mathcal{M}^-$ compared to retraining a model to create $\mathcal{M}_r^\theta$.

# 3 PRUNE AND REGROW PARADIGM

We argue that an effective unlearning approach for audio is dynamic and sensitive to both architecture and learned features. To create a dynamic unlearning method that can respond uniquely to features learned by different architectures on different datasets, we devise the *Prune and Regrow Paradigm* that employs sparsity unlearning. Pruning is an effective compression method across modalities; literature has shown that its efficacy relates to the functional preservation of the compressed model (Mason-Williams, 2024). The sparsity unlearning paradigm has emerged as a promising candidate for unlearning in computer vision (Liu et al., 2024; Wang et al., 2022). One Shot Magnitude Pruning (OMP) at 95% sparsity (based on empirical studies on CIFAR10) provides current SOTA unlearning in vision (Liu et al.,

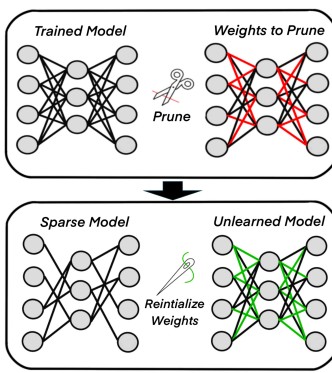

Figure 1: Prune and Regrow Process: Prune based on cosine similarity, remove mask weights and reinitialize zeroed weight and fine-tune.

2024). However, we argue that a one-size-fits-all sparsity unlearning cannot be optimal due to different learnt features across modalities. Additionally, network compression is not the aim of machine unlearning, and by imposing high sparsity, a machine unlearning budget is placed on $\mathcal{M}^-$ as repeatedly pruning the compressed model to 95% will eventually lead to model degradation.

Inspired by sparsity unlearning, we devise a novel unlearning method that is adaptive to modality and architecture. Through Cosine and Post Optimal Prune unlearning, we demonstrate the *Prune and Regrow Paradigm*. The paradigm, Figure 1, prunes a model to a sparsity determined by cosine similarity (Mason-Williams & Dahlqvist, 2024), as seen in Figure 2, and then removes the pruned masks and reinitializes the pruned weights to create $\mathcal{M}^-$ which is fine-tuned on $\mathcal{D}_{forget}$. As a result, more weights are available during fine-tuning, allowing for improved functional expression as more parameters are updated when $\mathcal{M}^-$ is fine-tuned on $\mathcal{D}_{forget}$. The unlearning budget is also increased, as this method can be performed repeatedly without pruning to the same representation each time. To address this we present CS and POP unlearning methods that operate under the Prune and Regrow Paradigm.

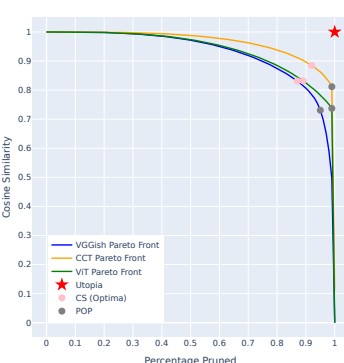

Figure 2: Cosine Similarity as Model is Pruned at 1% Intervals for SpeechCommands Models.

**Cosine Unlearning (CS):** By preserving the Cosine Similarity, it is possible to maintain functional similarity and maximally prune a model (Mason-Williams & Dahlqvist, 2024), by getting the minimum distance from the theoretical utopia where Cosine Similarity is 1 and pruning amount is 1, as seen in Figure 2. To perform Cosine pruning, a DNN is converted into a vectorised form and pruned at 1% intervals, computing the Cosine Similarity between the two vectorised DNNs (Mason-Williams & Dahlqvist, 2024). An optimisation preserves Cosine Similarity while pruning the model as much as possible, the minimum distance from Uptopia [1,1]. We leverage this to produce CS unlearning as it provides a principled way to identify the correct sparsity per architecture without extensive empirical experiments.

**Post Optimal Prune (POP):**   For POP unlearning, we use the maximum polar point [0,-1] from Utopia, Figure 2, to increase the percentage of pruning to reduce similarity without degrading performance to an unacceptable standard. By taking a post-optimal pruning step the overall function is preserved less than with CS. As a critical aspect of machine unlearning is to move away from the $\mathcal{M}^\theta$'s original function towards $M_r^\theta$, pruning more of the network increases the ability to remove $\mathcal{D}_{forget}$.

We employ the *Prune and Regrow Paradigm* to reinitialize zeroed weights and biases to enable better feature representation when fine tuning on $\mathcal{D}_{remain}$ for both CS and POP unlearning.

## 4   Experimental Setup

In this section we introduce our experimental setup. First the datasets we use: covering a range of learning task complexities on which to evaluate unlearning. Then, we introduce the architectures that are representative for audio tasks (Zaman et al., 2023). Unlearning experiments are conducted for both Item: 10%, 20% and 30% and Class: 1, 2 and 3 Removal in audio.

**Datasets**   Our results are collected by training models on AudioMNIST (Becker et al., 2023) (a low-complexity dataset), SpeechCommands V2 (Warden, 2017) and UrbanSounds8K (Salamon et al., 2014) (high-complexity datasets), presented in Table 1. All audio was converted to Mel Spectrograms as is standard practice for audio data due to reduced training time and improved generalisation (Wyse, 2017). To show the applicability of the Prune and Regrow Paradigm we also present results on CIFAR10 in Appendix F.

Table 1: Dataset features from strong machine unlearning experiments.

| Dataset | Hours of Recorded Audio | Training Instances | Testing Instances | Number of Classes |
|---|---|---|---|---|
| SpeechCommands V2 | 29.4 | 84,843 | 11,005 | 35 |
| UrbanSounds8K | 18.5 | 6,985 | 1,747 | 10 |
| AudioMNIST | 9.5 | 24,000 | 6,000 | 10 |
| CIFAR10 | N/A | 50,000 | 10,000 | 10 |

**Architectures:**   The architectures explored cover a range of capacities (Appendix Table 7) and core architecture differences with a model that only contain convolutions, a model that employs both convolutions and attention, to a model that only uses attention mechanisms via the VGGish (Hershey et al., 2017; Simonyan & Zisserman, 2014), Compact Convolutional Transformer (Hassani et al., 2021) (CCT) and Vision Transformer (Dosovitskiy et al., 2020) (ViT). The architectures are trained for 50 epochs (AudioMNIST and SpeechCommands) or 80 epochs (UrbanSounds8k and CIFAR10), optimising cross-entropy loss on the train set, using SGD as the optimiser with momentum=0.9, learning rate=0.01 and batch size of 256.

**Settings:**   All results provided for Item and Class Removal are **averaged across 10 experiments**. To conduct a fair comparison of unlearning methods, each unlearning method requiring an impair step is provided one epoch to maximise the loss on $\mathcal{D}_{forget}$, and each method is provided with 10% of the orignial train epochs for repair/fine tuning on $\mathcal{D}_{remain}$ to recover accuracy. All unlearning methods are compared with Naive Retraining ($\mathcal{M}_r^\theta$) on $\mathcal{D}_{remain}$. Further details on the unlearning setup are presented in Section B of the Appendix alongside implementation details of the evaluation metrics.

## 5   Results and Discussion

In the main body we present SpeechCommands and UrbanSounds8K. For Item Removal the Prune and Regrow Paradigm, via POP, is the best unlearning method on UA for both datasets and for Class Removal ST is the best for SpeechCommands and POP is the best for Urbansounds8K. AudioMNIST results are presented in Appendix E and show that the Prune and Regrow Paradigm, via CS, is the best for Item Removal and ST is the best for class removal. Finally, the results on CIFAR10 in Appendix F show the transferability of the Prune and Regrow Paradigm to other domains as it is the best for Item Removal.

## 5.1 ITEM REMOVAL

The results in Tables 2 and 3 provide exciting insights into how the mechanisms of unlearning manifest for SpeechCommands and UrbanSounds8K. From the results, it can be understood that the Prune and Regrow Paradigm performs the best (4/6) for UA overall across the architectures, with OMP being the second best. When considering the non-pruning methods (GA, FT, ST, AM), they mostly fail to remove $\mathcal{D}_{forget}$ from $\mathcal{M}^-$ when comparing the UA to the Naive Retraining $\mathcal{M}_r^\theta$ as they have an unacceptable deviation of circa 7, 20 and 12 on SpeechCommands and 10, 25 and 23 on UrbanSounds8K for the VGGish, CCT and ViT respectively. While these non-pruning-based unlearning methods retain RA given the failure of GA, FT, ST, AM of them to remove $\mathcal{D}_{forget}$ they are excluded from further analysis on Item removal.

Table 2: **10% Item Removal** results for **SpeechCommands**. Numbers in blue represent disparity from $\mathcal{M}_r^\theta$. $\mathcal{C}$ represents the objective to have the least disparity with $\mathcal{M}_r^\theta$. Otherwise arrows dictate the direction of best performance compared to $\mathcal{M}_r^\theta$.

| Model | Method | UA % ($\mathcal{C}$) | MIA Efficacy % ($\mathcal{C}$) | RA % ($\mathcal{C}$) | TA % ($\mathcal{C}$) | D AVE ($\mathcal{C}$) | A DIST ($\downarrow$) ($\times 10^{-1}$) | JS DIST ($\downarrow$) ($\times 10^{-3}$) | RTE % ($\uparrow$) |
|---|---|---|---|---|---|---|---|---|---|
| | | | | 10 % Item Removal | | | | | |
| VGGish | Naive | $12.09_{\pm0.50}(0.00)$ | $17.06_{\pm2.57}(0.00)$ | $97.84_{\pm1.52}(0.00)$ | $87.61_{\pm0.29}(0.00)$ | $0.00$ | $0.00_{\pm0.00}$ | $0.00_{\pm0.00}$ | $0.00$ |
| | GA | $4.74_{\pm1.70}(-7.35)$ | $9.73_{\pm3.25}(-7.33)$ | $97.71_{\pm0.92}(-0.13)$ | $87.26_{\pm0.32}(-0.35)$ | $3.79$ | $1.60_{\pm0.11}$ | $3.09_{\pm0.51}$ | $85.67$ |
| | FT | $4.77_{\pm1.45}(-7.32)$ | $9.92_{\pm2.60}(-7.14)$ | $97.62_{\pm0.90}(-0.22)$ | $87.27_{\pm0.44}(-0.34)$ | $3.76$ | $1.59_{\pm0.11}$ | $3.08_{\pm0.46}$ | $86.11$ |
| | ST | $67.69_{\pm34.79}(55.60)$ | $81.31_{\pm22.89}(64.25)$ | $33.16_{\pm35.97}(-64.68)$ | $32.27_{\pm34.87}(-55.34)$ | $59.97$ | $7.06_{\pm2.90}$ | $21.26_{\pm12.51}$ | $79.59$ |
| | AM | $4.78_{\pm1.52}(-7.31)$ | $9.97_{\pm2.76}(-7.09)$ | $97.90_{\pm0.93}(0.06)$ | $87.52_{\pm0.31}(-0.09)$ | $3.64$ | $1.56_{\pm0.08}$ | $2.97_{\pm0.43}$ | $85.87$ |
| | OMP | $8.41_{\pm1.29}(-3.68)$ | $18.31_{\pm3.60}(1.25)$ | $94.63_{\pm1.62}(-3.21)$ | $87.56_{\pm0.62}(-0.05)$ | $2.05$ | $1.59_{\pm0.07}$ | $2.33_{\pm0.14}$ | $85.27$ |
| | CS | $7.83_{\pm0.87}(-4.26)$ | $14.87_{\pm1.81}(-2.19)$ | $96.54_{\pm0.95}(-1.30)$ | $87.44_{\pm0.70}(-0.17)$ | $1.98$ | $1.57_{\pm0.11}$ | $2.50_{\pm0.30}$ | $84.73$ |
| | POP | $8.07_{\pm1.00}(-4.02)$ | $15.63_{\pm2.18}(-1.43)$ | $96.42_{\pm1.07}(-1.42)$ | $87.67_{\pm0.38}(0.06)$ | $1.73$ | $1.58_{\pm0.05}$ | $2.48_{\pm0.22}$ | $84.83$ |
| CCT | Naive | $20.92_{\pm0.32}(0.00)$ | $38.69_{\pm0.62}(0.00)$ | $99.94_{\pm0.02}(0.00)$ | $77.19_{\pm0.16}(0.00)$ | $0.00$ | $0.00_{\pm0.00}$ | $0.00_{\pm0.00}$ | $0.00$ |
| | GA | $0.74_{\pm1.73}(-20.18)$ | $7.05_{\pm7.37}(-31.64)$ | $99.46_{\pm1.39}(-0.48)$ | $77.15_{\pm1.19}(-0.04)$ | $13.08$ | $2.89_{\pm0.04}$ | $7.33_{\pm0.47}$ | $87.64$ |
| | FT | $0.49_{\pm0.99}(-20.43)$ | $6.24_{\pm6.54}(-32.45)$ | $99.83_{\pm0.33}(-0.11)$ | $77.37_{\pm0.82}(0.18)$ | $13.29$ | $2.88_{\pm0.05}$ | $7.36_{\pm0.42}$ | $87.88$ |
| | ST | $4.72_{\pm1.62}(-16.20)$ | $38.27_{\pm5.26}(-0.42)$ | $98.67_{\pm1.17}(-1.27)$ | $75.90_{\pm0.69}(-1.29)$ | $4.80$ | $2.70_{\pm0.07}$ | $5.30_{\pm0.34}$ | $83.41$ |
| | AM | $0.37_{\pm0.09}(-20.55)$ | $14.78_{\pm2.75}(-23.91)$ | $99.92_{\pm0.02}(-0.02)$ | $77.62_{\pm0.22}(0.43)$ | $11.23$ | $2.83_{\pm0.04}$ | $7.04_{\pm0.16}$ | $87.6$ |
| | OMP | $13.53_{\pm0.30}(-7.39)$ | $65.74_{\pm1.05}(27.05)$ | $93.78_{\pm0.33}(-6.16)$ | $74.34_{\pm0.43}(-2.83)$ | $10.86$ | $2.80_{\pm0.04}$ | $3.66_{\pm0.10}$ | $86.25$ |
| | CS | $15.72_{\pm0.93}(-5.20)$ | $54.96_{\pm2.27}(16.27)$ | $95.24_{\pm0.99}(-4.70)$ | $74.43_{\pm0.71}(-2.76)$ | $7.23$ | $2.56_{\pm0.13}$ | $3.29_{\pm0.19}$ | $86.82$ |
| | POP | $18.92_{\pm0.78}(-2.00)$ | $63.39_{\pm1.45}(24.70)$ | $92.52_{\pm0.90}(-7.42)$ | $74.31_{\pm0.60}(-2.88)$ | $9.25$ | $2.67_{\pm0.08}$ | $3.14_{\pm0.14}$ | $86.89$ |
| ViT | Naive | $14.23_{\pm1.07}(0.00)$ | $29.07_{\pm0.80}(0.00)$ | $99.82_{\pm0.05}(0.00)$ | $84.91_{\pm0.30}(0.00)$ | $0.00$ | $0.00_{\pm0.00}$ | $0.00_{\pm0.00}$ | $0.00$ |
| | GA | $0.69_{\pm1.26}(-13.54)$ | $7.39_{\pm5.75}(-21.68)$ | $99.46_{\pm1.27}(-0.36)$ | $84.92_{\pm1.18}(0.01)$ | $8.90$ | $1.90_{\pm0.05}$ | $4.71_{\pm0.22}$ | $84.67$ |
| | FT | $0.84_{\pm1.28}(-13.39)$ | $9.09_{\pm7.53}(-19.98)$ | $99.59_{\pm0.72}(-0.23)$ | $84.85_{\pm0.86}(-0.06)$ | $8.42$ | $1.87_{\pm0.04}$ | $4.56_{\pm0.44}$ | $85.03$ |
| | ST | $1.66_{\pm0.48}(-12.57)$ | $23.38_{\pm1.40}(-5.69)$ | $99.82_{\pm0.06}(0.00)$ | $85.27_{\pm0.33}(0.36)$ | $4.66$ | $1.73_{\pm0.04}$ | $3.71_{\pm0.20}$ | $79.38$ |
| | AM | $0.60_{\pm0.16}(-13.63)$ | $13.87_{\pm1.76}(-15.20)$ | $99.87_{\pm0.03}(0.05)$ | $85.29_{\pm0.24}(0.38)$ | $7.32$ | $1.82_{\pm0.04}$ | $4.35_{\pm0.16}$ | $84.69$ |
| | OMP | $13.99_{\pm0.38}(-0.24)$ | $70.21_{\pm1.90}(41.14)$ | $88.75_{\pm0.36}(-11.07)$ | $82.60_{\pm0.26}(-2.31)$ | $13.69$ | $2.06_{\pm0.05}$ | $2.29_{\pm0.07}$ | $83.94$ |
| | CS | $12.12_{\pm0.37}(-2.11)$ | $48.85_{\pm1.52}(19.78)$ | $94.82_{\pm0.40}(-5.00)$ | $83.24_{\pm0.44}(-1.67)$ | $7.14$ | $1.69_{\pm0.07}$ | $1.96_{\pm0.13}$ | $83.38$ |
| | POP | $14.07_{\pm0.38}(-0.16)$ | $57.58_{\pm1.69}(28.51)$ | $91.85_{\pm0.39}(-7.97)$ | $83.09_{\pm0.39}(-1.82)$ | $9.62$ | $1.84_{\pm0.06}$ | $2.10_{\pm0.08}$ | $83.47$ |

Table 3: **10% Item Removal** results for **UrbanSounds8K**. Numbers in blue represent disparity from $\mathcal{M}_r^\theta$. $\mathcal{C}$ represents the objective to have the least disparity with $\mathcal{M}_r^\theta$. Otherwise arrows dictate the direction of best performance compared to $\mathcal{M}_r^\theta$.

| Model | Method | UA % ($\mathcal{C}$) | MIA Efficacy % ($\mathcal{C}$) | RA % ($\mathcal{C}$) | TA % ($\mathcal{C}$) | D AVE ($\mathcal{C}$) | A DIST ($\downarrow$) ($\times 10^{-1}$) | JS DIST ($\downarrow$) ($\times 10^{-3}$) | RTE % ($\uparrow$) |
|---|---|---|---|---|---|---|---|---|---|
| | | | | 10% Item Removal | | | | | |
| VGGish | Niave | $26.18_{\pm3.82}(0.00)$ | $34.28_{\pm2.80}(0.00)$ | $95.24_{\pm1.56}(0.00)$ | $78.37_{\pm0.58}(0.00)$ | $0.00$ | $0.00_{\pm0.00}$ | $0.00_{\pm0.00}$ | $0.00$ |
| | GA | $15.74_{\pm5.41}(-10.44)$ | $32.31_{\pm10.11}(-1.97)$ | $89.95_{\pm5.62}(-5.29)$ | $74.28_{\pm3.70}(-4.09)$ | $5.45$ | $3.13_{\pm0.32}$ | $7.90_{\pm0.82}$ | $87.64$ |
| | FT | $10.04_{\pm3.73}(-16.14)$ | $22.52_{\pm8.45}(-11.76)$ | $95.63_{\pm2.43}(0.39)$ | $78.10_{\pm1.33}(-0.27)$ | $7.14$ | $2.81_{\pm0.16}$ | $7.81_{\pm1.36}$ | $88.27$ |
| | ST | $28.94_{\pm11.27}(2.76)$ | $61.96_{\pm16.64}(27.68)$ | $76.12_{\pm13.30}(-19.12)$ | $68.00_{\pm9.59}(-10.37)$ | $14.98$ | $3.85_{\pm1.23}$ | $8.54_{\pm4.96}$ | $77.97$ |
| | AM | $9.83_{\pm2.93}(-16.35)$ | $20.63_{\pm6.05}(-13.65)$ | $95.33_{\pm2.44}(0.09)$ | $77.80_{\pm1.75}(-0.57)$ | $7.66$ | $2.81_{\pm0.13}$ | $7.92_{\pm1.08}$ | $88.11$ |
| | OMP | $21.76_{\pm2.52}(-4.42)$ | $55.14_{\pm5.38}(20.86)$ | $80.45_{\pm2.31}(-14.79)$ | $71.44_{\pm1.42}(-6.93)$ | $11.75$ | $3.46_{\pm0.31}$ | $7.06_{\pm1.05}$ | $85.74$ |
| | CS | $20.41_{\pm8.66}(-5.77)$ | $40.74_{\pm15.48}(6.46)$ | $86.41_{\pm10.01}(-8.83)$ | $73.51_{\pm2.73}(-4.86)$ | $6.48$ | $3.13_{\pm0.90}$ | $7.20_{\pm2.91}$ | $86.05$ |
| | POP | $20.52_{\pm6.06}(-5.66)$ | $42.70_{\pm12.93}(8.42)$ | $85.80_{\pm7.48}(-9.44)$ | $73.64_{\pm5.02}(-4.73)$ | $7.06$ | $3.11_{\pm0.65}$ | $6.64_{\pm1.62}$ | $86.37$ |
| CCT | Niave | $29.84_{\pm2.01}(0.00)$ | $55.71_{\pm1.80}(0.00)$ | $99.39_{\pm0.18}(0.00)$ | $72.48_{\pm1.00}(0.00)$ | $0.00$ | $0.00_{\pm0.00}$ | $0.00_{\pm0.00}$ | $0.00$ |
| | GA | $1.22_{\pm1.44}(-28.62)$ | $11.21_{\pm11.64}(-44.50)$ | $99.42_{\pm0.24}(0.03)$ | $71.99_{\pm0.94}(-0.49)$ | $18.41$ | $3.99_{\pm0.22}$ | $15.77_{\pm1.69}$ | $84.34$ |
| | FT | $0.49_{\pm0.27}(-29.35)$ | $6.24_{\pm3.64}(-49.47)$ | $99.51_{\pm0.18}(0.12)$ | $72.35_{\pm0.68}(-0.13)$ | $19.77$ | $4.05_{\pm0.13}$ | $16.38_{\pm0.70}$ | $84.59$ |
| | ST | $4.55_{\pm1.96}(-25.29)$ | $45.05_{\pm3.88}(-10.66)$ | $99.37_{\pm0.14}(-0.02)$ | $71.27_{\pm1.16}(-1.21)$ | $9.30$ | $3.44_{\pm0.17}$ | $11.17_{\pm1.07}$ | $79.03$ |
| | AM | $2.39_{\pm1.47}(-27.45)$ | $26.08_{\pm12.68}(-29.63)$ | $99.44_{\pm0.12}(0.05)$ | $72.26_{\pm0.42}(-0.22)$ | $14.34$ | $3.73_{\pm0.24}$ | $13.78_{\pm1.79}$ | $84.34$ |
| | OMP | $16.57_{\pm1.42}(-13.27)$ | $75.20_{\pm2.10}(19.49)$ | $97.39_{\pm0.40}(-2.00)$ | $68.80_{\pm0.78}(-3.68)$ | $9.61$ | $3.12_{\pm0.14}$ | $6.24_{\pm0.45}$ | $82.41$ |
| | CS | $24.56_{\pm1.89}(-5.28)$ | $70.51_{\pm3.12}(14.80)$ | $97.63_{\pm0.91}(-1.76)$ | $69.09_{\pm1.25}(-3.39)$ | $6.31$ | $2.46_{\pm0.20}$ | $3.77_{\pm0.63}$ | $83.30$ |
| | POP | $29.54_{\pm1.97}(-0.30)$ | $77.68_{\pm4.15}(21.97)$ | $93.69_{\pm3.16}(-5.70)$ | $67.37_{\pm1.18}(-5.11)$ | $8.27$ | $2.89_{\pm0.19}$ | $4.49_{\pm0.54}$ | $83.37$ |
| ViT | Niave | $24.89_{\pm0.97}(0.00)$ | $46.53_{\pm1.65}(0.00)$ | $99.88_{\pm0.23}(0.00)$ | $76.25_{\pm0.72}(0.00)$ | $0.00$ | $0.00_{\pm0.00}$ | $0.00_{\pm0.00}$ | $0.00$ |
| | GA | $0.04_{\pm0.09}(-24.85)$ | $4.43_{\pm3.26}(-42.10)$ | $99.97_{\pm0.06}(0.09)$ | $76.62_{\pm0.77}(0.37)$ | $16.85$ | $3.53_{\pm0.10}$ | $14.46_{\pm0.52}$ | $86.84$ |
| | FT | $0.10_{\pm0.26}(-24.79)$ | $4.46_{\pm3.45}(-42.07)$ | $99.98_{\pm0.02}(0.10)$ | $76.63_{\pm0.78}(0.38)$ | $16.83$ | $3.52_{\pm0.10}$ | $14.40_{\pm0.56}$ | $87.04$ |
| | ST | $2.16_{\pm0.81}(-22.73)$ | $33.80_{\pm3.75}(-12.73)$ | $99.87_{\pm0.25}(-0.01)$ | $76.19_{\pm0.95}(-0.06)$ | $8.88$ | $3.14_{\pm0.11}$ | $11.17_{\pm0.71}$ | $82.35$ |
| | AM | $0.11_{\pm0.34}(-24.78)$ | $5.39_{\pm4.54}(-41.14)$ | $99.96_{\pm0.08}(0.08)$ | $76.62_{\pm0.76}(0.37)$ | $16.59$ | $3.51_{\pm0.10}$ | $14.34_{\pm0.62}$ | $86.86$ |
| | OMP | $33.89_{\pm1.41}(9.00)$ | $99.49_{\pm0.22}(52.96)$ | $69.13_{\pm1.23}(-30.75)$ | $62.12_{\pm1.09}(-14.13)$ | $26.71$ | $5.08_{\pm0.16}$ | $10.78_{\pm0.56}$ | $86.59$ |
| | CS | $24.19_{\pm1.02}(-0.70)$ | $83.36_{\pm2.07}(36.83)$ | $88.31_{\pm1.26}(-11.57)$ | $71.95_{\pm1.10}(-4.30)$ | $13.35$ | $2.92_{\pm0.16}$ | $4.78_{\pm0.42}$ | $85.90$ |
| | POP | $28.48_{\pm1.80}(3.59)$ | $92.77_{\pm1.40}(46.24)$ | $79.17_{\pm1.09}(-20.71)$ | $69.63_{\pm1.30}(-6.62)$ | $19.29$ | $3.66_{\pm0.14}$ | $6.62_{\pm0.57}$ | $85.97$ |

For MIA Efficacy, CS is often the closest out of the pruning methods to Naive Retraining, followed by POP and OMP. When considering MIA Efficacy, no methods on the VGGish architecture induce the Streisand Effect for SpeechCommands. Whereas, for the CCT and ViT, the Streisand Effect could be identified with the pruning methods on both SpeechCommands and UrbanSounds8K, as they largely exceed the MIA Efficacy reached by $\mathcal{M}_r^\theta$. However, it is important to note that overall OMP causes the most marked Streisand Effect. An interesting relationship exists between UA, TA and RA for the pruning methods, while they consistently reduce the UA disparity gap and have the lowest A DIST and JS DIST, their application can come at a cost to generalisation. Further, this

highlights that OMP may be too-aggressive a pruning strategy, which leads to a severe reduction in accuracy for transformer models. The distance metrics, A DIST and JS DIST, also reveal a concurrent story as they are low for POP and CS across all architectures. Moreover, for the task of 10% Item Removal, POP is the best for UA and second for MIA with low JS DIST values. However, its application comes at a slight cost to RA and TA which could be resolved with further fine tuning.

These results highlight the virtues of the *Prune and Regrow Paradigm* for Item Removal. When considering RTE reduction, all models are essentially equal. However, due to the knowledge distillation setup, unlearning with ST comes at a more substantial computational cost, which can be aligned with the inference required at both the impair and repair stages. In Appendix C.1 and D.1 we present radar plots that emphasises the failure of the non-pruning based methods to reach the UA and MIA of $\mathcal{M}_r^\theta$ with a nuanced relationship emerging between retention of TA and RA combined with the ability to remove $\mathcal{D}_{forget}$ in $\mathcal{M}^\theta$. Moreover, when considering the radar plots, POP and CS emerge as the most holistic unlearning mechanisms for Item Removal in audio, showing that our *Prune and Regrow Paradigm* represents state-of-the-art unlearning capacity in audio.

## 5.2 CLASS REMOVAL

Table 4: **1 Class Removal** results for **SpeechCommands**. Numbers in blue represent disparity from $\mathcal{M}_r^\theta$. $\mathcal{C}$ represents the objective to have the least disparity with $\mathcal{M}_r^\theta$. Otherwise arrows dictate the direction of best performance compared to $\mathcal{M}_r^\theta$.

| Model | Method | UA % ($\mathcal{C}$) | MIA Efficacy % ($\mathcal{C}$) | RA % ($\mathcal{C}$) | TA % ($\mathcal{C}$) | D AVE ($\mathcal{C}$) | A DIST ($\downarrow$) ($\times 10^{-1}$) | JS DIST ($\downarrow$) ($\times 10^{-3}$) | RTE % ($\uparrow$) |
|---|---|---|---|---|---|---|---|---|---|
| | | | | | **1 Class Removal** | | | | |
| VGGish | Niave | $100.00_{\pm 0.00}$(0.00) | $100.00_{\pm 0.00}$(0.00) | $98.48_{\pm 0.55}$(0.00) | $88.09_{\pm 0.20}$(0.00) | 0.00 | $0.00_{\pm 0.00}$ | $0.00_{\pm 0.00}$ | 0.00 |
| | GA | $47.38_{\pm 22.59}$(-52.62) | $62.16_{\pm 19.80}$(-37.84) | $88.22_{\pm 28.15}$(-10.26) | $79.07_{\pm 25.03}$(-9.02) | 27.44 | $9.98_{\pm 1.12}$ | $18.11_{\pm 8.47}$ | 87.79 |
| | FT | $40.25_{\pm 14.82}$(-59.75) | $56.58_{\pm 16.13}$(-43.42) | $97.48_{\pm 1.30}$(-1.00) | $87.45_{\pm 0.57}$(-0.64) | 26.20 | $10.13_{\pm 1.14}$ | $20.64_{\pm 6.25}$ | **87.90** |
| | ST | $96.41_{\pm 7.29}$(-3.59) | $99.95_{\pm 0.15}$(-0.05) | $58.88_{\pm 36.17}$(-39.60) | $56.82_{\pm 34.77}$(-31.27) | 18.63 | $7.54_{\pm 0.75}$ | $0.49_{\pm 1.00}$ | 83.62 |
| | AM | $98.75_{\pm 0.83}$(-1.25) | $99.89_{\pm 0.14}$(-0.11) | $97.78_{\pm 0.95}$(-0.70) | $87.58_{\pm 0.41}$(-0.51) | 0.64 | $7.07_{\pm 0.60}$ | $0.22_{\pm 0.15}$ | 87.85 |
| | OMP | $100.00_{\pm 0.00}$(0.00) | $100.00_{\pm 0.00}$(0.00) | $94.73_{\pm 1.64}$(-3.75) | $87.93_{\pm 0.57}$(-0.16) | 0.98 | $7.03_{\pm 0.47}$ | $0.00_{\pm 0.00}$ | 87.42 |
| | CS | $92.35_{\pm 4.59}$(-7.65) | $98.14_{\pm 1.71}$(-1.86) | $96.52_{\pm 1.14}$(-1.96) | $87.69_{\pm 0.56}$(-0.40) | 2.97 | $7.21_{\pm 0.48}$ | $1.81_{\pm 1.22}$ | 87.00 |
| | POP | $97.83_{\pm 2.74}$(-2.17) | $99.79_{\pm 0.38}$(-0.21) | $96.30_{\pm 1.06}$(-2.18) | $87.82_{\pm 0.40}$(-0.27) | 1.21 | $7.04_{\pm 0.52}$ | $0.44_{\pm 0.57}$ | 87.02 |
| CCT | Niave | $100.00_{\pm 0.00}$(0.00) | $100.00_{\pm 0.00}$(0.00) | $99.93_{\pm 0.02}$(0.00) | $77.84_{\pm 0.32}$(0.00) | 0.00 | $0.00_{\pm 0.00}$ | $0.00_{\pm 0.00}$ | 0.00 |
| | GA | $3.88_{\pm 7.29}$(-96.12) | $34.30_{\pm 15.69}$(-65.70) | $99.70_{\pm 0.56}$(-0.23) | $77.32_{\pm 0.67}$(-0.52) | 40.64 | $12.73_{\pm 0.74}$ | $36.87_{\pm 4.19}$ | 87.63 |
| | FT | $6.31_{\pm 12.16}$(-93.69) | $38.75_{\pm 16.73}$(-61.25) | $99.57_{\pm 1.02}$(-0.36) | $77.30_{\pm 0.64}$(-0.54) | 38.96 | $12.53_{\pm 1.03}$ | $35.67_{\pm 5.91}$ | **87.78** |
| | ST | $99.99_{\pm 0.02}$(-0.01) | $100.00_{\pm 0.00}$(0.00) | $99.61_{\pm 0.32}$(-0.32) | $76.92_{\pm 0.52}$(-0.92) | 0.31 | $6.02_{\pm 0.30}$ | $0.00_{\pm 0.00}$ | 83.45 |
| | AM | $85.03_{\pm 5.69}$(-14.97) | $98.91_{\pm 0.78}$(-1.09) | $99.89_{\pm 0.06}$(-0.04) | $77.57_{\pm 0.35}$(-0.27) | 4.09 | $6.31_{\pm 0.28}$ | $3.83_{\pm 1.42}$ | 87.67 |
| | OMP | $78.67_{\pm 3.57}$(-21.33) | $99.60_{\pm 0.33}$(-0.40) | $93.83_{\pm 0.33}$(-6.10) | $74.77_{\pm 0.41}$(-3.07) | 7.72 | $6.85_{\pm 0.26}$ | $4.78_{\pm 1.06}$ | 86.45 |
| | CS | $84.40_{\pm 5.29}$(-15.60) | $99.61_{\pm 0.39}$(-0.39) | $94.91_{\pm 0.80}$(-5.02) | $74.75_{\pm 0.25}$(-3.09) | 6.02 | $6.32_{\pm 0.29}$ | $3.64_{\pm 1.44}$ | 86.73 |
| | POP | $92.80_{\pm 2.30}$(-7.20) | $99.97_{\pm 0.04}$(-0.03) | $93.02_{\pm 0.67}$(-6.91) | $74.87_{\pm 0.39}$(-2.97) | 4.28 | $5.81_{\pm 0.17}$ | $1.43_{\pm 0.52}$ | 86.81 |
| ViT | Niave | $100.00_{\pm 0.00}$(0.00) | $100.00_{\pm 0.00}$(0.00) | $99.85_{\pm 0.05}$(0.00) | $85.40_{\pm 0.21}$(0.00) | 0.00 | $0.00_{\pm 0.00}$ | $0.00_{\pm 0.00}$ | 0.00 |
| | GA | $4.64_{\pm 8.20}$(-95.36) | $30.42_{\pm 14.89}$(-69.58) | $99.66_{\pm 0.57}$(-0.19) | $84.92_{\pm 0.91}$(-0.48) | 41.4 | $12.62_{\pm 0.80}$ | $37.00_{\pm 4.40}$ | 85.69 |
| | FT | $7.27_{\pm 10.69}$(-92.73) | $34.54_{\pm 18.60}$(-65.46) | $99.05_{\pm 1.57}$(-0.80) | $84.61_{\pm 1.36}$(-0.79) | 39.94 | $12.41_{\pm 0.95}$ | $35.71_{\pm 5.58}$ | **85.79** |
| | ST | $100.00_{\pm 0.00}$(0.00) | $100.00_{\pm 0.00}$(0.00) | $99.85_{\pm 0.05}$(0.00) | $85.43_{\pm 0.26}$(0.03) | 0.01 | $5.66_{\pm 0.34}$ | $0.00_{\pm 0.00}$ | 80.76 |
| | AM | $99.95_{\pm 0.05}$(-0.05) | $100.00_{\pm 0.00}$(0.00) | $99.85_{\pm 0.06}$(0.00) | $85.29_{\pm 0.38}$(-0.11) | 0.04 | $5.74_{\pm 0.35}$ | $0.01_{\pm 0.01}$ | 85.74 |
| | OMP | $92.28_{\pm 3.56}$(-7.72) | $100.00_{\pm 0.00}$(0.00) | $88.91_{\pm 0.45}$(-10.94) | $82.87_{\pm 0.34}$(-2.53) | 5.30 | $6.03_{\pm 0.27}$ | $1.20_{\pm 0.67}$ | 85.28 |
| | CS | $89.66_{\pm 4.46}$(-10.34) | $99.98_{\pm 0.05}$(-0.02) | $94.79_{\pm 0.46}$(-5.06) | $83.62_{\pm 0.60}$(-1.78) | 4.30 | $6.11_{\pm 0.38}$ | $2.15_{\pm 1.06}$ | 84.69 |
| | POP | $95.02_{\pm 1.82}$(-4.98) | $100.00_{\pm 0.00}$(0.00) | $91.86_{\pm 0.82}$(-7.99) | $83.24_{\pm 0.55}$(-2.16) | 3.78 | $5.81_{\pm 0.24}$ | $0.86_{\pm 0.38}$ | 84.82 |

Table 5: **1 Class Removal** results for **UrbanSounds8K**. Numbers in blue represent disparity from $\mathcal{M}_r^\theta$. $\mathcal{C}$ represents the objective to have the least disparity with $\mathcal{M}_r^\theta$. Otherwise arrows dictate the direction of best performance compared to $\mathcal{M}_r^\theta$.

| Model | Method | UA % ($\mathcal{C}$) | MIA Efficacy % ($\mathcal{C}$) | RA % ($\mathcal{C}$) | TA % ($\mathcal{C}$) | D AVE ($\mathcal{C}$) | A DIST ($\downarrow$) ($\times 10^{-1}$) | JS DIST ($\downarrow$) ($\times 10^{-3}$) | RTE % ($\uparrow$) |
|---|---|---|---|---|---|---|---|---|---|
| | | | | | **1 Class Removal** | | | | |
| VGGish | Niave | $100.00_{\pm 0.00}$(0.00) | $100.00_{\pm 0.00}$(0.00) | $96.65_{\pm 0.94}$(0.00) | $80.22_{\pm 0.57}$(0.00) | 0.00 | $0.00_{\pm 0.00}$ | $0.00_{\pm 0.00}$ | 0.00 |
| | GA | $62.46_{\pm 24.47}$(-37.54) | $74.54_{\pm 22.08}$(-25.46) | $91.69_{\pm 6.66}$(-4.96) | $76.36_{\pm 4.33}$(-3.86) | 17.95 | $8.83_{\pm 1.48}$ | $19.74_{\pm 14.96}$ | 88.83 |
| | FT | $58.11_{\pm 24.24}$(-41.89) | $70.66_{\pm 23.22}$(-29.34) | $95.10_{\pm 3.68}$(-1.55) | $78.33_{\pm 1.54}$(-1.89) | 18.67 | $8.96_{\pm 1.64}$ | $22.18_{\pm 14.90}$ | **89.36** |
| | ST | $97.96_{\pm 4.51}$(-2.04) | $89.97_{\pm 29.99}$(-10.03) | $78.64_{\pm 26.17}$(-18.01) | $67.80_{\pm 21.28}$(-12.42) | 10.62 | $6.71_{\pm 0.64}$ | $0.78_{\pm 1.17}$ | 79.79 |
| | AM | $78.55_{\pm 13.23}$(-21.45) | $90.12_{\pm 9.22}$(-9.88) | $94.92_{\pm 2.01}$(-1.73) | $77.92_{\pm 1.41}$(-2.03) | 8.77 | $7.54_{\pm 0.89}$ | $9.64_{\pm 6.63}$ | 89.26 |
| | OMP | $100.00_{\pm 0.00}$(0.00) | $100.00_{\pm 0.00}$(0.00) | $81.74_{\pm 4.92}$(-14.91) | $72.40_{\pm 3.58}$(-7.82) | 5.68 | $6.75_{\pm 0.39}$ | $0.04_{\pm 0.01}$ | 86.91 |
| | CS | $99.82_{\pm 0.53}$(-0.18) | $99.85_{\pm 0.44}$(-0.15) | $89.14_{\pm 10.17}$(-7.51) | $75.88_{\pm 7.01}$(-4.34) | 3.04 | $6.67_{\pm 0.60}$ | $0.22_{\pm 0.49}$ | 87.66 |
| | POP | $100.00_{\pm 0.00}$(0.00) | $100.00_{\pm 0.00}$(0.00) | $87.36_{\pm 8.55}$(-9.29) | $75.06_{\pm 5.92}$(-5.16) | 3.61 | $6.79_{\pm 0.65}$ | $0.04_{\pm 0.03}$ | 87.87 |
| CCT | Niave | $100.00_{\pm 0.00}$(0.00) | $100.00_{\pm 0.00}$(0.00) | $99.44_{\pm 0.27}$(0.00) | $74.00_{\pm 0.71}$(0.00) | 0.00 | $0.00_{\pm 0.00}$ | $0.00_{\pm 0.00}$ | 0.00 |
| | GA | $0.62_{\pm 1.38}$(-99.38) | $36.62_{\pm 14.63}$(-63.38) | $99.20_{\pm 0.86}$(-0.24) | $72.11_{\pm 1.43}$(-1.89) | 41.22 | $12.97_{\pm 0.27}$ | $61.24_{\pm 2.43}$ | 84.88 |
| | FT | $8.58_{\pm 18.42}$(-91.42) | $45.43_{\pm 24.24}$(-54.57) | $98.32_{\pm 2.46}$(-1.12) | $71.66_{\pm 1.30}$(-2.39) | 37.38 | $12.28_{\pm 1.60}$ | $54.99_{\pm 14.48}$ | **85.07** |
| | ST | $90.50_{\pm 13.54}$(-9.50) | $99.62_{\pm 0.90}$(-0.38) | $99.40_{\pm 0.15}$(-0.04) | $72.29_{\pm 1.00}$(-1.71) | 2.91 | $4.97_{\pm 0.78}$ | $3.57_{\pm 5.79}$ | 79.60 |
| | AM | $17.60_{\pm 17.01}$(-82.40) | $80.87_{\pm 11.73}$(-19.13) | $99.17_{\pm 0.90}$(-0.27) | $71.92_{\pm 1.41}$(-2.08) | 25.97 | $10.78_{\pm 1.56}$ | $43.65_{\pm 12.08}$ | 82.40 |
| | OMP | $89.97_{\pm 2.47}$(-10.03) | $100.00_{\pm 0.00}$(0.00) | $97.51_{\pm 0.43}$(-1.93) | $69.81_{\pm 0.77}$(-4.19) | 4.04 | $5.53_{\pm 0.28}$ | $3.10_{\pm 0.92}$ | 83.13 |
| | CS | $99.29_{\pm 0.65}$(-0.71) | $100.00_{\pm 0.00}$(0.00) | $97.16_{\pm 1.56}$(-2.28) | $70.07_{\pm 1.01}$(-3.93) | 1.73 | $4.15_{\pm 0.23}$ | $0.32_{\pm 0.25}$ | 83.89 |
| | POP | $99.85_{\pm 0.15}$(-0.15) | $100.00_{\pm 0.00}$(0.00) | $94.76_{\pm 2.45}$(-4.68) | $69.51_{\pm 1.19}$(-4.49) | 2.33 | $4.36_{\pm 0.33}$ | $0.10_{\pm 0.06}$ | 83.89 |
| ViT | Niave | $100.00_{\pm 0.00}$(0.00) | $100.00_{\pm 0.00}$(0.00) | $99.84_{\pm 0.16}$(0.00) | $77.04_{\pm 0.99}$(0.00) | 0.00 | $0.00_{\pm 0.00}$ | $0.00_{\pm 0.00}$ | 0.00 |
| | GA | $0.30_{\pm 0.42}$(-99.70) | $28.80_{\pm 9.27}$(-71.20) | $99.98_{\pm 0.01}$(0.14) | $76.47_{\pm 0.74}$(-0.57) | 42.9 | $13.24_{\pm 0.17}$ | $62.30_{\pm 1.21}$ | 86.81 |
| | FT | $0.68_{\pm 1.67}$(-99.32) | $28.45_{\pm 11.28}$(-71.55) | $99.97_{\pm 0.03}$(0.13) | $76.44_{\pm 0.83}$(-0.60) | 42.9 | $13.18_{\pm 0.39}$ | $61.96_{\pm 2.65}$ | **86.91** |
| | ST | $99.29_{\pm 0.46}$(-0.71) | $100.00_{\pm 0.00}$(0.00) | $99.90_{\pm 0.16}$(0.06) | $76.78_{\pm 0.93}$(-0.26) | 0.26 | $3.68_{\pm 0.28}$ | $0.37_{\pm 0.20}$ | 82.07 |
| | AM | $51.54_{\pm 6.03}$(-48.46) | $91.16_{\pm 1.80}$(-8.84) | $99.92_{\pm 0.16}$(0.08) | $76.53_{\pm 0.77}$(-0.51) | 14.47 | $7.51_{\pm 0.58}$ | $23.66_{\pm 3.37}$ | 86.81 |
| | OMP | $100.00_{\pm 0.00}$(0.00) | $100.00_{\pm 0.00}$(0.00) | $69.91_{\pm 1.27}$(-29.93) | $62.93_{\pm 1.18}$(-14.11) | 11.01 | $6.07_{\pm 0.28}$ | $0.07_{\pm 0.01}$ | 86.62 |
| | CS | $99.97_{\pm 0.09}$(-0.03) | $100.00_{\pm 0.00}$(0.00) | $87.23_{\pm 2.10}$(-12.61) | $71.27_{\pm 1.96}$(-5.77) | 4.60 | $4.39_{\pm 0.44}$ | $0.07_{\pm 0.02}$ | 85.91 |
| | POP | $100.00_{\pm 0.00}$(0.00) | $100.00_{\pm 0.00}$(0.00) | $79.81_{\pm 1.15}$(-20.03) | $70.54_{\pm 1.49}$(-6.50) | 6.63 | $4.87_{\pm 0.23}$ | $0.05_{\pm 0.01}$ | 85.96 |

When considering Class Removal results displayed in Table's 4 and 5 , we observe that GA and FT perform poorly on UA, suggesting that they cannot unlearn in the Class regime; therefore, they are excluded from further analysis.

Contrary to the results for Item Removal, ST and AM have an increased capacity to unlearn $\mathcal{D}_{forget}$ and often perform well across all metrics. For SpeechCommands it can be noted that ST performs the best for UA (2/3) and for UrbanSounds8K POP performs the best for UA (3/3).

While it does not perfrom the best, OMP is a competetive unlearning method for Class Removal but is ultimately superseded by ST and POP for SpeechCommands and UrbanSounds8K. While OMP also attains strong results, it degrades the TA more than POP, reiterating that the one-size-fits-all approach of OMP is inadequate. However, when considering the transformers, ST is the best method for unlearning across most accuracy and distance metrics in tandem with an increase in the effectiveness of AM.

The divergence in UA for Class Removal highlights the dichotomy between CS and POP. POP removes $\mathcal{D}_{forget}$ from $\mathcal{M}^-$, and alludes to the fact that a less functionally similar prune strategy is more effective for these requests, but pruning too much and not regrowing, as with OMP, is detrimental for accuracy. The *Prune and Regrow* notion is further strengthened and validated as POP almost always outperforms OMP for Class Removal. Overall, the radar plots in Appendix C.1 and D.1 shows ST constantly reaches the boundaries of $\mathcal{M}_r^\theta$ for the CCT and ViT with AM for SpeechCommands. The radar plots especially highlight that there does not appear to be such a nuanced relationship between Class Removal and accuracy degradation as there is for Item Removal.

## 5.3 UNLEARNING REQUEST SCALING

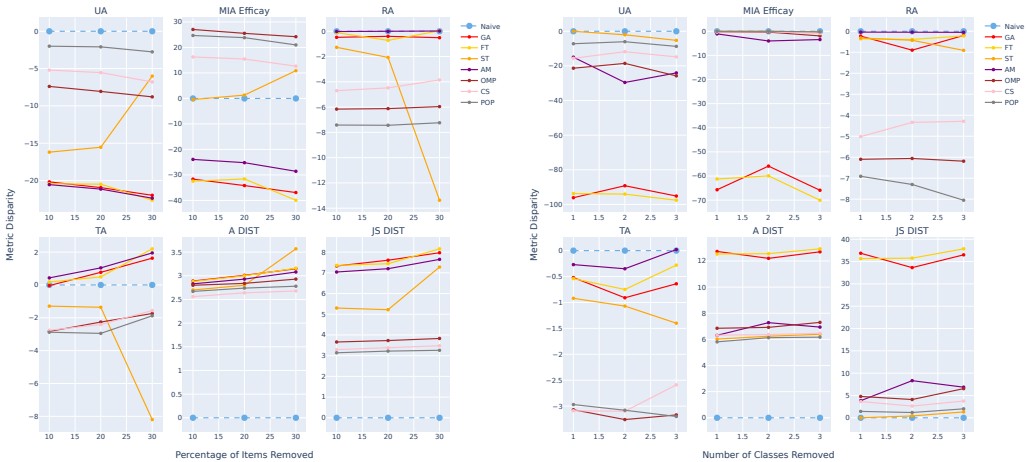

Figure 3: Unlearning efficacy scaling on **SpeechCommands** when considering disparity from the $\mathcal{M}_r^\theta$ for the **CCT**. Item Removal: 10%, 20% and 30% (left) and Class Removal: 1, 2 and 3 (right) the figures for the VGGish and ViT are presented in Appendix Section C.2.

Understanding the efficacy of current and novel unlearning methods as unlearning requests scale is essential. Figures 3 and 4 shows that each unlearning method's impacts are largely stable for Item Removal at 10%, 20% and 30%. Overall for both datasets the transformer architectures are the most robust to increased Item Removal requests compared to the VGGish. When observing the Class Removal scaling of 1, 2, and 3 classes, a similar trend is witnessed concerning the stability of the unlearning methods at scale. The stability of unlearning at scale in the transformer architectures could be linked to the fact that they are more over-parameterised than the VGGish architecture. However, further study would be necessary to make any conclusions on this. For Item Removal on the CCT, POP is the most robust to unlearning request scaling for both datasets; for Class Removal, ST is the best for SpeechCommands and POP is the best for UrbanSounds8K in Figures 3 and 4. Therefore, these results underscore the ability to comply with increased unlearning demands in the audio domain.

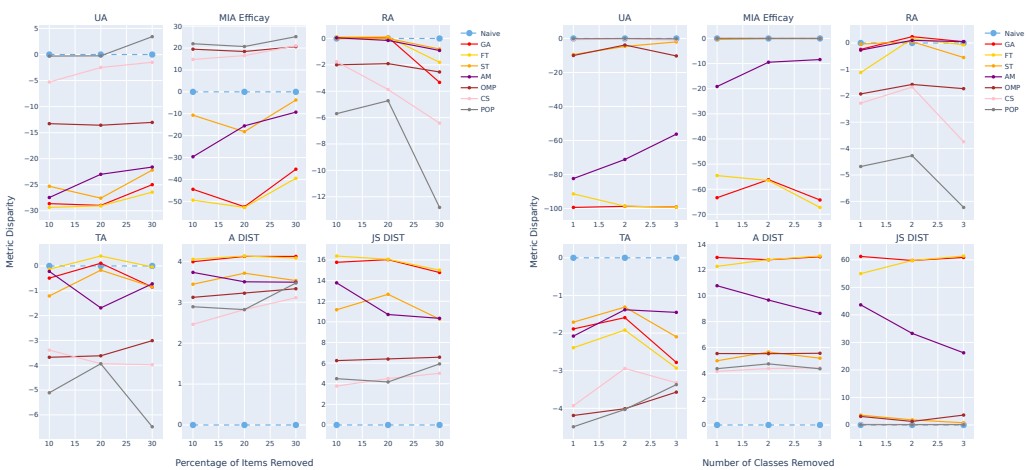

Figure 4: Unlearning efficacy scaling on **UrbanSounds8K** when considering disparity from the $\mathcal{M}_r^\theta$ for the **CCT**. Item Removal: 10%, 20% and 30% (left) and Class Removal: 1, 2 and 3 (right) the figures for the VGGish and ViT are presented in the Appendix in Section D.2.

## 5.4 LOSS DISTRIBUTION ANALYSIS

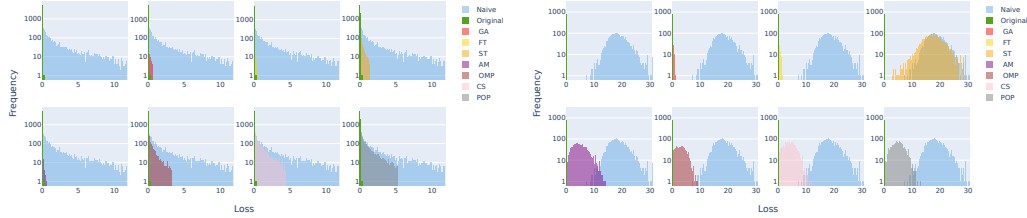

Figure 5: $\mathcal{D}_{forget}$ loss distribution on **SpeechCommands**, for unlearning methods averaged across all seeds for the **CCT**. 10% Item Removal (left) and 1 Class Removal (right). For each plot the unlearning method is compared to the loss distribution of $\mathcal{D}_{forget}$ on $\mathcal{M}^\theta$ and $\mathcal{M}_r^\theta$. The results for the **VGG** and **ViT** are presented in the Appendix C.3

To gain a nuanced insight into the dynamics of unlearning for audio, we probe the change of behaviours of $\mathcal{M}^\theta$ to $\mathcal{M}^-$ compared to $\mathcal{M}_r^\theta$. The loss distribution on $\mathcal{D}_{forget}$ for $\mathcal{M}^\theta$, $\mathcal{M}^-$, and $\mathcal{M}_r^\theta$ is leveraged to provide this. To produce this analysis, $D_{forget}$ is passed through $\mathcal{M}^\theta$, $\mathcal{M}^-$ and $\mathcal{M}_r^\theta$, and the loss for each is plotted as a histogram, allowing for a direct comparison of the loss distribution for each unlearning method. An effective unlearning method should be able to match a loss distribution of $\mathcal{M}_r^\theta$ and, therefore, would be dissimilar to $\mathcal{M}^\theta$ on $D_{forget}$.

Figures 5 and 6 show that, for Item Removal requests, POP shifts the loss distribution so that $\mathcal{M}^-$ resembles the loss distribution of $\mathcal{M}_r^\theta$. The visual depiction reaffirms the understanding that POP is the best Item Removal unlearning method and offers deeper insights into why it performs so well. The loss distributions reveal similar insights when considering the Class Removal loss distribution shift for $\mathcal{D}_{forget}$ in Figure 5 and 6, explains why some of the non-pruning methods excel. The non-pruning methods separate the loss values to shift them to a separated distribution, resulting in a low UA gap. However, this could show that they enforce incorrect memorisation over removal, as a similar trend is not witnessed for Item Removal. Tracking the loss this way highlights the nuances between OMP, CS and POP. In every loss distribution plot for OMP, it can be observed

that it has a more dense frequency of towards $\mathcal{M}^\theta$. An explanation could be that it is harder to increase loss on samples without employing the regrowth strategy when the function is restricted to a smaller portion of the network. The *Prune and Regrow strategy* for POP manifests as a loss distribution that fits within a possible distribution of $\mathcal{M}_r^\theta$. The loss plot figures show that none of the unlearning methods exceed the loss of $\mathcal{M}_r^\theta$ on $\mathcal{D}_{forget}$. It could be argued that any point which exceeds the loss of $\mathcal{M}_r^\theta$ on $\mathcal{D}_{forget}$ would induce the Streisand Effect. Therefore, by this definition, the Streisand Effect is not induced by these methods and could instead be an artifact of the black-box MIA. Subsequently, this prompts further inquiry into the existence of the Streisand Effect in machine unlearning.

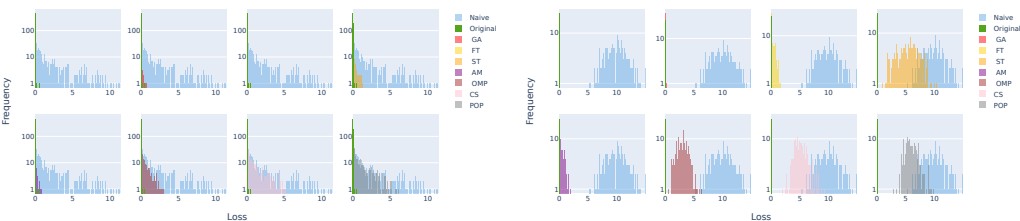

Figure 6: $\mathcal{D}_{forget}$ loss distribution on **UrbanSounds8K**, for unlearning methods averaged across all seeds for the **CCT**. 10% Item Removal (left) and 1 Class Removal (right). For each plot the unlearning method is compared to the loss distribution of $\mathcal{D}_{forget}$ on $\mathcal{M}^\theta$ and $\mathcal{M}_r^\theta$. The results for the **VGG** and **ViT** are presented in the Appendix D.3

## 6 CONCLUSION

Our paper is the first to comprehensively analyse the current state-of-the-art, strong machine unlearning techniques to lay the foundations and advance privacy endeavours within the audio domain for Item and Class Removal. Given that no other such studies exist for audio, our work represents the first of its kind. Our results show that current unlearning methods are partially effective for the most likely request, Item Removal, on lower complexity learning tasks such as AudioMNIST but struggle to transfer to higher-complexity tasks such as SpeechCommands and UrbanSounds8K. Our study introduces Cosine and Post Optimal Prune unlearning, using our novel *Prune and Regrow Paradigm* to address this. Post Optimal Prune was identified as a superior method for Item Removal across all datasets and architectures, regardless of request scaling, signifying an important step towards upholding privacy in the audio domain. Additionally it provides very competitive and consistent class unlearning capabilities. Through the *Prune and Regrow Pardigm* we champion unlearning methods that are dynamic to architecture; modality and enable repeated unlearning.

Despite the lack of consistent performance of current methods for Item Removal, Stochastic Teacher and Amnesiac unlearning successfully fulfill Class Removal requests on higher task complexity. However, these results may be related to memorising incorrect representations rather than causing direct unlearning. The results mandate further development of existing and novel methods to realise unlearning capabilities in audio. Our unique analysis of the scaling of machine unlearning methods uncovered that, for Item Removal, the most important unlearning case, dynamic unlearning approaches scale the best, while, for Class removal, scaling properties are often shared between effective methods. Furthermore, loss distribution analysis for Item and Class Removal revealed that the Streisand Effect may be a red herring caused by the reliance on black-box evaluation metrics, which requires further exploration.

In summary, this paper contributes a nuanced and novel understanding of machine unlearning within audio and provides two new state-of-the-art methods for unlearning via the *Prune and Regrow Paradigm*, improving privacy through removal fulfilment for Item Removal, enabling synergy between privacy and the application of deep learning in the audio domain and beyond.

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

# A    CURRENT UNLEARNING METHODS

Table 6: Evaluation of existing strong machine unlearning methods.

| Method | Definition | Advantages | Limitations |
|---|---|---|---|
| Gradient Ascent (GA) | Perform a loss maximisation operation (GA) for each mini-batch within the forget set then repair the model through fine-tuning on the remain set. | • Is an **intuitive method**.
• **Computationally inexpensive**.
• Actively **removes learnt representations** by targeting $\mathcal{D}_{forget}$. | • **Sensitive** to learning rate and requires hyper-parameter tuning to get the best results.
• **Less principled** than other methods. |
| Fine-tuning (FT) | Fine tuning on $\mathcal{D}_{remain}$ to initiate catastrophic forgetting to remove $\mathcal{D}_{forget}$. | • Can be used when **access to the original training dataset is limited**.
• Used **in conjunction with other methods** to improve unlearning. | • Often requires **more epochs of training** to evoke forgetting.
• Without impair step it is **hard to evoke catastrophic forgetting**. |
| Stochastic/ Incompetent Teacher (ST) | Use a stochastic teacher to remove data on $\mathcal{D}_{forget}$ and then use the original model to repair performance on $\mathcal{D}_{remain}$. | • Uses knowledge distillation that leads to **intuitive understanding**.
• Use of original model aids **simple implementation**. | • Inference at impair and repair steps and is **computationally expensive**.
• Literature shows weak functional preservation in knowledge distillation possibly showing it is **unprincipled** Mason-Williams (2024). |
| Amnesiac (AM) | Assign randomly incorrect labels to $\mathcal{D}_{forget}$ and minimise the loss to optimise for the incorrect labels. This is then followed by a fine tuning step on $\mathcal{D}_{remain}$. | • Employs **computationally inexpensive** method to generate randomly incorrect labels.
• Actively **removes learnt representations** of $\mathcal{D}_{forget}$. | • Forces the model to learn **incorrect representation** for $\mathcal{D}_{forget}$ so only obfuscates $\mathcal{D}_{forget}$ over unlearning it.
• Impacts decision boundaries of classes leading to **less robust predictions**. |
| One-Shot Magnitude Prune (OMP) | Prune the original model to 95% sparsity keeping only the most salient weights followed by fine tuning on $\mathcal{D}_{remain}$ to recover accuracy. | • Has shown to **drastically improve FT** and can be used in conjunction with other methods.
• Has **strong theoretical backing** with links to the Lottery Ticket Hypothesis. | • **Sparsity of 95%** is **based on empirical evidence only**.
• May be **too harsh** on some architectures.
• **Reduced unlearning budget** meaning less repeated unlearning. |

## B  FURTHER TRAINING DETAILS

**Architecture details:**   Table 7 shows the varying parameter scales that were employed to achieve similar baseline accuracy for each of the architectures on the respective datasets.

Table 7: Architectures used for machine unlearning exploration.

| Architecture | Trainable Parameters |
|---|---|
| VGGish | 4,839,075 |
| Compact Convolutional Transformers (CCT) | 10,531,625 |
| Vision Transformer (ViT) | 11,659,875 |

**Unlearning details:**   SGD optimises all impair step optimisations with momentum=0.9, learning rate=0.01 and batch size=256.   However, for GA the learning rate is reduced to lr = $(0.01/(|\mathcal{D}_{forget}|/256))$. Preliminary experiments showed that once GA exceeded one mini-batch update with a learning rate of 0.01, it became impossible to recover accuracy on $\mathcal{D}_{remain}$, so this intervention was made to stabilise the impact of GA. While in the image domain, a learning rate of 0.01 (Golatkar et al., 2020b) - 0.0001 (Liu et al., 2024) has shown to be successful for GA when using SGD; this was not the case during experimental analysis across all audio datasets. For the experiment of CIFAR10 we use the standard learning rate of 0.01.

For all repair step optimisations, SGD is the optimiser with momentum=0.9, learning rate=0.01, and batch size=256 - in line with experiments conducted in the vision domain (Liu et al., 2024). The unlearning methods are applied to each $\mathcal{M}^\theta$ and compared to the corresponding $\mathcal{M}_r^\theta$ and are **averaged across five independent experiments**.

The experiments are conducted across three scales for Item and Class Removal requests to assess the capabilities of current and novel unlearning methods comprehensively. For Item Removal, 10%, 20%, and 30% of random data from $\mathcal{D}_{train}$ is removed, and for Class Removal, 1, 2, and 3 random classes are removed. For Class Removal, it is noted that the classes to be removed are also removed from the test set.  Understanding how each method scales to a more complex unlearning request provides better insights into the robustness of each method and confirms the efficacy of current and novel unlearning methods in the audio domain.

**Evaluation metric details:**   For all accuracy-based metrics, the accuracy of $\mathcal{M}^-$ is reported, as well as the disparity between $\mathcal{M}^-$ and $\mathcal{M}_r^\theta$ on $\mathcal{D}_{forget}$ (UA), $\mathcal{D}_{remain}$ (RA) and $\mathcal{D}_{test}$ (TA). It is important to highlight that UA represents $1-\mathcal{M}^-(\mathcal{D}_{forget})$. Disparity Average (D AVE) is the average disparity across UA, RA, TA and MIA. For Activation Distance (A DIST) and Jensen Shannon Divergence (JS DIST), the distance is compared between the $\mathcal{M}_r^\theta$ and $\mathcal{M}^-$ outputs for $\mathcal{D}_{forget}$ on the respective softmax and loss outputs for each respective metric. RTE is reported as the reduction of time as a percentage of creating $\mathcal{M}^-$ against the time required to train $\mathcal{M}_r^\theta$ as it is more intuitive than providing the raw time duration; as a result a higher RTE percentage is preferable.

To perform the membership inference attack, in line with other literature (Liu et al., 2024; Graves et al., 2021), the attack method introduced by Shokri et al. (2017), described in Section 2.3, is used. Following the implementation of (Liu et al., 2024), the training datasets for the attack model, $\mathcal{M}_a^\theta$, were composed of a balanced dataset of the baseline models outputs on $D_{test}$ and $D_{train}$ for each of the five baseline models for each architecture and dataset. Three independent $\mathcal{M}_a^\theta$ are trained based on the loss outputs for each architecture and dataset. The attack models are trained for 50 epochs with early stopping.

Table 8: Machine unlearning evaluation metrics employed for strong machine unlearning experiments

| Evaluation Metric | Formula/Description | Category | Related Literature |
|---|---|---|---|
| Unlearning Accuracy (UA) | $1 - acc(\mathcal{D}_{forget})$ | Evaluating predictive distribution | (Chundawat et al., 2023a; Tarun et al., 2023; Golatkar et al., 2020b; Liu et al., 2024; Chundawat et al., 2023b) |
| Remaining Accuracy (RA) | $acc(\mathcal{D}_{remain})$ | Evaluating predictive distribution | (Chundawat et al., 2023a; Tarun et al., 2023; Golatkar et al., 2020b; Liu et al., 2024; Chundawat et al., 2023b) |
| Testing Accuracy (TA) | $acc(D_{test})$ | Evaluating predictive distribution | (Golatkar et al., 2020b; Liu et al., 2024; Chundawat et al., 2023b) |
| MIA Efficacy (MIA) | $\frac{TrueNegatives}{|\mathcal{D}_{forget}|}$ | Evaluating attack success | (Graves et al., 2021; Liu et al., 2024) |
| Disparity Average (D AVE) | $(\mathcal{M}_u^\theta(UA) - \mathcal{M}^-(UA) + \mathcal{M}_u^\theta(RA) - \mathcal{M}^-(RA)$ $+ \mathcal{M}_u^\theta(TA) - \mathcal{M}^-(TA) + \mathcal{M}_u^\theta(MIA) - \mathcal{M}^-(MIA))/4$ | Evaluating predictive distribution | (Liu et al., 2024) |
| Activation Distance (A DIST) | $L_2(\mathcal{M}^\theta(\mathcal{D}_{forget}), \mathcal{M}^-(\mathcal{D}_{forget}))$ | Similarity of unlearn distribution | (Chundawat et al., 2023a) |
| Jensen-Shannon Divergence (JS DIST) | $0.5 \cdot KL(\mathcal{M}_u^\theta(\mathcal{D}_{forget}), (\mathcal{M}_u^\theta(\mathcal{D}_{forget}) - \mathcal{M}^-(\mathcal{D}_{forget}))) +$ $0.5 \cdot KL(\mathcal{M}^-(\mathcal{D}_{forget}) - (\mathcal{M}_u^\theta(\mathcal{D}_{forget}) - \mathcal{M}^-(\mathcal{D}_{forget})))$ | Similarity of unlearn distribution | (Chundawat et al., 2023a) |
| Run-Time Efficiency (RTE) | $\left| \frac{\mathcal{M}_u^\theta(\mathcal{T}_{train}) - \mathcal{M}^-(\mathcal{T}_{impair} + \mathcal{T}_{repair})}{\mathcal{M}_u^\theta(\mathcal{T}_{train})} \right| \times 100$ | Comparative unlearning time | (Tarun et al., 2023; Liu et al., 2024) |

## C SPEECHCOMMANDS

In this section we present the radar plots for both Item and Class Removal for the SpeechCommands dataset, the plots highlight the interactions between UA, MIA Efficacy, TA and RA. Overall it can be noted that for Item Removal there is a distinction between methods that perform well at unlearning and a reduction in TA and RA compared to methods that perform worst on UA. However, this distinction is not apparent for Class Removal; there is little generalisation cost for methods that perform well on UA.

Additionally, we present the scaling results for the VGGish and ViT architectures, they show how the unlearning methods perform as the amount of Item's and Classes to remove increases. We see that most methods retain their performance as Item and Class Removal requests scale.

Finally, the loss distributions are presented for the VGGish and ViT architectures for both Item and Class Removal.

### C.1 RADAR PLOTS

For the radar plots on the VGGish, CCT and ViT architectures there is generally a trend that methods that match the Naive model on UA result in a trade off in generalization. For the CCT and ViT this is most apparent for example POP which performs best of UA for the CCT and ViT it often has a higher MIA Efficacy and lower ability to retain RA and TA. The same is true for both POP and OMP. This emphasises that unlearning sometimes results in a degradation in performance. It would be of interest in future work to explore how many epochs of fine tuning would be required to completely restore accuracy that is degraded. It is important to note that the Prune and Regrow methods perform better overall at recovering RA and TA which speaks to the success of the regrow phase of the paradigm.

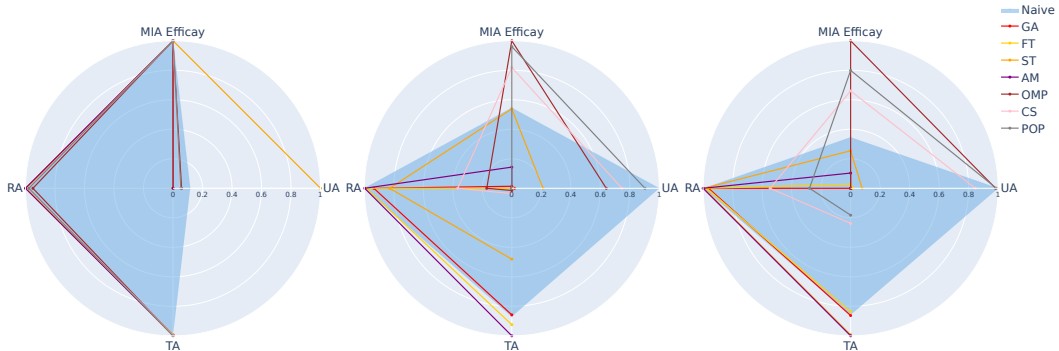

Figure 7: **10% Item Removal** radar plots on unlearning metrics based on min-max normalisation for **SpeeechCommands**: VGGish (left), CCT (middle), and ViT (right).

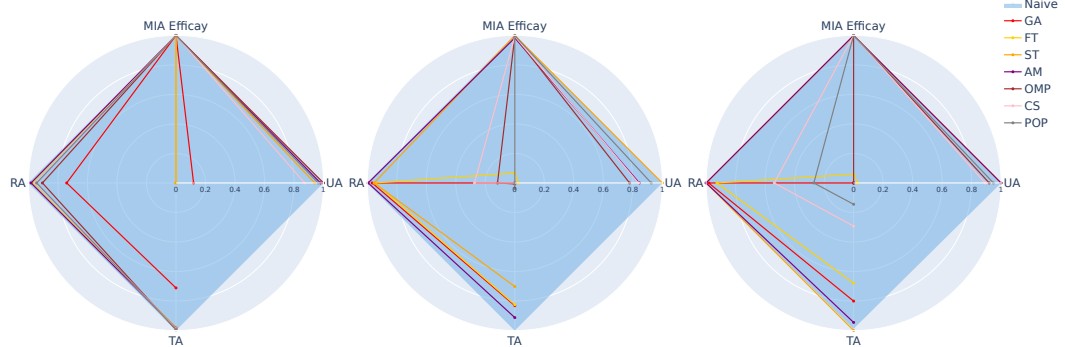

Figure 8: **1 Class Removal** radar plots on unlearning metrics based on min-max normalisation for **SpeeechCommands**: VGGish (left), CCT (middle), and ViT (right).

For Class Removal there is less of a trade-off between UA, MIA Efficacy, RA and TA. The non-pruning methods appear to balance all of the factor equally in application. For the pruning methods it can still be observed that the trade off is in place so while they perform well for UA they would require more training to be truly competitive to the non-pruning based methods for Class Removal overall on SpeechCommands. However, it should be noted that CS and POP usually recover better than POP on RA and TA compared to POP which yet again speaks to the ability to recover accuracy given the regrow phase of the Prune and Regrow Paradigm.

## C.2 REQUEST SCALLING

As the proportion of unlearning requests scale it can be observed that most of the methods have a stable impact across key metrics such as UA, MIA Efficacy and RA for Item Removal and Class Removal. Therefore the analysis matches that presented in the main body.

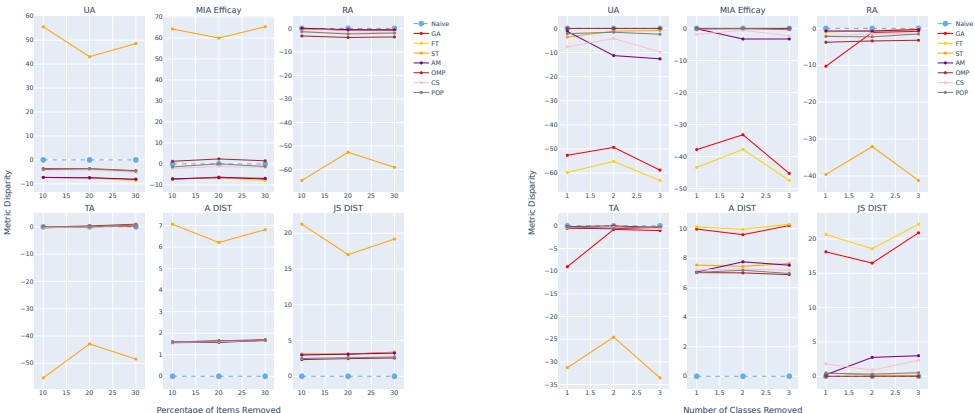

Figure 9: Unlearning efficacy scaling on **SpeechCommands** when considering disparity from the $\mathcal{M}_r^\theta$ (dotted line) for the **VGGish**. With Item Removal: 10%, 20% and 30% (left) and Class Removal: 1, 2 and 3 (right).

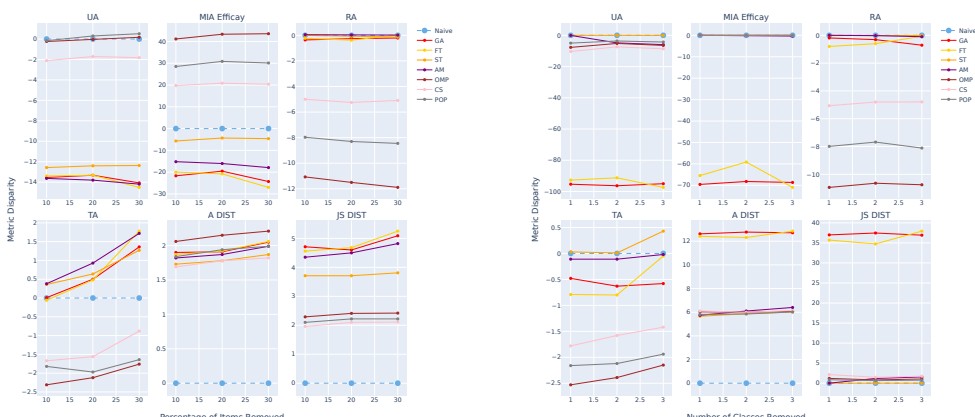

Figure 10: Unlearning efficacy scaling on **SpeechCommands** when considering disparity from the $\mathcal{M}_r^\theta$ (dotted line) for the **ViT**. With Item Removal: 10%, 20% and 30% (left) and Class Removal: 1, 2 and 3 (right).

### C.3 LOSS DISTRIBTUIONS

For SpeechCommands we see that for both Item and Class removal across the VGGish and ViT architectures that the methods which have the lowest UA disparity gap often have a close loss distribution to that of the Naive model on the forget set. For the VGGish on Item Removal the best method for matching the loss distribution appears to be OMP and for the ViT it is POP. For Class removal the best method appears to be OMP for the VGGish and ST joint with AM for the ViT.

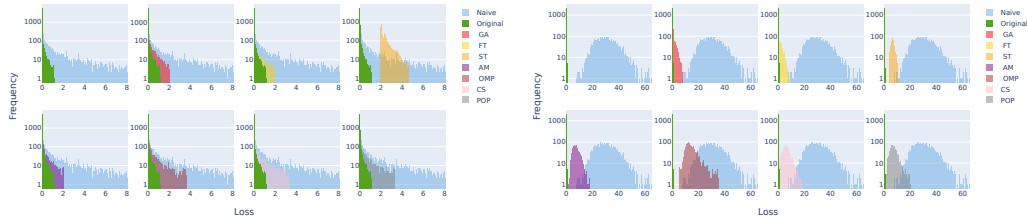

Figure 11: $\mathcal{D}_{forget}$ loss distribution on **SpeechCommands**, for unlearning methods averaged across all seeds for **VGGish**. 10% Item Removal (left) and 1 Class Removal (right). For each plot the unlearning method is compared to the loss distribution of $\mathcal{D}_{forget}$ on $\mathcal{M}^\theta$ and $\mathcal{M}_r^\theta$

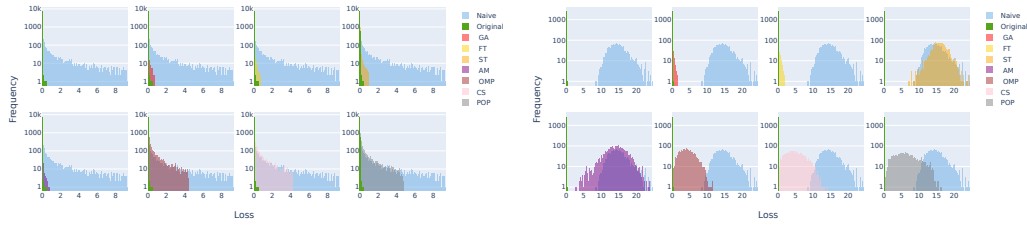

Figure 12: $\mathcal{D}_{forget}$ loss distribution on **SpeechCommands**, for unlearning methods averaged across all seeds for the **ViT**. 10% Item Removal (left) and 1 Class Removal (right). For each plot the unlearning method is compared to the loss distribution of $\mathcal{D}_{forget}$ on $\mathcal{M}^\theta$ and $\mathcal{M}_r^\theta$.

# D URBANSOUNDS8K

In this section, we present the radar plots for both Item and Class Removal for the UrbanSounds8K dataset; the plots highlight the interactions between UA, MIA Efficacy, TA and RA. Overall, for Item Removal, there is a distinction between methods that perform well at unlearning and a reduction in TA and RA compared to methods that perform worst on UA. However, this distinction is not apparent for Class Removal; there is little generalisation cost for methods that perform well on UA.

Additionally, we present the scaling results for the VGGish and ViT architectures; they show how the unlearning methods perform as the number of Items and Classes to remove increases. We see that most methods apart from ST retain their performance as Item and Class Removal requests scale.

Finally, the loss distributions are presented for the VGGish and ViT architectures for both Item and Class Removal.

## D.1 RADAR PLOTS

When examining the radar plots on UrbanSounds8k, it becomes clear that a trade-off similar to the one observed for Item Removal on SpeechCommands exists. The trade-off indicates that methods that perform well on UA often exceed the MIA Efficacy while also experiencing a reduction for RA and TA. In the context of Item Removal on the CCT and ViT architecture, CS emerges as the most comprehensive unlearning method. It is capable of recovering more accuracy than POP when considering RA and RA, while still maintaining a high performance on UA.

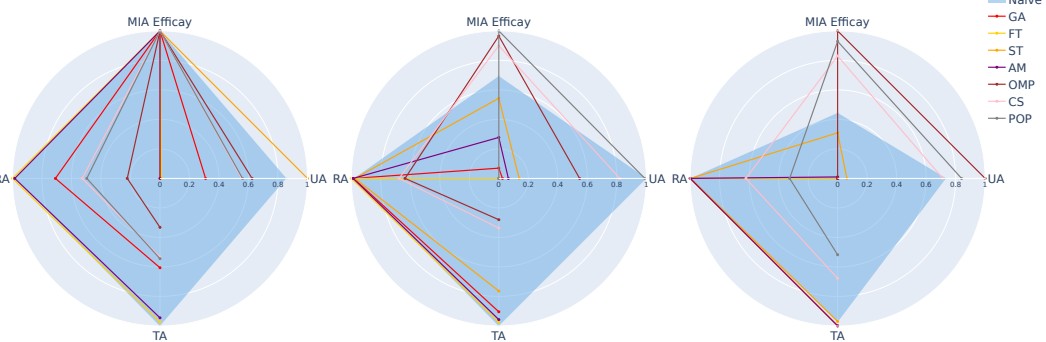

Figure 13: **10% Item Removal** radar plots on unlearning metrics based on min-max normalisation for **UrbanSounds8K**: VGGish (left), CCT (middle), and ViT (right).

When we consider Class removal, a distinct trend on UrbanSounds8K emerges. Methods that perform well also incur a slight trade-off in generalisation, a unique characteristic of UrbanSounds8K. This finding suggests that there are instances where more fine-tuning is required to recover accuracy for Class Removal. However, the lack of consensus on the best method for Class Removal on UrbanSounds8K is evident. For the transformer architectures, ST appears to be the most effective, while for VGGish, CS has the most substantial holistic impact, despite not achieving the best UA disparity.

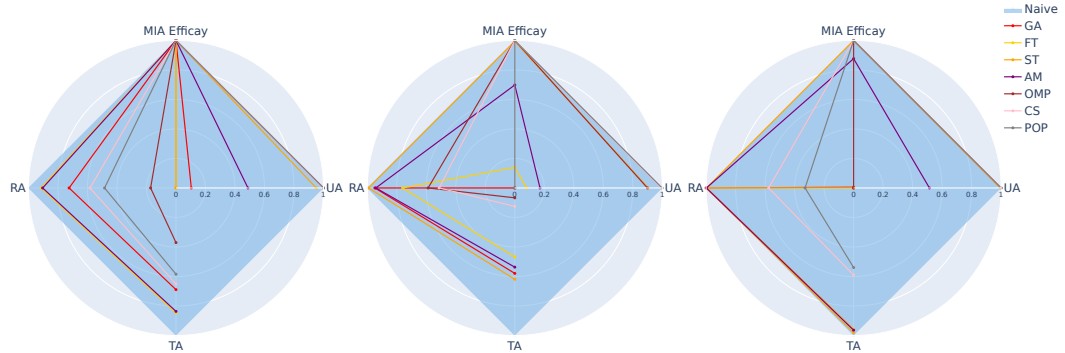

Figure 14: **1 Class Removal** radar plots on unlearning metrics based on min-max normalisation for **UrbanSounds8K**: VGGish (left), CCT (middle), and ViT (right).

## D.2 SCALING RESULTS

As unlearning requests scale for both Item and Class removal, it can be observed for both the VGGish that most methods remain stable apart from ST and GA. For the ViT architecture, all methods are stable as requests grow, and there's a good trends of most methods becoming slightly more effective as unlearning requirements increase. This growing effectiveness for the ViT architecture is a promising sign of the unlearning methods potential. The stability between the VGGish and ViT broadly speaks to the ability of most methods to have a consistent unlearning impact.

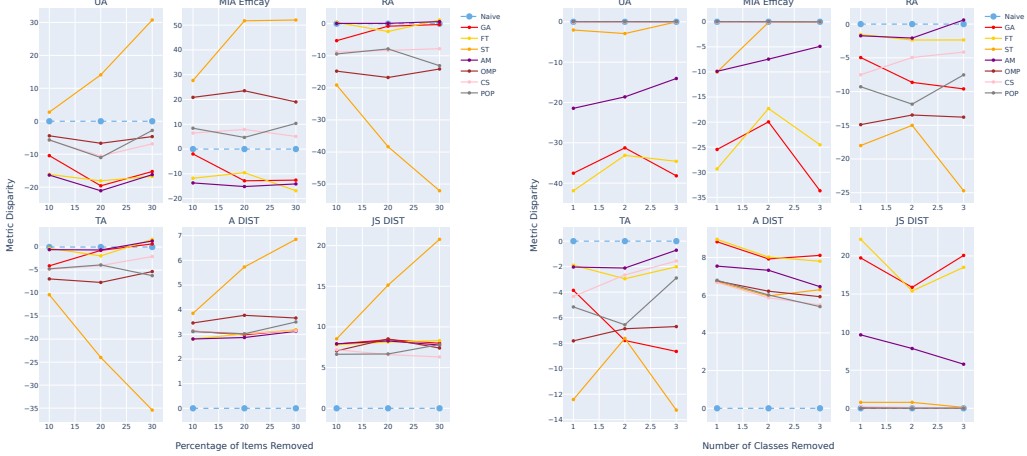

Figure 15: $\mathcal{D}_{forget}$ loss distribution on **UrbanSounds8K**, for unlearning methods averaged across all seeds for the **VGG**. 10% Item Removal (left) and 1 Class Removal (right). For each plot the unlearning method is compared to the loss distribution of $\mathcal{D}_{forget}$ on $\mathcal{M}^{\theta}$ and $\mathcal{M}^{\theta}_r$.

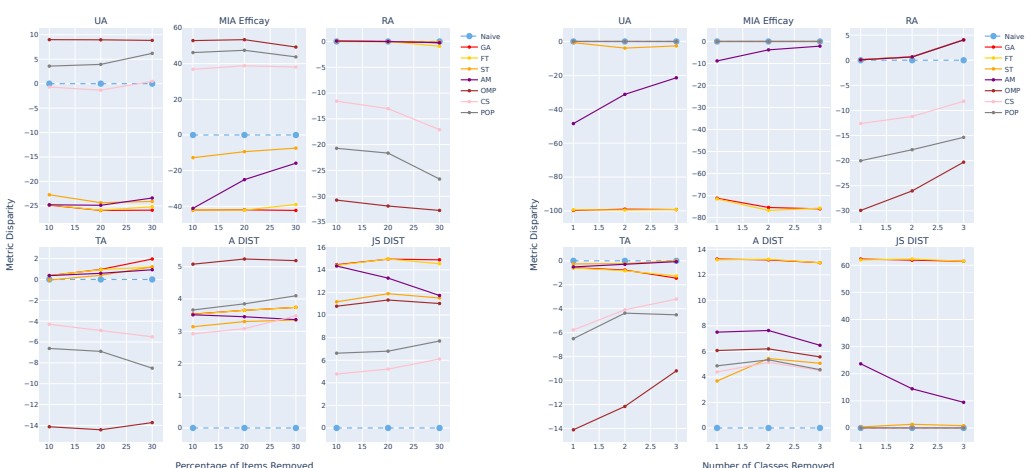

Figure 16: $\mathcal{D}_{forget}$ loss distribution on **UrbanSounds8K**, for unlearning methods averaged across all seeds for the **ViT**. 10% Item Removal (left) and 1 Class Removal (right). For each plot the unlearning method is compared to the loss distribution of $\mathcal{D}_{forget}$ on $\mathcal{M}^{\theta}$ and $\mathcal{M}_{r}^{\theta}$.

### D.3 Loss Distributions

Similarly to the results on SpeechCommands it can be observed for the VGGish and ViT for Item and Class Removal methods that approximate the distribution of the Naive model on the forget set also perform well at on UA disparity.

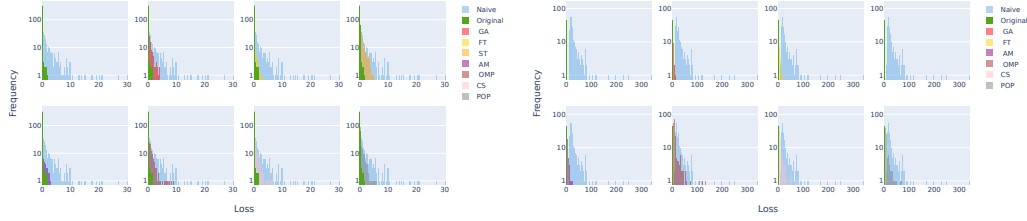

Figure 17: $\mathcal{D}_{forget}$ loss distribution on **UrbanSounds8K**, for unlearning methods averaged across all seeds for the **VGG**. 10% Item Removal (left) and 1 Class Removal (right). For each plot the unlearning method is compared to the loss distribution of $\mathcal{D}_{forget}$ on $\mathcal{M}^{\theta}$ and $\mathcal{M}_{r}^{\theta}$.

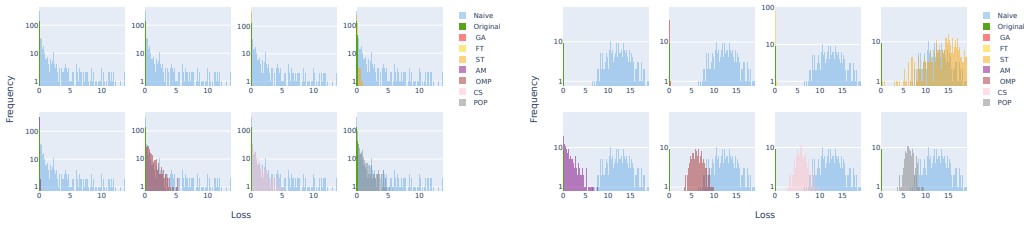

Figure 18: $\mathcal{D}_{forget}$ loss distribution on **UrbanSounds8K**, for unlearning methods averaged across all seeds for the **ViT**. 10% Item Removal (left) and 1 Class Removal (right). For each plot the unlearning method is compared to the loss distribution of $\mathcal{D}_{forget}$ on $\mathcal{M}^{\theta}$ and $\mathcal{M}_{r}^{\theta}$.

# E AUDIOMNIST RESULTS

## E.1 ITEM REMOVAL

Table 9: **10% Item Removal** results for **AudioMNIST**. Numbers in blue represent disparity from $\mathcal{M}_r^\theta$. $\mathcal{C}$ represents the objective to have the least disparity with $\mathcal{M}_r^\theta$. Otherwise arrows dictate the direction of best performance compared to $\mathcal{M}_r^\theta$.

| Model | Method | UA % ($\mathcal{C}$) | MIA Efficacy % ($\mathcal{C}$) | RA % ($\mathcal{C}$) | TA % ($\mathcal{C}$) | D AVE ($\mathcal{C}$) | A DIST (↓) (×10⁻¹) | JS DIST (↓) (×10⁻³) | RTE % (↑) |
|---|---|---|---|---|---|---|---|---|---|
| | | | | | **10% Item Removal** | | | | |
| VGGish | Naive | $1.47_{\pm 0.30}$(0.00) | $3.02_{\pm 0.24}$(0.00) | $99.74_{\pm 0.06}$(0.00) | $98.97_{\pm 0.10}$(0.00) | 0.00 | $0.00_{\pm 0.00}$ | $0.00_{\pm 0.00}$ | 0.00 |
| | GA | $0.79_{\pm 0.31}$(-0.68) | $1.93_{\pm 0.44}$(-1.09) | $99.78_{\pm 0.18}$(0.04) | $98.92_{\pm 0.15}$(-0.05) | 0.46 | $0.17_{\pm 0.03}$ | $0.35_{\pm 0.08}$ | 89.26 |
| | FT | $1.25_{\pm 1.34}$(-0.22) | $3.05_{\pm 2.89}$(0.03) | $99.31_{\pm 1.24}$(-0.43) | $98.52_{\pm 1.24}$(-0.45) | 0.28 | $0.24_{\pm 0.18}$ | $0.54_{\pm 0.49}$ | 89.62 |
| | ST | $20.08_{\pm 23.78}$(18.61) | $43.50_{\pm 21.33}$(40.48) | $80.17_{\pm 23.70}$(-19.57) | $79.73_{\pm 23.78}$(-19.24) | 24.48 | $2.95_{\pm 2.25}$ | $10.07_{\pm 12.49}$ | 84.31 |
| | AM | $4.04_{\pm 4.45}$(2.57) | $9.12_{\pm 8.11}$(6.10) | $96.77_{\pm 4.77}$(-2.97) | $96.10_{\pm 4.45}$(-2.87) | 3.63 | $0.61_{\pm 0.67}$ | $1.62_{\pm 2.34}$ | 89.47 |
| | OMP | $2.85_{\pm 0.63}$(1.38) | $10.60_{\pm 2.39}$(7.58) | $97.73_{\pm 0.58}$(-2.01) | $97.03_{\pm 0.53}$(-1.94) | 3.23 | $0.55_{\pm 0.14}$ | $1.01_{\pm 0.35}$ | 88.74 |
| | CS | $1.48_{\pm 0.58}$(0.01) | $4.16_{\pm 1.00}$(1.14) | $99.24_{\pm 0.41}$(-0.50) | $98.52_{\pm 0.34}$(-0.45) | 0.52 | $0.23_{\pm 0.05}$ | $0.41_{\pm 0.15}$ | 88.16 |
| | POP | $1.39_{\pm 0.35}$(-0.08) | $4.04_{\pm 0.52}$(1.02) | $99.35_{\pm 0.15}$(-0.39) | $98.56_{\pm 0.28}$(-0.41) | 0.48 | $0.22_{\pm 0.03}$ | $0.37_{\pm 0.06}$ | 88.27 |
| CCT | Naive | $2.82_{\pm 0.31}$(0.00) | $10.38_{\pm 0.80}$(0.00) | $99.96_{\pm 0.04}$(0.00) | $97.99_{\pm 0.11}$(0.00) | 0.00 | $0.00_{\pm 0.00}$ | $0.00_{\pm 0.00}$ | 0.00 |
| | GA | $0.10_{\pm 0.10}$(-2.72) | $3.15_{\pm 1.29}$(-7.23) | $99.96_{\pm 0.04}$(0.00) | $98.01_{\pm 0.25}$(0.02) | 2.49 | $0.39_{\pm 0.04}$ | $1.22_{\pm 0.16}$ | 88.01 |
| | FT | $0.21_{\pm 0.32}$(-2.61) | $4.47_{\pm 2.95}$(-5.91) | $99.87_{\pm 0.22}$(-0.09) | $97.90_{\pm 0.41}$(-0.09) | 2.17 | $0.39_{\pm 0.04}$ | $1.19_{\pm 0.19}$ | 88.28 |
| | ST | $1.69_{\pm 0.73}$(-1.13) | $19.34_{\pm 3.60}$(8.96) | $99.10_{\pm 0.58}$(-0.86) | $96.84_{\pm 0.51}$(-1.15) | 3.02 | $0.45_{\pm 0.08}$ | $1.02_{\pm 0.23}$ | 83.91 |
| | AM | $1.01_{\pm 0.93}$(-1.81) | $11.20_{\pm 2.45}$(0.82) | $99.52_{\pm 0.73}$(-0.44) | $97.23_{\pm 0.85}$(-0.76) | 0.96 | $0.41_{\pm 0.11}$ | $1.07_{\pm 0.34}$ | 88.00 |
| | OMP | $1.38_{\pm 0.37}$(-1.44) | $27.76_{\pm 1.06}$(17.38) | $99.28_{\pm 0.24}$(-0.68) | $96.80_{\pm 0.25}$(-1.19) | 5.17 | $0.46_{\pm 0.04}$ | $0.82_{\pm 0.09}$ | 86.76 |
| | CS | $3.82_{\pm 0.56}$(1.00) | $27.54_{\pm 1.88}$(17.16) | $97.79_{\pm 0.66}$(-2.17) | $95.61_{\pm 0.62}$(-2.38) | 5.68 | $0.63_{\pm 0.10}$ | $1.28_{\pm 0.29}$ | 87.11 |
| | POP | $4.29_{\pm 0.82}$(1.47) | $32.38_{\pm 2.80}$(22.00) | $97.71_{\pm 0.78}$(-2.25) | $95.71_{\pm 0.72}$(-2.28) | 7.00 | $0.66_{\pm 0.12}$ | $1.25_{\pm 0.38}$ | 87.13 |
| ViT | Naive | $0.62_{\pm 0.12}$(0.00) | $3.92_{\pm 0.52}$(0.00) | $99.99_{\pm 0.01}$(0.00) | $99.24_{\pm 0.06}$(0.00) | 0.00 | $0.00_{\pm 0.00}$ | $0.00_{\pm 0.00}$ | 0.00 |
| | GA | $0.00_{\pm 0.01}$(-0.62) | $1.19_{\pm 0.74}$(-2.73) | $99.99_{\pm 0.03}$(0.00) | $99.23_{\pm 0.12}$(-0.01) | 0.84 | $0.11_{\pm 0.02}$ | $0.31_{\pm 0.07}$ | 87.33 |
| | FT | $0.02_{\pm 0.03}$(-0.60) | $1.52_{\pm 1.07}$(-2.40) | $99.99_{\pm 0.01}$(0.00) | $99.25_{\pm 0.11}$(0.01) | 0.75 | $0.11_{\pm 0.01}$ | $0.31_{\pm 0.06}$ | 87.63 |
| | ST | $0.57_{\pm 0.16}$(-0.05) | $9.21_{\pm 1.19}$(5.29) | $99.76_{\pm 0.13}$(-0.23) | $98.80_{\pm 0.19}$(-0.44) | 1.50 | $0.16_{\pm 0.03}$ | $0.28_{\pm 0.07}$ | 83.03 |
| | AM | $0.30_{\pm 0.11}$(-0.32) | $5.77_{\pm 0.70}$(1.85) | $99.95_{\pm 0.03}$(-0.04) | $99.03_{\pm 0.10}$(-0.21) | 0.60 | $0.12_{\pm 0.01}$ | $0.22_{\pm 0.06}$ | 87.34 |
| | OMP | $1.44_{\pm 0.20}$(0.82) | $31.22_{\pm 2.37}$(27.30) | $98.62_{\pm 0.19}$(-1.37) | $98.13_{\pm 0.20}$(-1.11) | 7.65 | $0.43_{\pm 0.04}$ | $0.62_{\pm 0.07}$ | 87.30 |
| | CS | $1.22_{\pm 0.45}$(0.60) | $12.79_{\pm 1.38}$(8.87) | $99.25_{\pm 0.29}$(-0.74) | $98.35_{\pm 0.34}$(-0.89) | 2.78 | $0.24_{\pm 0.06}$ | $0.40_{\pm 0.17}$ | 86.31 |
| | POP | $1.73_{\pm 0.28}$(1.11) | $17.53_{\pm 1.29}$(13.61) | $98.77_{\pm 0.32}$(-1.22) | $98.03_{\pm 0.27}$(-1.21) | 4.29 | $0.33_{\pm 0.05}$ | $0.62_{\pm 0.14}$ | 86.34 |

When analysing the results for 10% Item Removal on AudioMNIST in Table 9, it is evident that for VGGish, all unlearning methods are competitive on UA. However, CS is best, with POP as the second best and ST performs well but is inconsistent. Surprisingly, all methods perform equally well on RA and TA and the distance based metrics, but there is a divergence when considering MIA Efficacy. While CS and POP are competitive for RA and TA, there is a decrease in performance, which suggests that the best unlearning methods may result in worse generalisation. The same is true when observing the results for the CCT architecture. CS, OMP and POP perform best on UA but lead to a reduction in RA and TA compared to other less effective unlearning methods. Further suggesting that unlearning methods that successfully remove the influence of $\mathcal{D}_{forget}$ from $\mathcal{M}^-$ may cause a slight reduction in generalisation capabilities. In this case for the CCT ST performs well and does not lead to a major deviation on RA and TA but due is hindered by its large divergence from the VVGish.

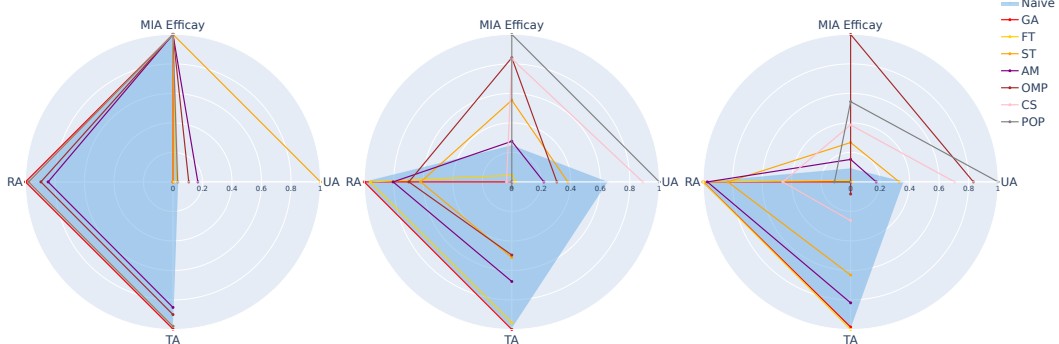

Figure 19: **10% Item Removal** radar plots on unlearning metrics based on min-max normalisation for **AudioMNIST**: VGGish (left), CCT (middle) and ViT (right).

For the ViT results, it can be observed that ST emerges as an effective unlearning method when considering UA. There is a notable divergence in MIA Efficacy for CCT and ViT when using ST, OMP, CS and POP. The increased MIA Efficacy means that the application of ST, OMP, CS and POP may trigger the Streisand Effect as they exceed the MIA Efficacy achieved by $\mathcal{M}_r^\theta$. It is worth noting that overall OMP triggers the most significant divergence for MIA Efficacy. Consequently,

when considering the unlearning methods for Item Removal for AudioMNIST, most methods appear promising. The radar plot in Figure 19 provides a more intuitive sense of this and highlights the potential Streisand Effect emerging for the CCT and ViT when using some unlearning methods.

## E.2 CLASS REMOVAL

Table 10: **1 Class Removal** results for **AudioMNIST**. Numbers in blue represent disparity from $\mathcal{M}_r^\theta$. $\mathcal{C}$ represents the objective to have the least disparity with $\mathcal{M}_r^\theta$. Otherwise arrows dictate the direction of best performance compared to $\mathcal{M}_r^\theta$.

| Model | Method | UA % ($\mathcal{C}$) | MIA Efficacy % ($\mathcal{C}$) | RA % ($\mathcal{C}$) | TA % ($\mathcal{C}$) | D AVE ($\mathcal{C}$) | A DIST ($\downarrow$) ($\times 10^{-1}$) | JS DIST ($\downarrow$) ($\times 10^{-3}$) | RTE % ($\uparrow$) |
|---|---|---|---|---|---|---|---|---|---|
| | | | | **1 Class Removal** | | | | | |
| VGGish | Naive | $100.00_{\pm 0.00}$(0.00) | $100.00_{\pm 0.00}$(0.00) | $99.73_{\pm 0.09}$(0.00) | $99.09_{\pm 0.11}$(0.00) | 0.00 | $0.00_{\pm 0.00}$ | $0.00_{\pm 0.00}$ | 0.00 |
| | GA | $25.61_{\pm 34.49}$(-74.39) | $31.81_{\pm 33.23}$(-68.19) | $99.25_{\pm 1.45}$(-0.48) | $98.49_{\pm 1.53}$(-0.60) | 35.91 | $11.22_{\pm 3.22}$ | $46.46_{\pm 21.97}$ | 88.71 |
| | FT | $12.07_{\pm 12.30}$(-87.93) | $19.46_{\pm 15.39}$(-80.54) | $\mathbf{99.76_{\pm 0.13}(0.03)}$ | $\mathbf{99.04_{\pm 0.13}(-0.05)}$ | 42.14 | $12.29_{\pm 1.47}$ | $54.87_{\pm 8.97}$ | **89.04** |
| | ST | $100.00_{\pm 0.00}$(0.00) | $100.00_{\pm 0.00}$(0.00) | $79.73_{\pm 24.26}$(-20.00) | $79.70_{\pm 24.18}$(-19.39) | 9.85 | $5.12_{\pm 1.57}$ | $0.04_{\pm 0.04}$ | 84.35 |
| | AM | $99.86_{\pm 0.19}$(-0.14) | $99.99_{\pm 0.03}$(-0.01) | $99.11_{\pm 0.50}$(-0.62) | $98.46_{\pm 0.43}$(-0.63) | 0.35 | $3.34_{\pm 0.90}$ | $0.07_{\pm 0.07}$ | 88.91 |
| | OMP | $100.00_{\pm 0.00}$(0.00) | $100.00_{\pm 0.00}$(0.00) | $97.99_{\pm 0.65}$(-1.74) | $97.23_{\pm 0.70}$(-1.86) | 0.90 | $\mathbf{2.19_{\pm 0.53}}$ | $\mathbf{0.01_{\pm 0.00}}$ | 88.18 |
| | CS | $91.84_{\pm 5.23}$(-8.16) | $97.30_{\pm 2.14}$(-2.70) | $99.53_{\pm 0.17}$(-0.20) | $98.80_{\pm 0.23}$(-0.29) | 2.84 | $3.04_{\pm 0.47}$ | $3.71_{\pm 2.50}$ | 87.87 |
| | POP | $99.67_{\pm 0.51}$(-0.33) | $99.96_{\pm 0.09}$(-0.04) | $99.23_{\pm 0.29}$(-0.50) | $98.58_{\pm 0.29}$(-0.51) | **0.34** | $3.04_{\pm 0.87}$ | $0.14_{\pm 0.20}$ | 88.00 |
| CCT | Naive | $100.00_{\pm 0.00}$(0.00) | $100.00_{\pm 0.00}$(0.00) | $99.96_{\pm 0.03}$(0.00) | $98.20_{\pm 0.16}$(0.00) | 0.00 | $0.00_{\pm 0.00}$ | $0.00_{\pm 0.00}$ | 0.00 |
| | GA | $0.73_{\pm 0.84}$(-99.27) | $13.78_{\pm 2.86}$(-86.22) | $\mathbf{99.82_{\pm 0.37}(-0.14)}$ | $\mathbf{97.86_{\pm 0.54}(-0.34)}$ | 46.49 | $13.46_{\pm 0.13}$ | $63.10_{\pm 0.84}$ | 88.73 |
| | FT | $0.52_{\pm 0.98}$(-99.48) | $12.03_{\pm 3.46}$(-87.97) | $99.77_{\pm 0.44}$(-0.19) | $97.84_{\pm 0.46}$(-0.36) | 47.0 | $13.51_{\pm 0.16}$ | $63.50_{\pm 1.08}$ | **89.00** |
| | ST | $100.00_{\pm 0.00}$(0.00) | $100.00_{\pm 0.00}$(0.00) | $99.47_{\pm 0.30}$(-0.49) | $97.31_{\pm 0.31}$(-0.89) | **0.34** | $\mathbf{3.05_{\pm 0.64}}$ | $\mathbf{0.01_{\pm 0.00}}$ | 84.76 |
| | AM | $99.65_{\pm 0.87}$(-0.35) | $99.37_{\pm 1.22}$(-0.59) | $99.22_{\pm 0.29}$(-0.59) | $97.42_{\pm 1.25}$(-0.78) | 0.43 | $3.92_{\pm 0.66}$ | $0.16_{\pm 0.36}$ | 88.76 |
| | OMP | $37.63_{\pm 4.95}$(-62.37) | $88.96_{\pm 3.60}$(-11.04) | $99.22_{\pm 0.29}$(-0.74) | $96.91_{\pm 0.39}$(-1.29) | 18.86 | $8.75_{\pm 0.62}$ | $31.27_{\pm 3.34}$ | 88.00 |
| | CS | $85.34_{\pm 6.25}$(-14.66) | $99.92_{\pm 0.14}$(-0.08) | $98.05_{\pm 0.79}$(-1.91) | $96.08_{\pm 0.65}$(-2.12) | 4.69 | $3.75_{\pm 0.76}$ | $5.75_{\pm 2.68}$ | 87.87 |
| | POP | $96.73_{\pm 2.42}$(-3.27) | $100.00_{\pm 0.00}$(0.00) | $97.19_{\pm 0.74}$(-2.77) | $95.48_{\pm 0.75}$(-2.72) | 2.19 | $3.29_{\pm 0.68}$ | $1.19_{\pm 0.82}$ | 87.88 |
| ViT | Naive | $100.00_{\pm 0.00}$(0.00) | $100.00_{\pm 0.00}$(0.00) | $99.99_{\pm 0.00}$(0.00) | $99.34_{\pm 0.07}$(0.00) | 0.00 | $0.00_{\pm 0.00}$ | $0.00_{\pm 0.00}$ | 0.00 |
| | GA | $0.44_{\pm 0.66}$(-99.56) | $7.74_{\pm 4.24}$(-92.26) | $99.95_{\pm 0.07}$(-0.04) | $99.26_{\pm 0.18}$(-0.08) | 47.98 | $13.74_{\pm 0.13}$ | $64.26_{\pm 0.96}$ | 89.07 |
| | FT | $0.80_{\pm 1.04}$(-99.20) | $8.75_{\pm 4.45}$(-91.25) | $\mathbf{99.99_{\pm 0.01}(0.00)}$ | $\mathbf{99.28_{\pm 0.15}(-0.06)}$ | 47.63 | $13.69_{\pm 0.16}$ | $63.93_{\pm 1.13}$ | **89.32** |
| | ST | $100.00_{\pm 0.00}$(0.00) | $100.00_{\pm 0.00}$(0.00) | $99.60_{\pm 0.26}$(-0.39) | $98.80_{\pm 0.23}$(-0.54) | 0.23 | $3.04_{\pm 0.83}$ | $0.02_{\pm 0.00}$ | 85.57 |
| | AM | $100.00_{\pm 0.00}$(0.00) | $100.00_{\pm 0.00}$(0.00) | $99.98_{\pm 0.02}$(-0.01) | $99.23_{\pm 0.14}$(-0.11) | **0.03** | $6.48_{\pm 1.26}$ | $0.02_{\pm 0.00}$ | 89.09 |
| | OMP | $99.83_{\pm 0.28}$(-0.17) | $100.00_{\pm 0.00}$(0.00) | $98.82_{\pm 0.15}$(-1.17) | $98.38_{\pm 0.18}$(-0.96) | 0.57 | $2.48_{\pm 0.51}$ | $0.15_{\pm 0.14}$ | 89.17 |
| | CS | $98.63_{\pm 1.81}$(-1.37) | $100.00_{\pm 0.00}$(0.00) | $99.48_{\pm 0.19}$(-0.51) | $98.80_{\pm 0.17}$(-0.54) | 0.60 | $2.83_{\pm 1.33}$ | $0.50_{\pm 0.57}$ | 88.33 |
| | POP | $100.00_{\pm 0.00}$(0.00) | $100.00_{\pm 0.00}$(0.00) | $98.89_{\pm 0.35}$(-1.10) | $98.34_{\pm 0.32}$(-1.00) | 0.52 | $\mathbf{2.41_{\pm 0.96}}$ | $\mathbf{0.01_{\pm 0.01}}$ | 88.34 |

Conversely, when considering Class Removal requests on AudioMNIST in Table 10, there is a much clearer perspective on the most efficacious methods. For the VGGish, The best method is found when using OMP and ST. AM and POP are competitive on UA and MIA and result in small accuracy fluctuations for RA and TA, making them more effective than OMP and ST. GA and FT become ineffective on the VGGish when considering the Class Removal request as they are incapable of removing $\mathcal{D}_{forget}$ from $\mathcal{M}^-$; this remains the case across all architectures. The inability to remove $\mathcal{D}_{forget}$ in UA highlights their lack of suitability for harsher unlearning requests that demand increased weight perturbation. For the transformer architectures, the best methods in order are ST, AM and POP for the CCT and AM, ST and POP for the ViT across accuracy and distance metrics as highlighted in Figure 20. However, it is essential to note that ST has the highest computational cost (lowest RTE) for unlearning on all architectures. Additionally, AM, under its application, could negatively impact decision boundaries and downstream tasks.

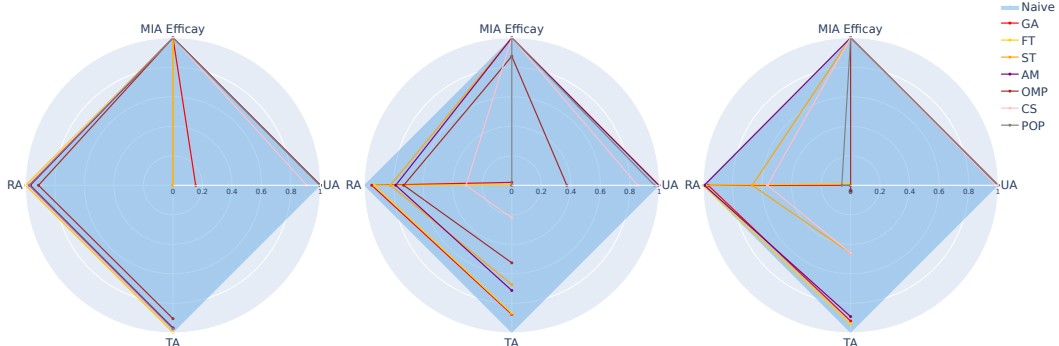

Figure 20: **1 Class Removal** radar plots on unlearning metrics based on min-max normalisation for **AudioMNIST**: VGGish (left), CCT (middle) and ViT (right).

### E.3 MACHINE UNLEARNING REQUEST SCALING

Due to the close performance of various unlearning methods in Item Removal across different architectures, it is crucial to investigate the scaling laws of these methods. The objective is to identify any fluctuations that occur as the size of removal requests increases for both Item and Class Removal. An effective unlearning method should maintain consistent performance as the scale of unlearning requests grows, thereby ensuring the protection of privacy.

Figures 21, 22, and 23 present the scaling relationships for VGGish, CCT, and ViT, respectively. In the context of Item Removal, most unlearning methods demonstrate reasonable scalability. However, across all examined architectures, the ST method performs inadequately and deteriorates compared to the baseline across nearly all metrics. Conversely, the methods POP, CS, and OMP exhibit the best performance, as they remain close to the baseline in terms of Unlearning Accuracy (UA), while maintaining stable impacts on the other metrics as the number of Item Removal requests increases.

In the scenario of Class Removal, the stability of the various unlearning methods is evident across the board. Notably, the OMP, CS, and POP pruning methods display similar scaling trends, highlighting the overall reliability of pruning strategies in unlearning and the subtle nuances among each approach. When considering the scaling of Class Removal requests for the CCT and ViT, method AM emerges as the most effective unlearning strategy in this context.

An unlearning method designed for the audio domain should ideally possess qualities of universality and demonstrate consistent performance as the complexity of tasks increases. Any methodology that fails to achieve this would undermine the universal requirement of an effective unlearning technique. As shown in the results for SpeechCommands and UrbanSounds8K presented in the main body of the study, there is a slight variation in the efficacy of the unlearning methods when task complexity is heightened.

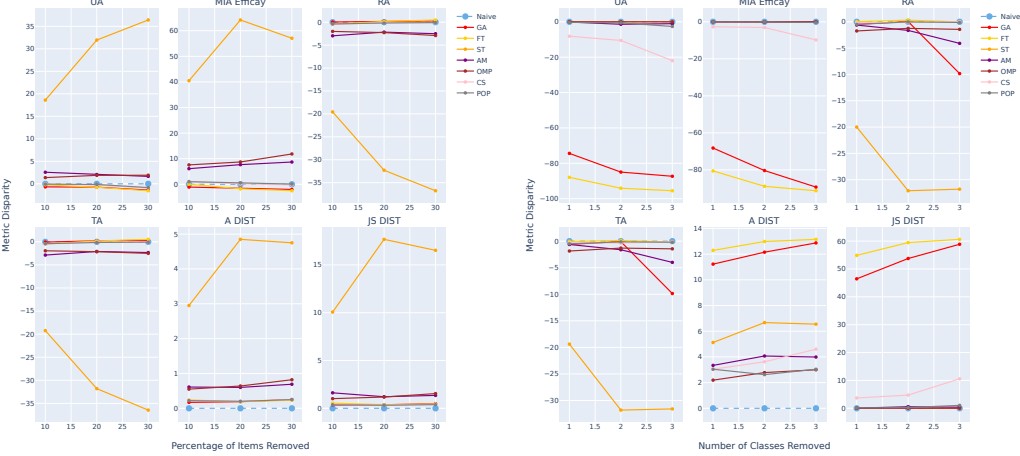

Figure 21: $\mathcal{D}_{forget}$ loss distribution on **AudioMNIST**, for unlearning methods averaged across all seeds for the **VGGish**. 10% Item Removal (left) and 1 Class Removal (right). For each plot the unlearning method is compared to the loss distribution of $\mathcal{D}_{forget}$ on $\mathcal{M}^{\theta}$ and $\mathcal{M}^{\theta}_{r}$.

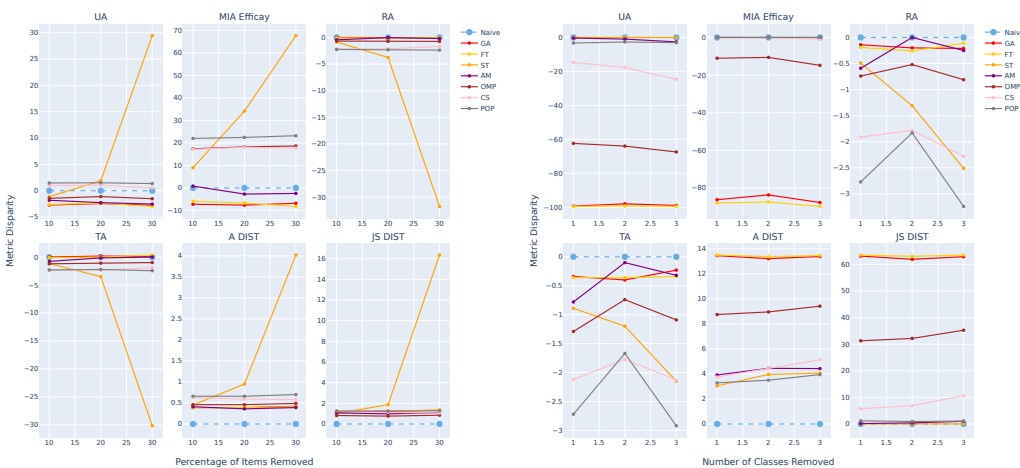

Figure 22: $\mathcal{D}_{forget}$ loss distribution on **AudioMNIST**, for unlearning methods averaged across all seeds for the **CCT**. 10% Item Removal (left) and 1 Class Removal (right). For each plot the unlearning method is compared to the loss distribution of $\mathcal{D}_{forget}$ on $\mathcal{M}^{\theta}$ and $\mathcal{M}^{\theta}_{r}$.

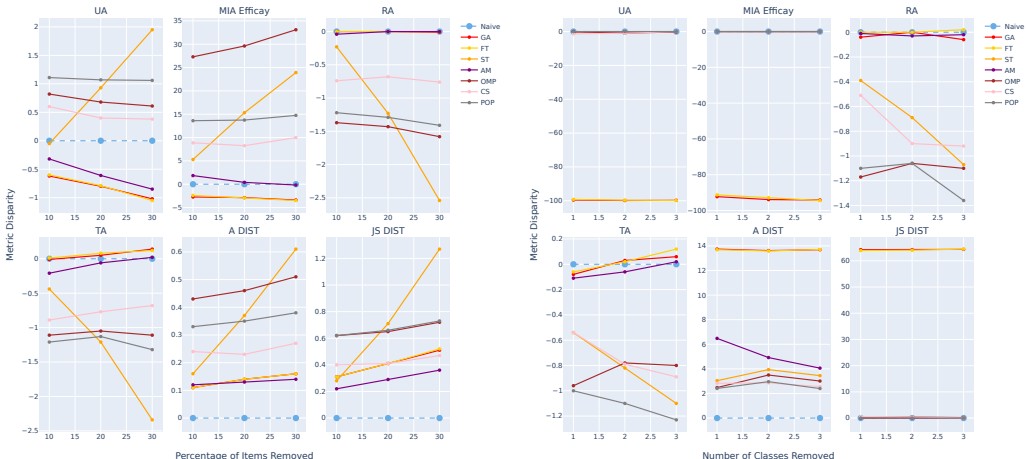

Figure 23: $\mathcal{D}_{forget}$ loss distribution on **AudioMNIST**, for unlearning methods averaged across all seeds for the **ViT**. 10% Item Removal (left) and 1 Class Removal (right). For each plot the unlearning method is compared to the loss distribution of $\mathcal{D}_{forget}$ on $\mathcal{M}^{\theta}$ and $\mathcal{M}^{\theta}_{r}$.

### E.4 LOSS DISTRIBUTION

For the loss distributions, we can see that for Item Removal, most methods can force the distribution for the forget set into a distribution of the Naive Retraining for the VGGish; however, for the ST method, it is clear that it has a higher density of increased loss values which exceeds that of the Naive models. However, when we consider the transformer architectures the best methods in order are POP and CS as they best match the loss distribution created by the Naive model consistently. However, when we consider class removal, it is evident that the best methods for matching the loss distribution of the Naive models in order are AM, ST and POP, and they manage to separate the loss sufficiently from the baseline. In conclusion, the loss distributions largely match the results witnessed for UA divergence, providing a strong indication that the loss perspective is a reliable proxy for identifying efficacious unlearning methods.

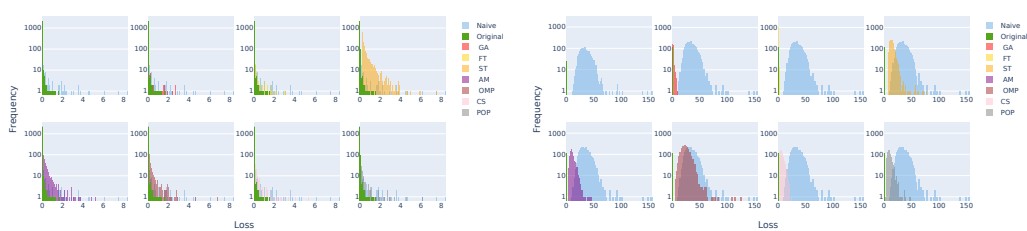

Figure 24: $\mathcal{D}_{forget}$ loss distribution on **AudioMNIST**, for unlearning methods averaged across all seeds for the **VGGish**. 10% Item Removal (left) and 1 Class Removal (right). For each plot the unlearning method is compared to the loss distribution of $\mathcal{D}_{forget}$ on $\mathcal{M}^\theta$ and $\mathcal{M}_r^\theta$.

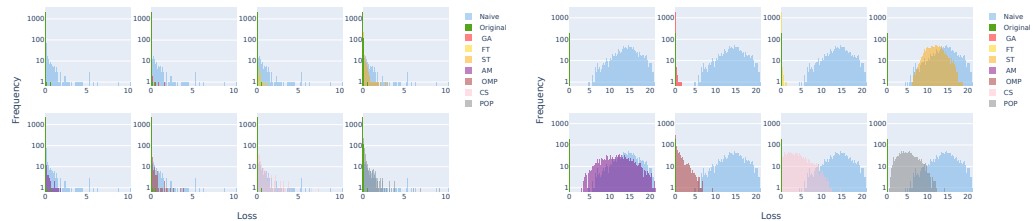

Figure 25: $\mathcal{D}_{forget}$ loss distribution on **AudioMNIST**, for unlearning methods averaged across all seeds for the **CCT**. 10% Item Removal (left) and 1 Class Removal (right). For each plot the unlearning method is compared to the loss distribution of $\mathcal{D}_{forget}$ on $\mathcal{M}^\theta$ and $\mathcal{M}_r^\theta$.

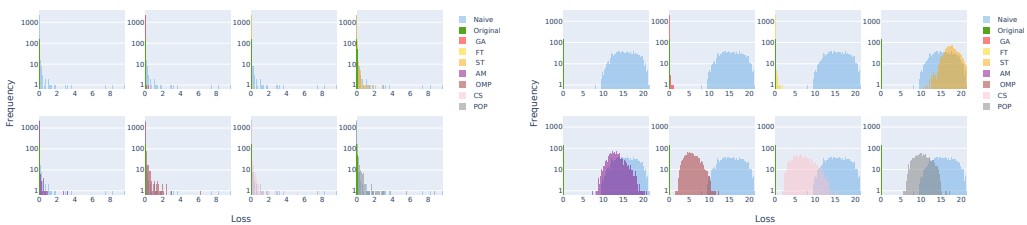

Figure 26: $\mathcal{D}_{forget}$ loss distribution on **AudioMNIST**, for unlearning methods averaged across all seeds for the **ViT**. 10% Item Removal (left) and 1 Class Removal (right). For each plot the unlearning method is compared to the loss distribution of $\mathcal{D}_{forget}$ on $\mathcal{M}^\theta$ and $\mathcal{M}_r^\theta$.

# F CIFAR10 RESULTS

We present the results for networks trained on CIFAR10 to show the method's viability across domains. To match the experimental setup in the paper's main body, we use the same optimizer and loss, with the only difference being the use of 80 epochs for training, 1 impair step for unlearning and 8 repair steps for retraining. Additionally the architectures have been modified to take in the correct input and have increased their capacity to improve performance on the dataset. Overall, from the results presented in Table 11, it can be noted that the dynamic sparsity unlearning methods vastly outperform all other unlearning methods for Item Removal across architectures. When considering Class Removal, Table 12, this gap between the methods is less pronounced, but both the Prune and Regrow methods perform well, with POP performing the best.

Table 11: **10% Item Removal** results for **CIFAR10**. Numbers in blue represent disparity from $\mathcal{M}_r^\theta$. $\mathcal{C}$ represents the objective to have the least disparity with $\mathcal{M}_r^\theta$. Otherwise arrows dictate the direction of best performance compared to $\mathcal{M}_r^\theta$.

| Model | Method | UA % ($\mathcal{C}$) | MIA Efficacy % ($\mathcal{C}$) | RA % ($\mathcal{C}$) | TA % ($\mathcal{C}$) | D AVE ($\mathcal{C}$) | A DIST ($\downarrow$) ($\times 10^{-1}$) | JS DIST ($\downarrow$) ($\times 10^{-3}$) | RTE % ($\uparrow$) |
|---|---|---|---|---|---|---|---|---|---|
| | | | | 10% Item Removal | | | | | |
| VGG16 | Naive | $14.12_{\pm 0.28}$(0.00) | $40.51_{\pm 1.08}$(0.00) | $100.00_{\pm 0.00}$(0.00) | $85.49_{\pm 0.29}$(0.00) | 0.00 | $0.00_{\pm 0.00}$ | $0.00_{\pm 0.00}$ | 0.00 |
| | GA | $0.00_{\pm 0.00}$(-14.12) | $2.14_{\pm 0.67}$(-38.37) | $100.00_{\pm 0.00}$(0.00) | $86.05_{\pm 0.30}$(0.56) | 13.26 | $2.00_{\pm 0.03}$ | $8.25_{\pm 0.15}$ | 87.36 |
| | FT | $0.00_{\pm 0.00}$(-14.12) | $2.07_{\pm 0.69}$(-38.44) | $100.00_{\pm 0.00}$(0.00) | $86.02_{\pm 0.31}$(0.53) | 13.27 | $2.00_{\pm 0.03}$ | $8.25_{\pm 0.15}$ | **87.55** |
| | ST | $1.04_{\pm 0.14}$(-13.08) | $47.16_{\pm 2.27}$(6.65) | $100.00_{\pm 0.00}$(0.00) | $\mathbf{85.61_{\pm 0.28}}$(0.12) | 4.96 | $2.01_{\pm 0.03}$ | $7.81_{\pm 0.16}$ | 82.77 |
| | AM | $0.00_{\pm 0.01}$(-14.12) | $27.27_{\pm 1.41}$(-13.24) | $100.00_{\pm 0.00}$(0.00) | $85.81_{\pm 0.27}$(0.32) | 6.92 | $2.00_{\pm 0.03}$ | $8.21_{\pm 0.15}$ | 87.40 |
| | OMP | $4.35_{\pm 0.50}$(-9.77) | $41.52_{\pm 0.90}$(1.01) | $100.00_{\pm 0.00}$(0.00) | $84.72_{\pm 0.19}$(-0.77) | **2.89** | $\mathbf{1.85_{\pm 0.04}}$ | $6.10_{\pm 0.20}$ | 87.15 |
| | CS | $13.29_{\pm 1.60}$(-0.83) | $56.38_{\pm 2.29}$(15.87) | $97.85_{\pm 1.04}$(-2.15) | $81.50_{\pm 1.25}$(-3.99) | 5.71 | $2.16_{\pm 0.16}$ | $\mathbf{5.94_{\pm 0.53}}$ | 86.36 |
| | POP | $17.73_{\pm 1.48}$(3.61) | $64.49_{\pm 2.18}$(23.98) | $96.47_{\pm 1.49}$(-3.53) | $80.46_{\pm 1.33}$(-5.03) | 9.04 | $2.41_{\pm 0.22}$ | $6.50_{\pm 0.78}$ | 86.36 |
| CCT | Naive | $27.02_{\pm 0.47}$(0.00) | $52.24_{\pm 1.99}$(0.00) | $100.00_{\pm 0.00}$(0.00) | $72.85_{\pm 0.46}$(0.00) | 0.00 | $0.00_{\pm 0.00}$ | $0.00_{\pm 0.00}$ | 0.00 |
| | GA | $0.00_{\pm 0.00}$(-27.02) | $2.01_{\pm 1.70}$(-50.23) | $100.00_{\pm 0.00}$(0.00) | $73.34_{\pm 0.29}$(0.49) | 19.43 | $3.80_{\pm 0.05}$ | $15.97_{\pm 0.28}$ | 81.79 |
| | FT | $0.00_{\pm 0.00}$(-27.02) | $1.93_{\pm 1.57}$(-50.31) | $100.00_{\pm 0.00}$(0.00) | $73.35_{\pm 0.29}$(0.50) | 19.46 | $3.80_{\pm 0.05}$ | $15.97_{\pm 0.28}$ | **82.06** |
| | ST | $7.10_{\pm 1.67}$(-19.92) | $\mathbf{47.80_{\pm 4.92}}$(-4.44) | $98.79_{\pm 0.93}$(-1.21) | $70.80_{\pm 0.64}$(-2.05) | **6.91** | $3.57_{\pm 0.05}$ | $12.17_{\pm 0.54}$ | 75.79 |
| | AM | $0.12_{\pm 0.19}$(-26.90) | $19.53_{\pm 3.07}$(-32.71) | $100.00_{\pm 0.00}$(0.00) | $73.27_{\pm 0.24}$(0.42) | 15.01 | $3.77_{\pm 0.06}$ | $15.64_{\pm 0.40}$ | 81.83 |
| | OMP | $8.09_{\pm 0.75}$(-18.93) | $71.67_{\pm 1.04}$(19.43) | $\mathbf{99.63_{\pm 0.18}}$(-0.37) | $70.89_{\pm 0.40}$(-1.96) | 10.17 | $3.41_{\pm 0.05}$ | $10.18_{\pm 0.26}$ | 80.07 |
| | CS | $17.65_{\pm 2.45}$(-9.37) | $67.87_{\pm 4.14}$(15.63) | $95.97_{\pm 2.05}$(-4.03) | $69.47_{\pm 0.74}$(-3.38) | 8.10 | $\mathbf{3.29_{\pm 0.16}}$ | $8.55_{\pm 0.34}$ | 80.27 |
| | POP | $\mathbf{22.27_{\pm 1.09}}$(-4.75) | $79.40_{\pm 1.69}$(27.16) | $93.82_{\pm 1.57}$(-6.18) | $69.12_{\pm 1.03}$(-3.73) | 10.46 | $3.30_{\pm 0.13}$ | $\mathbf{7.34_{\pm 0.37}}$ | 80.23 |
| ViT | Naive | $31.11_{\pm 0.87}$(0.00) | $56.74_{\pm 1.68}$(0.00) | $100.00_{\pm 0.00}$(0.00) | $68.28_{\pm 0.58}$(0.00) | 0.00 | $0.00_{\pm 0.00}$ | $0.00_{\pm 0.00}$ | 0.00 |
| | GA | $0.00_{\pm 0.00}$(-31.11) | $2.15_{\pm 2.43}$(-54.59) | $100.00_{\pm 0.00}$(0.00) | $69.00_{\pm 0.36}$(0.72) | 21.60 | $4.37_{\pm 0.11}$ | $18.55_{\pm 0.51}$ | 86.21 |
| | FT | $0.00_{\pm 0.00}$(-31.11) | $2.09_{\pm 2.38}$(-54.65) | $100.00_{\pm 0.00}$(0.00) | $68.98_{\pm 0.34}$(0.70) | 21.62 | $4.37_{\pm 0.11}$ | $18.55_{\pm 0.51}$ | **86.40** |
| | ST | $1.88_{\pm 0.23}$(-29.23) | $\mathbf{44.11_{\pm 1.70}}$(-12.63) | $100.00_{\pm 0.00}$(0.00) | $\mathbf{68.59_{\pm 0.38}}$(0.31) | 10.54 | $4.27_{\pm 0.11}$ | $16.85_{\pm 0.54}$ | 81.69 |
| | AM | $0.15_{\pm 0.10}$(-30.96) | $29.66_{\pm 2.22}$(-27.08) | $100.00_{\pm 0.00}$(0.00) | $68.63_{\pm 0.34}$(0.35) | 14.60 | $4.33_{\pm 0.11}$ | $18.02_{\pm 0.52}$ | 86.22 |
| | OMP | $32.66_{\pm 1.30}$(1.55) | $99.52_{\pm 0.31}$(42.78) | $70.35_{\pm 1.43}$(-29.65) | $63.38_{\pm 0.83}$(-4.90) | 19.72 | $4.51_{\pm 0.15}$ | $8.35_{\pm 0.44}$ | 86.35 |
| | CS | $24.59_{\pm 1.15}$(-6.52) | $82.10_{\pm 1.85}$(25.36) | $92.94_{\pm 1.22}$(-7.06) | $65.37_{\pm 0.76}$(-2.91) | **10.46** | $\mathbf{3.72_{\pm 0.17}}$ | $8.49_{\pm 0.60}$ | 85.60 |
| | POP | $\mathbf{30.19_{\pm 1.07}}$(-0.92) | $95.60_{\pm 1.31}$(38.86) | $82.97_{\pm 1.35}$(-17.03) | $65.74_{\pm 0.74}$(-2.54) | 14.84 | $3.91_{\pm 0.09}$ | $\mathbf{7.34_{\pm 0.35}}$ | 85.60 |

Table 12: **1 Class Removal** results for **CIFAR10**. Numbers in blue represent disparity from $\mathcal{M}_r^\theta$. $\mathcal{C}$ represents the objective to have the least disparity with $\mathcal{M}_r^\theta$. Otherwise arrows dictate the direction of best performance compared to $\mathcal{M}_r^\theta$.

| Model | Method | UA % ($\mathcal{C}$) | MIA Efficacy % ($\mathcal{C}$) | RA % ($\mathcal{C}$) | TA % ($\mathcal{C}$) | D AVE ($\mathcal{C}$) | A DIST ($\downarrow$) ($\times 10^{-1}$) | JS DIST ($\downarrow$) ($\times 10^{-3}$) | RTE % ($\uparrow$) |
|---|---|---|---|---|---|---|---|---|---|
| | | | | 1 Class Removal | | | | | |
| VGG16 | Naive | $100.00_{\pm 0.00}$(0.00) | $100.00_{\pm 0.00}$(0.00) | $100.00_{\pm 0.00}$(0.00) | $85.56_{\pm 0.24}$(0.00) | 0.00 | $0.00_{\pm 0.00}$ | $0.00_{\pm 0.00}$ | 0.00 |
| | GA | $100.00_{\pm 0.00}$(0.00) | $100.00_{\pm 0.00}$(0.00) | $97.57_{\pm 2.19}$(-2.43) | $81.30_{\pm 1.19}$(-4.26) | 1.67 | $3.79_{\pm 0.60}$ | $0.01_{\pm 0.00}$ | 87.98 |
| | FT | $0.11_{\pm 0.28}$(-99.89) | $7.67_{\pm 10.75}$(-92.33) | $100.00_{\pm 0.00}$(0.00) | $85.38_{\pm 0.32}$(-0.18) | 48.1 | $13.82_{\pm 0.06}$ | $65.03_{\pm 0.42}$ | **88.16** |
| | ST | $100.00_{\pm 0.00}$(0.00) | $100.00_{\pm 0.00}$(0.00) | $100.00_{\pm 0.00}$(0.00) | $85.62_{\pm 0.30}$(0.06) | 0.02 | $3.03_{\pm 0.19}$ | $0.02_{\pm 0.00}$ | 83.64 |
| | AM | $100.00_{\pm 0.00}$(0.00) | $100.00_{\pm 0.00}$(0.00) | $100.00_{\pm 0.00}$(0.00) | $\mathbf{85.58_{\pm 0.31}}$(0.02) | **0.00** | $2.97_{\pm 0.19}$ | $0.02_{\pm 0.00}$ | 88.03 |
| | OMP | $100.00_{\pm 0.00}$(0.00) | $100.00_{\pm 0.00}$(0.00) | $100.00_{\pm 0.00}$(0.00) | $84.52_{\pm 0.28}$(-1.04) | 0.26 | $\mathbf{2.93_{\pm 0.23}}$ | $0.01_{\pm 0.00}$ | 87.50 |
| | CS | $100.00_{\pm 0.00}$(0.00) | $100.00_{\pm 0.00}$(0.00) | $98.34_{\pm 0.88}$(-1.66) | $82.11_{\pm 1.05}$(-3.45) | 1.28 | $2.98_{\pm 0.59}$ | $0.01_{\pm 0.00}$ | 87.36 |
| | POP | $100.00_{\pm 0.00}$(0.00) | $100.00_{\pm 0.00}$(0.00) | $95.90_{\pm 1.11}$(-4.10) | $80.12_{\pm 1.21}$(-5.44) | 2.38 | $3.54_{\pm 0.74}$ | $0.01_{\pm 0.00}$ | 87.38 |
| CCT | Naive | $100.00_{\pm 0.00}$(0.00) | $100.00_{\pm 0.00}$(0.00) | $100.00_{\pm 0.00}$(0.00) | $73.54_{\pm 0.32}$(0.00) | 0.00 | $0.00_{\pm 0.00}$ | $0.00_{\pm 0.00}$ | 0.00 |
| | GA | $79.75_{\pm 39.48}$(-20.25) | $85.09_{\pm 30.14}$(-14.91) | $81.24_{\pm 35.08}$(-18.76) | $59.50_{\pm 24.23}$(-14.04) | 16.99 | $6.79_{\pm 3.94}$ | $12.93_{\pm 25.30}$ | 85.65 |
| | FT | $0.00_{\pm 0.00}$(-100.00) | $17.96_{\pm 6.10}$(-82.04) | $100.00_{\pm 0.00}$(0.00) | $\mathbf{72.80_{\pm 0.25}}$(-0.74) | 45.7 | $13.76_{\pm 0.04}$ | $64.88_{\pm 0.20}$ | **85.89** |
| | ST | $100.00_{\pm 0.00}$(0.00) | $100.00_{\pm 0.00}$(0.00) | $93.22_{\pm 5.46}$(-6.78) | $69.48_{\pm 0.78}$(-4.06) | **2.71** | $4.38_{\pm 0.64}$ | $\mathbf{0.02_{\pm 0.00}}$ | 81.15 |
| | AM | $94.82_{\pm 6.18}$(-5.18) | $99.75_{\pm 0.45}$(-0.25) | $99.50_{\pm 1.00}$(-0.50) | $72.47_{\pm 1.06}$(-1.07) | 1.75 | $4.49_{\pm 0.40}$ | $2.43_{\pm 3.03}$ | 85.67 |
| | OMP | $77.99_{\pm 1.92}$(-22.01) | $99.92_{\pm 0.06}$(-0.08) | $99.64_{\pm 0.17}$(-0.36) | $71.18_{\pm 0.32}$(-2.36) | 6.20 | $5.72_{\pm 0.20}$ | $9.37_{\pm 0.96}$ | 84.07 |
| | CS | $97.85_{\pm 1.72}$(-2.15) | $100.00_{\pm 0.00}$(0.00) | $94.72_{\pm 2.98}$(-5.28) | $68.92_{\pm 1.29}$(-4.62) | 3.01 | $4.36_{\pm 0.49}$ | $0.85_{\pm 0.64}$ | 85.08 |
| | POP | $99.63_{\pm 0.40}$(-0.37) | $100.00_{\pm 0.00}$(0.00) | $92.60_{\pm 2.65}$(-7.40) | $69.44_{\pm 1.76}$(-4.10) | 2.97 | $\mathbf{4.07_{\pm 0.59}}$ | $0.16_{\pm 0.14}$ | 85.04 |
| ViT | Naive | $100.00_{\pm 0.00}$(0.00) | $100.00_{\pm 0.00}$(0.00) | $100.00_{\pm 0.00}$(0.00) | $68.55_{\pm 0.54}$(0.00) | 0.00 | $0.00_{\pm 0.00}$ | $0.00_{\pm 0.00}$ | 0.00 |
| | GA | $10.02_{\pm 29.99}$(-89.98) | $24.46_{\pm 26.98}$(-75.54) | $95.14_{\pm 14.59}$(-4.86) | $66.31_{\pm 5.61}$(-2.24) | 43.16 | $12.97_{\pm 2.26}$ | $58.29_{\pm 19.41}$ | 84.84 |
| | FT | $0.00_{\pm 0.00}$(-100.00) | $15.15_{\pm 8.29}$(-84.85) | $100.00_{\pm 0.00}$(0.00) | $\mathbf{68.23_{\pm 0.54}}$(-0.32) | 46.29 | $13.73_{\pm 0.03}$ | $64.86_{\pm 0.18}$ | **85.05** |
| | ST | $100.00_{\pm 0.00}$(0.00) | $100.00_{\pm 0.00}$(0.00) | $97.07_{\pm 8.51}$(-2.93) | $67.92_{\pm 1.58}$(-0.63) | 0.89 | $4.85_{\pm 0.12}$ | $\mathbf{0.02_{\pm 0.01}}$ | 79.73 |
| | AM | $98.96_{\pm 1.86}$(-1.04) | $99.96_{\pm 0.08}$(-0.04) | $100.00_{\pm 0.00}$(0.00) | $69.02_{\pm 0.56}$(0.47) | **0.39** | $5.12_{\pm 0.18}$ | $0.47_{\pm 0.82}$ | 84.84 |
| | OMP | $99.98_{\pm 0.03}$(-0.02) | $100.00_{\pm 0.00}$(0.00) | $70.68_{\pm 1.33}$(-29.32) | $63.87_{\pm 0.41}$(-4.68) | 8.50 | $4.73_{\pm 0.20}$ | $0.07_{\pm 0.01}$ | 85.02 |
| | CS | $99.82_{\pm 0.27}$(-0.18) | $100.00_{\pm 0.00}$(0.00) | $93.87_{\pm 1.15}$(-6.13) | $66.17_{\pm 0.66}$(-2.38) | 2.17 | $4.72_{\pm 0.45}$ | $0.10_{\pm 0.09}$ | 84.21 |
| | POP | $100.00_{\pm 0.00}$(0.00) | $100.00_{\pm 0.00}$(0.00) | $82.85_{\pm 1.81}$(-17.15) | $66.63_{\pm 0.76}$(-1.92) | 4.77 | $\mathbf{4.37_{\pm 0.25}}$ | $0.03_{\pm 0.01}$ | 84.21 |

## F.1 RADAR PLOTS

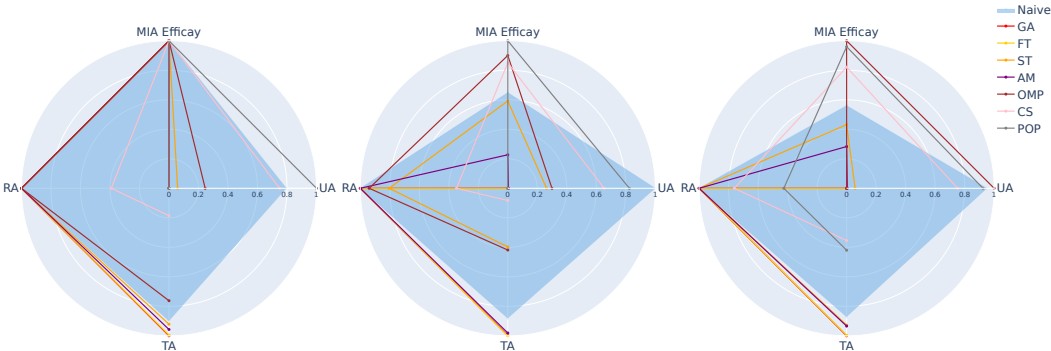

Figure 27: **10% Item Removal** radar plots on unlearning metrics based on min-max normalisation for **CIFAR 10**: VGG 16 (left), CCT (middle), and ViT (right).

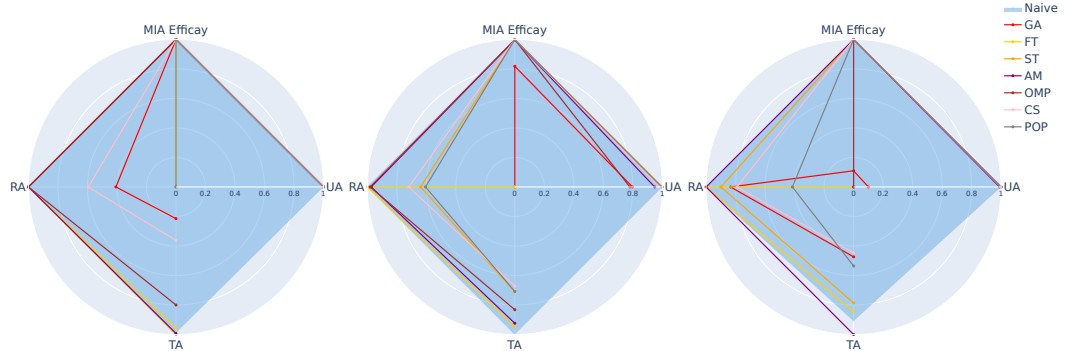

Figure 28: **1 Class Removal** radar plots on unlearning metrics based on min-max normalisation for **CIFAR 10**: VGG 16 (left), CCT (middle), and ViT (right).

## F.2 LOSS DISTRIBUTION PLOTS

The loss distribution plot clearly shows why POP and CS perform far more than the other methods for Item Removal. They can sufficiently move the forget set into a feasible distribution for Naive training when the other methods fail to. As a result, this shows that the Prune and Regrow Paradigm represents the best method for Item removal in different domains and speaks to its broader applicability. When we consider the class removal for CIFAR10, it can be observed that all methods bar FT do an excellent job at shifting the distribution, with AM, ST and POP performing the best.

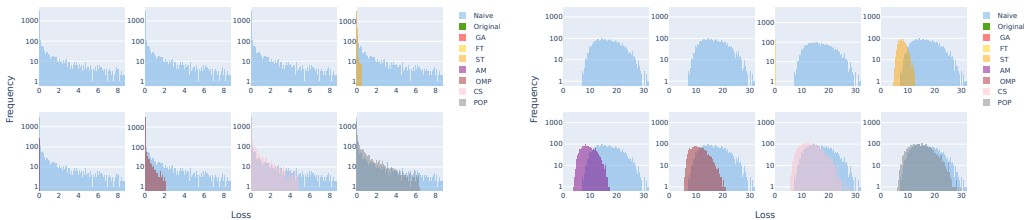

Figure 29: $\mathcal{D}_{forget}$ loss distribution on **CIFAR10**, for unlearning methods averaged across all seeds for the **VGG16**. 10% Item Removal (left) and 1 Class Removal (right). For each plot the unlearning method is compared to the loss distribution of $\mathcal{D}_{forget}$ on $\mathcal{M}^{\theta}$ and $\mathcal{M}_r^{\theta}$.

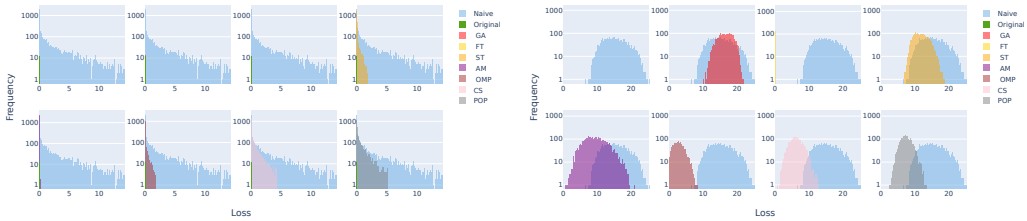

Figure 30: $\mathcal{D}_{forget}$ loss distribution on **CIFAR10**, for unlearning methods averaged across all seeds for the **CCT**. 10% Item Removal (left) and 1 Class Removal (right). For each plot the unlearning method is compared to the loss distribution of $\mathcal{D}_{forget}$ on $\mathcal{M}^{\theta}$ and $\mathcal{M}_r^{\theta}$.

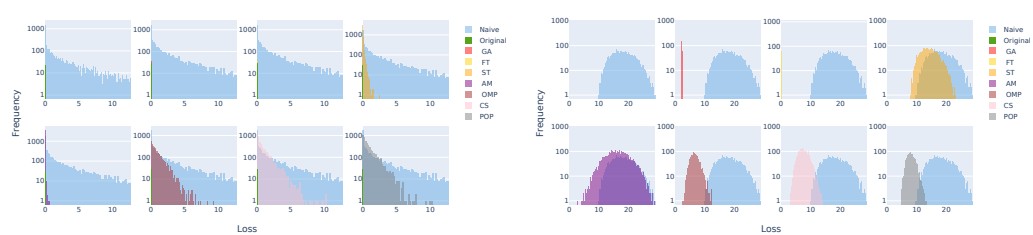

Figure 31: $\mathcal{D}_{forget}$ loss distribution on **CIFAR10**, for unlearning methods averaged across all seeds for the **ViT**. 10% Item Removal (left) and 1 Class Removal (right). For each plot the unlearning method is compared to the loss distribution of $\mathcal{D}_{forget}$ on $\mathcal{M}^{\theta}$ and $\mathcal{M}_r^{\theta}$.

### F.3 SCALLING RESULTS

When considering scaling, it can be observed that all methods for both Item and Class removal scale well across architectures apart from GA, which experience a large deviation and the amount of requests increases. Overall, this speaks to the stability of existing unlearning methods and the novel unlearning methods presented in the paper and the results align with what is observed for AudioMNIST, SpeechCommands and UrbanSounds8K.

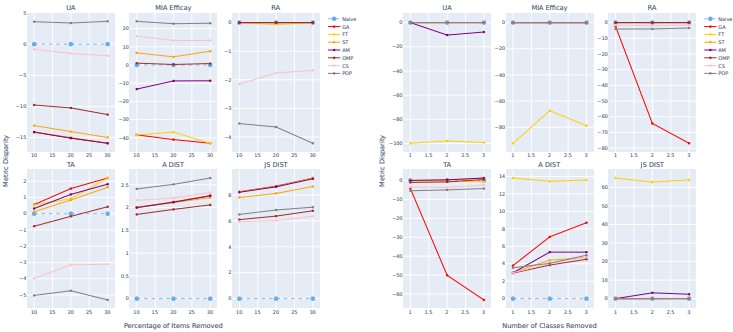

Figure 32: $\mathcal{D}_{forget}$ loss distribution on **CIFAR10**, for unlearning methods averaged across all seeds for the **VGG16**. 10% Item Removal (left) and 1 Class Removal (right). For each plot the unlearning method is compared to the loss distribution of $\mathcal{D}_{forget}$ on $\mathcal{M}^{\theta}$ and $\mathcal{M}_r^{\theta}$.

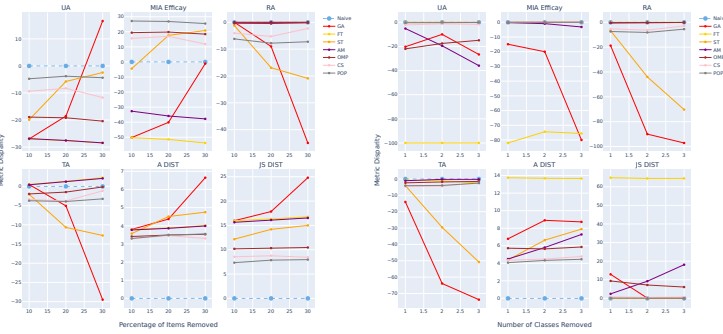

Figure 33: $\mathcal{D}_{forget}$ loss distribution on **CIFAR10**, for unlearning methods averaged across all seeds for the **CCT**. 10% Item Removal (left) and 1 Class Removal (right). For each plot the unlearning method is compared to the loss distribution of $\mathcal{D}_{forget}$ on $\mathcal{M}^{\theta}$ and $\mathcal{M}_r^{\theta}$.

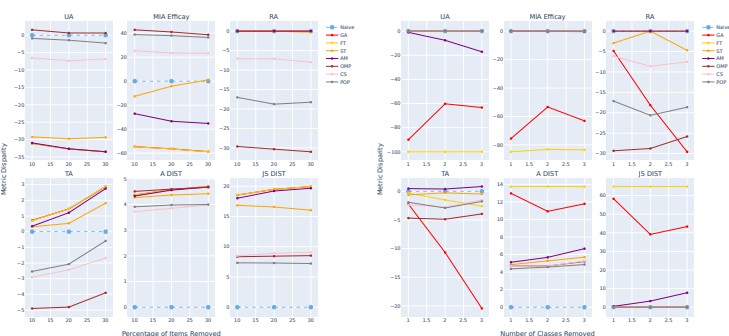

Figure 34: $\mathcal{D}_{forget}$ loss distribution on **CIFAR10**, for unlearning methods averaged across all seeds for the **ViT**. 10% Item Removal (left) and 1 Class Removal (right). For each plot the unlearning method is compared to the loss distribution of $\mathcal{D}_{forget}$ on $\mathcal{M}^{\theta}$ and $\mathcal{M}_r^{\theta}$.

## F.4 TRANSFERABILITY OF THE PRUNE AND REGROW PARADIGM

The results we present on the audio datasets and CIFAR10 demonstrate the potential of the Prune and Regrows dynamic sparsity and regrow process in improving unlearning capacity, particularly for Item Removal, a key unlearning challenge. While our study is primarily focused on the unlearning modality gap, we believe that our approach could be applied to other domains such as language and multi modal domains. This potential for broader application is an exciting avenue for future research.

