# OpenReview forum: "Machine Unlearning in Audio: Bridging The Modality Gap Via the Prune and Regrow Paradigm"
_ICLR.cc/2025/Conference — Submitted to ICLR 2025_

### Official Review · Reviewer_VLYY · 2024-10-30

**Soundness:** 3
**Presentation:** 3
**Contribution:** 3
**Rating:** 5
**Confidence:** 4

**Summary:**

This paper explores machine unlearning for audio classification. Five existing methods are used, and the authors use these methods to demonstrate the extent to which existing machine unlearning methods can remove individual items from the training data of three model architectures on two small datasets. The authors then propose a modified pruning method for machine unlearning that iteratively sparsifies and fine-tunes the network.

**Strengths:**

Machine unlearning has been neglected within the audio domain, and this is a nice paper that demonstrates the efficacy of existing SOTA machine unlearning on some simple audio classification models. It is a good step towards performing unlearning on modern audio models and tasks (e.g., speaker verification systems).

The motivation and background are well-written, concise, and not overly complicated.

**Weaknesses:**

This paper lacks any demonstration of generality due to the selected datasets (SpeechCommands and AudioMNIST) being too small. I don't think that is a reason to reject the paper, as machine unlearning has thus far not been sufficiently studied on any audio dataset.

Tables 2 and 3 do not paint a clear picture of the proposed method significantly improving over previous methods on the selected metrics. And tables in Appendix C show the proposed method performing the worst of all methods in some conditions (e.g., Table 7, A DIST). So while this paper contains good analysis, the proposed novel method may be an incremental improvement.

Section 3 should be more precise regarding details of your method. Right now, it says "the Prune and Regrow paradigm... focuses on...", but the methods section should provide sufficient details for both implementation and distinction from prior methods, not just the focus.

Please fix capitalizations, consistent conference/journal naming, arXiv papers that have been accepted, and other formatting issues in the references section

Check bolded values in Tables 2 and 3; there are maximal/best values that are not bolded

"we argue that a one-size-fits-all sparsity unlearning cannot be optimal due to different learnt features across modalities" -> This is not a well-supported claim, as the proposed method is not tested on any modality other than audio.

**Questions:**

One would assume that video tasks would also benefit from better machine unlearning methods for audio given the shared temporal dimension. Do you believe your method is amenable to video tasks? As well, are there existing machine unlearning methods for video tasks that would be applicable to the audio tasks performed in this paper?

Industry machine learning models are typically periodically retrained to add more data and experiment with the training and model architecture. It seems far more straightforward to simply remove the D_{forget} set when retraining. Are there concrete, real-world cases where the model is not retrained and machine unlearning is needed?

---

> ### Author Response · Authors · 2024-11-27
> **Response to Reviewer VLYY**
>
> Thank you for the time you have dedicated to reviewing our paper. Your comments have helped us improve both the method and its description immensely.
>
> 1. This paper lacks any demonstration of generality due to the selected datasets (SpeechCommands and AudioMNIST) being too small.
>
> Please see ‘Overall comments to reviewers, point 3’ for accompanying explanation.
>
> In response to this and other reviewers, we have expanded the datasets we test on to include UrbanSound8K, an event classification audio dataset, and CIFAR 10, an image classification dataset as a preliminary study on another modality. We find that the Prune and Regrow Paradigm is by far the best on CIFAR10 (Appendix F) for Item Removal. As this study represents one of the first comprehensive studies on unlearning in audio, we establish an essential foundation for future research to expand upon by exploring larger models and datasets.
>
> 2. The proposed novel method may be an incremental improvement.
>
> Our study represents one of the first comprehensive studies on unlearning in audio, where we establish an essential foundation for future research to expand upon by exploring larger models and datasets. Furthermore, our proposed approach achieves superior results on a range of datasets, demonstrating its effectiveness and usefulness for machine unlearning. We therefore argue that our contributions are substantial.
>
> Another advantage of the Prune and Regrow Paradigm is that it allows for repeated unlearning, given the fact that all the parameters are updated during fine-tuning (due to the Regrow phase). This means that an unlearnt model can repeatedly be unlearned on and reach a different representation. This is not true for the standard OMP method as no weights are updated bar those at the 0.95% Sparsity, and repeatedly pruning these values would lead to model collapse.
>
> Moreover, any improvement towards unlearning represents a step towards improved data privacy and the ability to adhere to GDPR in a sustainable manner.
>
> 3. "we argue that a one-size-fits-all sparsity unlearning cannot be optimal due to different learnt features across modalities" -> This is not a well-supported claim, as the proposed method is not tested on any modality other than Audio.
>
> We would like to provide further clarification on the intent behind this statement.  The OMP unlearning method proposed by [2] states that 0.95 sparsity is effective for unlearning. The value of 0.95 is decided based on empirical studies [2] conducted on CIFAR 10. The statement we made above refers to the fact that 0.95% sparsity may provide a good unlearning capacity but is not optimal for all modalities, or architectures, as seen in our results.
>
>  As a result, we have employed L1 pruning, preserving cosine similarity to pick the appropriate sparsity value (Prune), and then reinitialise the zeroed weights (Regrow) and fine-tune to improve performance. We have updated this description to reflect the above better and apologise for any confusion.
>
> As a side note, we have also demonstrated our results on CIFAR10 and shown that our Prune and Regrow Paradigm provides the best Item removal for all explored architectures (Appendix  F).
>
> References:
>
> [1] Mason-Williams, G. and Dahlqvist, F., 2024. What Makes a Good Prune? Maximal Unstructured Pruning for Maximal Cosine Similarity. In The Twelfth International Conference on Learning Representations.
>
> [2] Liu, J., Ram, P., Yao, Y., Liu, G., Liu, Y., SHARMA, P. and Liu, S., 2024. Model sparsity can simplify machine unlearning. Advances in Neural Information Processing Systems, 36.

---

> > ### Author Response · Authors · 2024-12-02
> > **Response to VLYY**
> >
> > Hi,
> >
> > We wanted to reach out and ask if there are any further points of clarification you require on our work?
> >
> > Your feedback was very useful in helping to improve our work and we wanted to make sure that you feel your suggestions have been addressed. Also, we are more than happy to explain our rationale behind our decisions if you are unhappy with the revisions.
> >
> > Many thanks.

---

> > > ### Comment · Reviewer_VLYY · 2024-12-02
> > >
> > > I acknowledge the authors' rebuttal and the corresponding improvements made to the paper. I maintain my current review score.

---

> ### Author Response · Authors · 2024-12-03
> **Clarification**
>
> Please may you clarify why the score has remained as it is or where you feel we did not meet your requirements for a higher score?
>
> We have addressed the issue around using minimal datasets by increasing the number to four and show that our method presents state of the art results for Item Removal for more complex datasets such as SpeechCommands, UrbanSounds8K and CIFAR10 which is the most important task in machine unlearning literature.
>
>
> Additionally, we have described why our method is more performant and the benefits it yields over using fixed sparsity pruning. If you would like further clarification on why this is important we can provide it. Furthermore, the description has been modified to highlight this.
>
> The issues surround formatting errors have been addressed and finally we have further supported our claim of generalisability of the method by showing its success on CIFAR - being the best for Item removal and highly competitive for Class removal with the corresponding scaling experiments and loss distribution analysis.

---

### Official Review · Reviewer_Zz5w · 2024-11-03

**Soundness:** 2
**Presentation:** 2
**Contribution:** 2
**Rating:** 3
**Confidence:** 4

**Summary:**

This paper addresses a gap in the unlearning literature by adapting and evaluating existing unlearning methods, previously applied in other domains, to audio data—specifically speech recognition datasets such as AudioMNIST and SpeechCommands V2—across various architectures. The authors find that while current unlearning techniques perform adequately for Class Removal, they fall short for Item Removal, an important task in data unlearning. To address this, they introduce a novel Prune and Regrow paradigm that applies dynamic sparsity unlearning. Additionally, their investigation into unlearning scaling shows that this method maintains superior performance even as Item Removal requests increase.

**Strengths:**

The paper’s strengths lie in its focus on adapting unlearning methods to audio data, filling a gap in existing literature, and addressing unique challenges in speech recognition. The introduction of the Prune and Regrow paradigm is particularly interesting, as it not only enhances unlearning effectiveness for Item Removal—an area where current unlearning methods struggle—but also proves scalable for increasing unlearning requirements.

**Weaknesses:**

1. The explanation of the Prune and Regrow paradigm lacks clarity. I recommend that the authors include a structured figure illustrating each step—such as the specific pruning methods within the paper. Additionally, the authors could create space by trimming subsections 2.2 and 2.3, as the details on unlearning methods and evaluation metrics are already available in the appendix.

2. While the authors claim that the Prune and Regrow paradigm outperforms prior unlearning methods for Item Removal, this appears to hold primarily when measured by Unlearning Accuracy (UA) (as shown in Table 2). I would be interested to know why the results do not generalize across other metrics.

3. I recommend that the authors add details to clearly distinguish between Item Removal and Class Removal unlearning methods.

4. The paper lacks comparisons on common speech recognition benchmarks. How well does this method generalize to widely used speech recognition datasets, such as Libri-Light and LibriSpeech?

**Questions:**

I have addressed all my questions in the Weakness section.

---

> ### Author Response · Authors · 2024-11-27
> **Response to Reviewer Zz5w**
>
> We greatly appreciate your review of our paper and thank you for recognising the scalability of the Prune and Regrow Paradigm and its ability to bridge the modality gap in unlearning for audio.
>
> 1. While the authors claim that the Prune and Regrow Paradigm outperforms prior unlearning methods for Item Removal, this appears to hold primarily when measured by Unlearning Accuracy (UA) (as shown in Table 2). I would be interested to know why the results do not generalise across other metrics.
>
> We consider UA the most useful metric for assessing machine unlearning. Firstly, a machine unlearning method may have a sizable unlearning gap for UA but retain performance on RA and TA. This is often witnessed in the results for the GA and FT methods. This means that, while it has been unable to unlearn, it has retained its accuracy and generalisation. Therefore, when calculating the disparity average, it can average out its discrepancy on the UA with its performance on RA and TA.
>
> For example, say the disparity for an unlearning method X and Y to the Naive method across UA, MIA, RA and TA is:
>
> X: UA = -7 MIA = -10 RA = 1  TA=1 then D AVE = ((-7)+(-10) + (1) + (1))/4 = -3.75
>
> Y: UA =1 MIA= 1 RA = -10 TA=-7 then D AVE = ((1)+(1) + (-10) + (-7))/4 = -3.75
>
> From a D AVE perspective, these methods could be considered equivalent. However, it is evident that method Y has better ability to unlearn the data in the forget set; while this comes at the cost of the remaining accuracy (RA)  and generalisation (TA), it is better at unlearning, which is the primary focus. Also, for method Y, TA and RA could be recovered with further fine-tuning. As a result, D AVE calculations in the paper have been modified to take in only absolute values to capture the results better. Additionally, given method X’s inability to remove UA, it should be disregarded from analysis as it is not an effective unlearning method and should not be compared to methods that do perform effectively on UA, which is the key metric of interest.
>
> Additionally, for distance-based metrics, arguments have been made that, while they are indicative, they only offer a proxy for the amount unlearned, as the comparison is only to  one possible model that is retrained on the forget set for each seed. Given models can have different representations when trained on the same data and reaching similar accuracy and loss [2], this is a challenge, and the same is true for JS-Divergence. While these metrics are imperfect, they represent literature's best attempt to quantify strong unlearning [3]. As a result, we consider UA as the most critical metric. Additionally, with the modifications to the Prune and Regrow Paradigm, we see in the new results that the methods generalise better. We have increased the robustness of the results by averaging over 10 instead of 5 seeds.
>
> 2. I recommend that the authors add details to clearly distinguish between Item Removal and Class Removal unlearning methods.
>
> Thank you for your suggestion. We agree that methods that perform well on one removal task but not the other simply indicate that they are specialized for one specific purpose and may not generalize to both. This does not mean they fail at their intended task; rather, it highlights their limitations in broader contexts.
>
> We believe that a successful unlearning method is one that can unlearn competitively in both Item and Class removal regimes. In the results, this is witnessed for the non-pruning-based methods, GA, ST and AM, which perform well on Class removal but fail to deliver for Item removal. Existing literature compares unlearning methods for both Item and Class removal without distinction [4] [5].
>
> 3. The paper lacks comparisons on common speech recognition benchmarks. How well does this method generalise to widely used speech recognition datasets, such as Libri-Light and LibriSpeech?
>
> To increase the diversity of audio tasks we evaluate on, we run experiments on the event sound detection dataset, UrbanSound8K (presented in the main body of our revised draft). Additionally, to show the Prune and Regrow’s applicability in another modality (in response to VLYY, Byqm, and PAef) and present results on CIFAR10 (Appendix F), which demonstrate the effectiveness of  the Prune and Regrow Paradigm. In future work, it would be interesting to extend our method to other datasets such as the ones suggested here, taking into account the scalability implications for large-scale data.

---

> > ### Author Response · Authors · 2024-11-27
> > **References**
> >
> > References:
> >
> > [1] Mason-Williams, G. and Dahlqvist, F., 2024. What Makes a Good Prune? Maximal Unstructured Pruning for Maximal Cosine Similarity. In The Twelfth International Conference on Learning Representations.
> >
> > [2] Fort, S., Hu, H. and Lakshminarayanan, B., 2019. Deep ensembles: A loss landscape perspective. arXiv preprint arXiv:1912.02757.
> >
> > [3] Nguyen, T.T., Huynh, T.T., Nguyen, P.L., Liew, A.W.C., Yin, H. and Nguyen, Q.V.H., 2022. A survey of machine unlearning. arXiv preprint arXiv:2209.02299.
> >
> > [4] Liu, J., Ram, P., Yao, Y., Liu, G., Liu, Y., SHARMA, P. and Liu, S., 2024. Model sparsity can simplify machine unlearning. Advances in Neural Information Processing Systems, 36.
> >
> > [5] Oesterling, A., Ma, J., Calmon, F. and Lakkaraju, H., 2024, April. Fair machine unlearning: Data removal while mitigating disparities. In International Conference on Artificial Intelligence and Statistics (pp. 3736-3744). PMLR.

---

> > > ### Author Response · Authors · 2024-12-02
> > > **Response to Zz5w**
> > >
> > > Hi,
> > >
> > > We hope you are doing well and wanted to reach out concerning further feedback on our work. We understand that it is a very busy period but wanted to reach out just in case you had any further questions concerning the revisions we have completed based on your feedback.
> > >
> > > We would be more than happy to discuss any final points and clarify anything further if required.

---

### Official Review · Reviewer_2Puu · 2024-11-03

**Soundness:** 1
**Presentation:** 1
**Contribution:** 2
**Rating:** 3
**Confidence:** 3

**Summary:**

The authors introduce the Prune and Regrow method and tested on AudioMNIST and Speech Commands V2, showing that the proposed method is effective for item removal and class removal.

**Strengths:**

Overall, this is an important topic, especially as AI is trained on large datasets crawled from websites. There is an urgent need for machine unlearning in the audio domain.

**Weaknesses:**

Overall, I believe this paper has a large space for improvement:

1/ The method is described in too simple a way. The background and introduction are 3 pages, but the method is described in only 2/3 of a page and mostly in vague text descriptions. I am not an expert in machine unlearning, but I suspect such a general method (Prune and Regrow) is not being proposed for the first time. If I am wrong, please correct me (e.g., Prune and Regrow is the first time proposed in this area).

2/ The experiment is limited to 2 datasets, both not related to privacy directly. I understand that for some theoretical papers, such small datasets are needed for analysis purposes, but this is not a theoretical paper. The gap between a real-world model (let's say OpenAI's audio model Whisper vs. the model presented in this paper) can be huge, and thus the analysis in this paper may not transfer.

3/ The presentation is not clear. The core method is described in too brief a way. Figures are not easy to read (font too small, why a radar plot?). Figure 3 presents too many plots, and I don't easily understand the message the authors want to convey.

**Questions:**

See in the weakness.

---

> ### Author Response · Authors · 2024-11-27
> **Response to Reviewer 2Puu**
>
> Thank you for your review. We agree that machine unlearning is urgently needed in the audio domain.
>
> 1. I suspect such a general method (Prune and Regrow) is not being proposed for the first time
>
> This is the first time the Prune and Regrow Paradigm has been proposed and applied to machine unlearning research. Typically, pruning is used as a compression technique to reduce the number of parameters [1], [2], and, therefore, people do not consider using it as a regrow method as it subverts its use case. Additionally, sparsity-based unlearning, such as OMP, was only introduced as a viable unlearning method (outside of federated settings [3]) in December 2023 at NeurIPS [4] and since there has been little work improving sparsity unlearning.
>
> We are pleased that you think the method is simple as it is a requirement for any unlearning method to be intelligible [5] and we think the simplicity and effectiveness of our method achieves this.
>
> 2. The experiment is limited to 2 datasets, both not related to privacy directly.
>
> The GDPR act states: “Biometric data means personal data resulting from specific technical processing relating to the physical, physiological or behavioural characteristics of a natural person, which allow or confirm the unique identification of that natural person, such as facial images or dactyloscopic data” (https://gdpr-info.eu/art-4-gdpr/). As a result, any voice data can be considered biometric data and have grounds for the Right To Be Forgotten. Consequently, both datasets are privacy datasets by definition as they have human voice data. Typically unlearning experiments are conducted on SHVN, MNIST and CIFAR10 [6][7][8] which are more abstract in regards to biometric data than the datasets explored in our work.
>
>
> 3. The presentation is not clear. The core method is described in too brief a way.
>
> To address this, we have reduced the number of figures in the main body of the paper, split them up, put some into the appendix, and referenced the figures where necessary. We hope that this makes the results and the subsequent discussion easier to follow. We have also increased the font sizes to improve readability. We hope this improves the clarity of the results.
>
> 4. Why a radar plot?
>
> We use radar plots in line with the presentation provided by the NeurIPS spotlight paper [4]. We believe that it provides an intuitive and holistic understanding of the impact of unlearning approaches across all of the accuracy and attack-based metrics. We have increased their size and font size to improve readability. We have also now moved these to the appendix to highlight other results; however, they do contextualize the relationship between UA, RA and TA for Item Removal.
>
> References:
>
> [1] LeCun, Y., Denker, J. and Solla, S., 1989. Optimal brain damage. Advances in neural information processing systems, 2.
>
> [2] Mason-Williams, G. and Dahlqvist, F., 2024. What Makes a Good Prune? Maximal Unstructured Pruning for Maximal Cosine Similarity. In The Twelfth International Conference on Learning Representations.
>
> [3] Wang, J., Guo, S., Xie, X. and Qi, H., 2022, April. Federated unlearning via class-discriminative pruning. In Proceedings of the ACM Web Conference 2022 (pp. 622-632).
>
> [4] Liu, J., Ram, P., Yao, Y., Liu, G., Liu, Y., SHARMA, P. and Liu, S., 2024. Model sparsity can simplify machine unlearning. Advances in Neural Information Processing Systems, 36.
>
> [5] Bourtoule, L., Chandrasekaran, V., Choquette-Choo, C.A., Jia, H., Travers, A., Zhang, B., Lie, D. and Papernot, N., 2021, May. Machine unlearning. In 2021 IEEE Symposium on Security and Privacy (SP) (pp. 141-159). IEEE.
>
> [6] Shaik, T., Tao, X., Xie, H., Li, L., Zhu, X. and Li, Q., 2024. Exploring the landscape of machine unlearning: A comprehensive survey and taxonomy. IEEE Transactions on Neural Networks and Learning Systems.
> [7] Nguyen, T.T., Huynh, T.T., Nguyen, P.L., Liew, A.W.C., Yin, H. and Nguyen, Q.V.H., 2022. A survey of machine unlearning. arXiv preprint arXiv:2209.02299.
>
> [8] Xu, J., Wu, Z., Wang, C. and Jia, X., 2024. Machine unlearning: Solutions and challenges. IEEE Transactions on Emerging Topics in Computational Intelligence.

---

> > ### Comment · Reviewer_2Puu · 2024-12-02
> > **thanks for the response**
> >
> > I appreciate the authors' efforts in improving the paper and addressing my questions. I believe the paper has been improved. Nonetheless, in my opinion, the explanations for points 2 and 3 are still quite limited, and the overall quality remains below the bar for ICLR. For example, there are still too many numbers and plots, not organized in a good way.

---

> ### Author Response · Authors · 2024-12-02
> **Further questions for Reviewer 2Puu**
>
> Thank you for acknowledging the improvements in our paper and responding to our comments.
>
> We have further questions to clarify some of the further points you have raised.
>
> 1. Not addressing "The experiment is limited to 2 datasets, both not related to privacy directly."
>
> We have addressed this point by expanding the experiment to cover 4 datasets (AudioMNIST, SpeechCommands, UrbanSounds8K and CIFAR10) instead of 2. What number of datasets would be sufficient to expand the number of datasets?
>
> Furthermore, given that GDPR dictates that any personal voice data is Biometric data and, therefore, is protected by privacy law - how exactly are the audio datasets not related to privacy?
>
> Are you familiar with any unlearning studies that use such privacy-related datasets you are requesting? As most studies, as acknowledged by surveys [6][7][8], show that this is not standard practice in unlearning literature, which primarily uses CIFAR10, SHVN and MNIST, which do not contain biometric data - our audio study does contain such biometric data (protected by GDPR).
>
> 2. Not addressing: "The presentation is not clear. The core method is described in too brief a way."
>
> We would like some constructive feedback on how we could better explain the core method if you feel the figures we added are insufficient.
>
> Given that our method is based on using a sparsity value determined by a method presented at ICLR last year [2], we feel we have described it well without recreating the original paper's work and we also provide the supporting references which sufficiently explain the cosine based pruning.
>
> Do you have any suggestions of what more you would like to see to explain the method better?
>
> Given your last comments, we reduced the number of figures in the main body substantially; however, due to your request for more datasets, we added two more tables, further supporting our method's superior ability to unlearn and strengthen our original conclusion.
>
> How would you like our current presentation to change to satisfy your requirements for a straightforward paper?
>
> We could address any presentation issues in a revised version at the camera-ready version, so we would appreciate your suggestions.
>
> (For references please look at our previous response to this review).

---

### Official Review · Reviewer_Byqm · 2024-11-03

**Soundness:** 3
**Presentation:** 3
**Contribution:** 2
**Rating:** 6
**Confidence:** 3

**Summary:**

The authors study and evaluate different unlearning methods on two speech datasets, AudioMNIST and SpeechCommandsV2. They compare different setups of item and class removal, different sizes of the forget set, and different architectures. They report results on a wide variety of metrics.

The authors propose a new method of unlearning "Prune and Regrow" as an improvement of pruning methods for unlearning. They show the proposed method does offer better performance on various metrics on the studied datasets.

Edit: In light of the authors reply and the added experiments, I have raised my scores for the paper.

**Strengths:**

- Extensive evaluation of different metrics and comparing different methods.
- Comparing different sizes of the forget set and different architectures.
- Good presentation and structure of the paper.
- Good explanation of the metrics and methods.

**Weaknesses:**

1. Unjustified focus on audio: Although the authors focus their presentation on audio, I didn't understand what makes the study or proposed method relevant to audio beside the datasets. The method can be applied to any NN. I don't see what makes audio special in this case and why the authors evaluated the proposed methods only on speech, or what characteristics of speech are relevant for this method to work. Suggestions:  (a) Comparing the results to similar experiments on non-audio tasks to highlight any audio-specific findings.
 (b) Explaining any unique challenges or considerations for unlearning in audio domains.

2. Limited evaluation for audio considering only speech tasks: I think the audio domain is quite diverse. However, the authors focused the study on simple speech recognition tasks. I think the claims made in the paper are too broad for audio in general; other tasks must be considered, for example, music tasks (music genre classification), environmental sound recognition, and acoustic event detection and classification tasks. I agree that privacy concerns are paramount in speech tasks, but licensing and copyrights can also highlight the importance of unlearning in other audio tasks. Suggestions: Audioset[3] and FSD50k[7] , ESC50[6] for general audio tagging, GTZAN[4] and Jamendo[5] for music tasks
3. Training setup and not using audio models: The authors use (with the exception of the old VGGish model) models not adapted for audio. For example, on speech commands, why did the authors use ViT instead of the adapted AST [1] (ViT with patch overlap and pre-training) ? AST achieves 98% test accuracy on speech commands v2 35 classes, compared to the reported results of 84% ViT. This casts doubts on the validity of the results. Suggestions: (a) Consider running additional experiments with audio-specific architectures; (b) A discussion how the choice of architecture can impact the results; (c) A discussion whether the conclusions would hold for more specialized audio models

Minor comments:
- I think the figures can be made larger to be more readable using the white space surrounding the plots.
- typo in line 172

[1] Gong, Yuan, Yu-An Chung, and James Glass. "Ast: Audio spectrogram transformer." arXiv preprint arXiv:2104.01778 (2021).

[2] https://paperswithcode.com/sota/keyword-spotting-on-google-speech-commands

[3] J. F. Gemmeke, D. P. W. Ellis, D. Freedman, A. Jansen, W. Lawrence, R. C. Moore, M. Plakal, and M. Ritter, “Audio set: An ontology and human-labeled dataset for audio events,” in ICASSP 2017, New Orleans, LA, 2017.

[4] https://www.kaggle.com/datasets/andradaolteanu/gtzan-dataset-music-genre-classification

[5] https://www.jamendo.com/?language=en

[6] K. J. Piczak, “ESC: Dataset for Environmental Sound Classification,” in Proceedings of the 23rd Annual ACM Conference on Multimedia. ACM Press, 2015, pp. 1015–1018.

[7] E. Fonseca, X. Favory, J. Pons, F. Font, and X. Serra, “FSD50K: an open dataset of human-labeled sound events,” IEEE ACM Trans. Audio Speech Lang. Process., vol. 30, pp. 829–852, 2022.

**Questions:**

Can you please address the weeknesses above.

---

> ### Author Response · Authors · 2024-11-27
> **Response to Reviewer Byqm**
>
> We thank the reviewer for dedicating their time to providing an in-depth review of our submission. We appreciate your recognition of the paper’s strengths, including the comprehensive evaluation, scaling experiments, presentation, and detailed explanations.
>
> As a direct response to your review, we have included the results for Item and Class removal for the event classification dataset UrbanSound8K (in the main body) and the results of the same experiments for the vision CIFAR10 dataset (in Appendix F), with a discussion on the method's effectiveness across domains.
>
> 1. Unjustified focus on audio The method can be applied to any NN:
>
> Existing machine unlearning literature has covered almost every domain besides the audio domain (see TABLE I Public Datasets For Machine Unlearning With Supplementary Information For Popularity from the supplementary material of [1]; and Table 5, Published Datasets from [2]). While reviewer PAef pointed to another contemporaneous, small-scale study in audio unearning [3], there is no other literature in this direction.
>
> Ensuring that machine unlearning approaches are universal is crucial, and, therefore, research needs to extend to every data domain - as the Right To Be Forgotten does not discriminate by modality. This sentiment has been shared by reviewers 2Puu, Zz5w and VLYY. Given that audio data undergoes different transformations, such as conversions into mel spectrogram or Cochleagram or wavelet transform formats, they present a different unlearning challenge that needs to be studied and understood.
>
> We have created our method to be dynamic so that it can be applied to any NN. This is because current literature creates static methods that are unlikely to perform well in different domains without high levels of specialisation. Our results on CIFAR 10 (Appendix F) show that this dynamic approach extends to other data domains too. Our findings can serve as a benchmark for the community, encouraging the development of more dynamic unlearning methods that are specifically tailored to neural networks by design, rather than relying on hyperparameter optimization.
>
> 2. Limited evaluation for audio considering only speech tasks:
>
> To address this, we ran the Item and Class removal experiments on the event classification task of UrbanSounds8K - we present the tables and scaling results for this dataset. The results show that the Prune and Regrow Paradigm is superior for Item removal in this dataset.. However, in response to yourself, reviewer VLYY and PAef we ran the experiments on CIFAR10 (Appendix F) to show that the Prune and Regrow Paradigm provides superior Item removal for all architectures. This indicates that the Prune and Regrow’s dynamic sparsity and ability for regrow can transfer to other modalities. We would like future work to analyze its success in a range of modalities including written language.
>
> 3. Training setup:
>
> Our training setup has been devised to reduce complexity to isolate the impact of each unlearning method's effects without interference from other factors. As a result, we only use SGD with lr=0.01 and momentum of 0.9 and try to remove the influence of explicit regularisers such as weight decay and dropout to do this. Additionally, no data augmentation is used to improve accuracy as the core focus of this setup is to evaluate unlearning. As a result, the accuracy of the models trained in this vanilla setting is lower than that of the models trained with such explicit regularisation. While the accuracy is lower, it provides a more precise insight.
>
> For example, the models in this setup on AudioMNIST have a tiny generalisation gap. As a result, each unlearning method appears quite successful; however, on SpeechCommands, UbranSound8K and CIFAR10, where this generalisation gap is more significant, the difference between the performance of the unlearning methods is more stark for item removal. In this case, the generalisation gap helps to identify between methods capable of item removal and those not, making the generalisation gap useful too for unlearning method evaluation. If the generalisation gap was less, it may have created the illusion of a more efficacious unlearning when this is not actually the case, as we see for AudioMNSIT. Our concise training setup highlights these factors nicely.
>
> Additionally, using a model that has been trained on another dataset can make the interpretation of the results less apparent, as it is unclear how the data in the pre-trained set could represent the data to remove.

---

> ### Author Response · Authors · 2024-11-27
> **Part 2:**
>
> 4. Not using Audio Models/AST
>
> In the review, the Audio Spectrogram Transformer is referenced as a specialized audio model. Hugging face (https://huggingface.co/MIT/ast-finetuned-audioset-10-10-0.4593) describes the AST as: “The Audio Spectrogram Transformer is equivalent to ViT, but applied on audio. Audio is first turned into an image (as a spectrogram), after which a Vision Transformer is applied.” and the AST paper remarks that “Even with no patch split overlap, AST can still outperform the previous best system” [4]. In our paper, we use mel spectrogram and then pass it to a ViT which, by the hugging face standard, is equivalent to the AST - however the AST uses pre-training to get better results. As a result we believe we do cover this architecture but in a manner that is consistent with our experimental set up.
>
> In the paper, we state that "the architectures explored cover a range of capacities (Table 5) and core architecture differences". To elaborate on this further, we choose models that represent core architectural differences. The VGGish model comprises only convolutional layers representing a class of models; the CCT, conversely, has a combination of convolutional layers and attention mechanisms; this distinct class of models may interact differently with unlearning methods, and we wanted to explore this.
> Finally, the ViT uses mainly attention mechanisms, representing a different class of models that may interact differently with unlearning methods, making an excellent architecture to examine. As a result, the architectural differences well represent the classes of available models and, therefore, should be used for these experiments.
>
> We would expect our findings to generalise as we cover the main model classes for deep learning, convolutional networks, convolution and attention and purely attention based architectures.
>
> References:
>
> [1] Shaik, T., Tao, X., Xie, H., Li, L., Zhu, X. and Li, Q., 2024. Exploring the landscape of machine unlearning: A comprehensive survey and taxonomy. IEEE Transactions on Neural Networks and Learning Systems.
>
> [2] Nguyen, T.T., Huynh, T.T., Nguyen, P.L., Liew, A.W.C., Yin, H. and Nguyen, Q.V.H., 2022. A survey of machine unlearning. arXiv preprint arXiv:2209.02299.
>
> [3] Cheng, J. and Amiri, H., 2024. Mu-bench: A multitask multimodal benchmark for machine unlearning. arXiv preprint arXiv:2406.14796.
>
> [4] Gong, Y., Chung, Y.A. and Glass, J., 2021. Ast: Audio spectrogram transformer. arXiv preprint arXiv:2104.01778.

---

> ### Comment · Reviewer_Byqm · 2024-12-01
> **Reply to the authors**
>
> I thank the authors for the reply and the added experiments to the paper. In light of the new additions and clarification, I will assign to the paper a higher score.
>
> I appreciate the importance of the authors work in investigating the unlearning methods for audio and agree that it's understudied in the literature. I wouldn't give the paper a higher score because the following points still stand: I understand the use of pre-trained models would create a more complex problem, but it's a standard in the audio literature. Since the focus is Audio, I think the study should use the common methods that perform the best on the respective tasks. Otherwise, I think the study is using audio datasets but is not particularly relevant for the audio community. Additionally, models like AST, are pretrained on images which are out of domain, and I think it is not relevant to forgetting parts of the audio training set.

---

> > ### Author Response · Authors · 2024-12-02
> > **Question for reviewer:**
> >
> > We thank the reviewer for considering our further work and increasing the score of our paper. Your suggestions significantly improved the work, and we thank you for that.
> >
> > We have a question for the reviewer that may help further describe our perspective on not employing the pre-trained model.
> >
> > If pretraining (even on Image datasets such as the case for the AST) enables a more informed prior (initialisation point) that allows improved accuracy, is the model not using features learned from the previous dataset to enable it to get better accuracy?
> >
> > As we see that this is the case (with the AST and other models), is it not unlikely that some examples within the training set will be better represented by the pre-trained weights - as a result, would this not hypothetically impact the ability to unlearn an example in a way that is unclear and hard to quantify?
> >
> > We see this as a critical barrier to exploring unlearning methods for pre-trained models for the first time in audio and believe that a separate paper would be required to explore this question sufficiently.

---

### Official Review · Reviewer_PAef · 2024-11-05

**Soundness:** 2
**Presentation:** 2
**Contribution:** 2
**Rating:** 6
**Confidence:** 4

**Summary:**

This paper investigates the effectiveness of existing machine unlearning methods in the audio domain, focusing on the datasets AudioMNIST and SpeechCommands. The authors show that current unlearning techniques work for class removal but not item removal. To address this, they introduce Cosine Similarity and Post Optimal Prune within a Prune and Regrow paradigm and show improved performance (83% unlearning accuracy gap) for Post Optimal Prune.

**Strengths:**

1. This is the first detailed analysis of machine unlearning in audio.

2. The proposed Post Optimal Prune method shows significantly improved performance for item removal unlearning accuracy gap.

**Weaknesses:**

1. The paper does not refer to [1], which provides a multimodal database for machine unlearning including audio and shows issues with machine unlearning with speech commands (see Figure 11). The submitted paper has a more detailed analysis with proposed improvement for machine unlearning, but this claim is false: "Our analysis highlights, for the first time, that, in audio, existing methods fail to remove data for the most likely case of unlearning – Item Removal."

2. Where is the 83% unlearning accuracy gap claimed in the abstract and line 69 explained? I don't see this in Table 2 or anywhere in the paper. If I average the unlearning accuracy gap for VGGish, CCT, and ViT I get 82.0%.

3. No source code is provided. The description of Cosine Similarity and Post Optimal Prune is insufficient to reproduce.

4. The author should explain why VGGish, CCT, and ViT are used. The models used in [1] make more sense: HuBERT (Hsu et al., 2021a) (Base, Large, X-Large), Wav2Vec2.0 (Baevski et al., 2020a) (Base, Large), Whisper (Radford et al., 2022) (Tiny, Base)

5. There should be more motivation in the introduction why this problem is important. For example, I understand why item removal is important (a person wants to remove all of their audio in a dataset) but why is class removal important for request to be forgotten?

6. There is no discussion if the proposed solution would comply with the GDPR requirements. What level of machine unlearning performance is required to comply with GDPR?

7. The paper does not explore the applicability of the Prune and Regrow paradigm to other data modalities like images, text, or video, which limits the impact of the findings.

[1] https://arxiv.org/pdf/2406.14796

**Questions:**

Figure 1-4 are not readable - please increase the font sizes.

---

> ### Author Response · Authors · 2024-11-27
> **Response to Reviewer PAef:**
>
> The authors thank PAef's for their time and thorough review of the paper and appreciate the highlighted weaknesses that we have improved upon during the rebuttal period.
>
> 1. The paper does not refer to [1]
>
> We thank the reviewer for bringing [1] to our attention. We note that this paper was only released in the last few months (late June 2024). Additionally, we have updated our paper to state that "Our analysis highlights that, in audio, existing methods fail to remove data for the most likely case of unlearning – Item Removal."
>
> Our work is different from [1] in many ways: we do not use pre-trained models as they cause issues with comparisons to the Naive model; we explore three audio datasets as opposed to just SpeechCommands; investigate both Class and Item Removal instead of only Item Removal; investigate the loss distribution of unlearning methods and how methods scale as the proportion of Items or Classes to be removed increases.
>
> Additionally [1] uses pre-trained models as uses the baseline of random accuracy to measure unlearning success which is incorrect due to models having the ability to generalise to unseen data as shown through Naive retraining.
>
> Furthermore, we present a novel Prune and Regrow Paradigm that receives the best UA for the majority of Item Removal (the most likely unlearning request) experiments and is very competitive on Class removal. It also gets the best performance on Item Removal on the CIFAR10 dataset (Appendix F)  which demonstrates the method’s versatility and generalisability.
>
> 2. Where is the 83% unlearning accuracy gap claimed in the abstract and line 69 explained?
>
> Our claim for the Prune and Regrow Paradigm achieving the best unlearning accuracy for 83% of Item Removal experiments is based on the results presented in the Item removal tables for SpeechCommands and AudioMNIST. In total, 6 Item removal experiments are conducted for the results in the tables - we record that for ⅚ of those results, CS or POP have the best UA result compared to the Naive method: (⅚)*100 = 83.3%. We have since updated this figure in response to the new results we have presented. We note that now all results are averaged over 10 random seeds, enhancing the robustness and reliability of our conclusions.
>
> 3. No source code is provided. The description of Cosine Similarity and Post Optimal Prune is insufficient to reproduce.
>
> We will be releasing all source code and models used in our paper upon acceptance. We have also submitted a  revised version of our paper,  where we have provided a more in-depth description of our method and its improvements. We have also included Figures 1 and 2 in the main body to better explain the process of the Prune and Regrow Paradigm and the relationship between cosine similarity and pruning.
>
> 4. The author should explain why VGGish, CCT, and ViT are used. The models used in [1] make more sense: HuBERT (Hsu et al., 2021a) (Base, Large, X-Large), Wav2Vec2.0 (Baevski et al., 2020a) (Base, Large), Whisper (Radford et al., 2022) (Tiny, Base)
>
> Please see ‘Overall comments to reviewers, point 2’ for accompanying explanation.
>
> In the paper, we state that "the architectures explored cover a range of capacities (Table 5) and core architecture differences". To elaborate on this further, we choose models that represent core architectural differences. The VGGish model comprises only convolutional layers representing a class of models; the CCT, conversely, has a combination of convolutional layers and attention mechanisms. This distinct class of models may interact differently with unlearning methods, which we set to investigate in our work
>
> Finally, the ViT uses mainly attention mechanisms, representing a different class of models that may interact uniquely with unlearning methods, making it a good architecture for analysis. As a result, the architectural differences effectively capture the diversity of available models and are, therefore, well suited for these experiments. As the other architectures you have asked us to explore are variants of these core model classes it is unlikely that there would be a different interaction of the unlearning methods in these scenarios.

---

> ### Author Response · Authors · 2024-11-27
> **Part 2:**
>
> 5. There should be more motivation in the introduction why this problem is important. For example, I understand why item removal is important (a person wants to remove all of their audio in a dataset) but why is class removal important for request to be forgotten?
>
> Item and class removal are unlearning tasks that are well represented in the literature and have been studied in isolation (Item Removal:[2],[3],[4])  (Class Removal: ,[5],[6],[7],[8],[9])  as well as together [10],[11]. In particular, we would like to draw the reviewer's attention to [10] which was a spotlight paper at NeurIPS.  As ours is one of the first audio unlearning studies, we follow  existing research in other domains and conduct both Item and Class removal to not discriminate between unlearning tasks.
>
> A simple example of where Class removal may be necessary is where one finds that a single class contains biased data. In such cases, you may want to remove the existing representation for the class and fine-tune on the clean data without retraining the model from scratch. Many such examples of the importance of class removal can be considered. Additionally, class removal represents a different type of unlearning task to item removal—the results show that some methods are effective for item removal and others are not.
>
> We argue that exploring both Item and Class removal is rather a strength of the paper and not a weakness, and contributes to a comprehensive understanding of unlearning in audio.
>
> 6. There is no discussion if the proposed solution would comply with the GDPR requirements. What level of machine unlearning performance is required to comply with GDPR?
>
> Machine unlearning is a nascent field created in response to Article 17 of the GDPR Act, and there is no reference to unlearning performance in GDPR. The field itself is trying to offer more sustainable approaches to fulfilling data removal requests over retraining a model, as this is both expensive and incurs a significant environmental cost. We feel it is outside this paper's scope to dictate what unlearning performance is necessary to adhere to GDPR.
>
> We have refactored the introduction to make this point more salient, and discuss GDPR compliance. and we hope you now agree that we have successfully represented the machine unlearning domain in the context of GDPR.
>
>
>
> 7. The paper does not explore the applicability of the Prune and Regrow Paradigm to other data modalities like images, text, or video, which limits the impact of the findings.
>
> We note that our paper focuses on highlighting the modality gap of audio within machine unlearning research and hence experiments on a range of different modalities are beyond the scope of this work. We do not believe that our focus on one modality diminishes the novelty of our contribution. We created the Prune and Regrow Paradigm, a dynamic sparsity unlearning method that outperforms existing methods created for other domains by being dynamic to the models trained rather than offering a one-size-fits-all approach. We have furthermore included more experiments on additional audio datasets that further demonstrate the generalizability of our method.
>
> However, to investigate generalisation to other modalities, we run experiments (Item and Class Removal experiments with scaling) on CIFAR10 (see Appendix F) and show that our method achieves  the best results on Item Removal. Future work could explore other modalities in greater detail to further validate the benefits of our prune and regrow method.

---

> > ### Author Response · Authors · 2024-11-27
> > **References:**
> >
> > [1] Cheng, J. and Amiri, H., 2024. Mu-bench: A multitask multimodal benchmark for machine unlearning. arXiv preprint arXiv:2406.14796.
> >
> > [2] Cheng, J., Dasoulas, G., He, H., Agarwal, C. and Zitnik, M., 2023. Gnndelete: A general strategy for unlearning in graph neural networks. arXiv preprint arXiv:2302.13406.
> >
> > [3] He, Y., Meng, G., Chen, K., He, J. and Hu, X., 2021. Deepobliviate: a powerful charm for erasing data residual memory in deep neural networks. arXiv preprint arXiv:2105.06209.
> >
> > [4] Bourtoule, L., Chandrasekaran, V., Choquette-Choo, C.A., Jia, H., Travers, A., Zhang, B., Lie, D. and Papernot, N., 2021, May. Machine unlearning. In 2021 IEEE Symposium on Security and Privacy (SP) (pp. 141-159). IEEE.
> >
> > [5] Golatkar, A., Achille, A. and Soatto, S., 2020. Eternal sunshine of the spotless net: Selective forgetting in deep networks. In Proceedings of the IEEE/CVF Conference on Computer Vision and Pattern Recognition (pp. 9304-9312).
> >
> > [6] Graves, L., Nagisetty, V. and Ganesh, V., 2021, May. Amnesiac machine learning. In Proceedings of the AAAI Conference on Artificial Intelligence (Vol. 35, No. 13, pp. 11516-11524).
> >
> > [7] Chundawat, V.S., Tarun, A.K., Mandal, M. and Kankanhalli, M., 2023, June. Can bad teaching induce forgetting? unlearning in deep networks using an incompetent teacher. In Proceedings of the AAAI Conference on Artificial Intelligence (Vol. 37, No. 6, pp. 7210-7217).
> >
> > [8] Wang, J., Guo, S., Xie, X. and Qi, H., 2022, April. Federated unlearning via class-discriminative pruning. In Proceedings of the ACM Web Conference 2022 (pp. 622-632).
> >
> > [9] Tarun, A.K., Chundawat, V.S., Mandal, M. and Kankanhalli, M., 2023. Fast yet effective machine unlearning. IEEE Transactions on Neural Networks and Learning Systems.
> >
> > [10] Liu, J., Ram, P., Yao, Y., Liu, G., Liu, Y., SHARMA, P. and Liu, S., 2024. Model sparsity can simplify machine unlearning. Advances in Neural Information Processing Systems, 36.
> >
> > [11] Oesterling, A., Ma, J., Calmon, F. and Lakkaraju, H., 2024, April. Fair machine unlearning: Data removal while mitigating disparities. In International Conference on Artificial Intelligence and Statistics (pp. 3736-3744). PMLR.

---

> > > ### Author Response · Authors · 2024-12-02
> > > **Response to PAef:**
> > >
> > > We hope you are well.
> > >
> > > We are messaging to ask if you have any additional questions concerning our responses. We would be more than happy to discuss any issues such that we can address them adequately.
> > >
> > > Your feedback is important to us and we appreciate the time that you have already dedicated to providing feedback on our work.

---

### Author Response · Authors · 2024-11-27
**Overall Reviewer Responses**

We want to thank all of the reviewers for the time they have dedicated to providing their feedback. Below we address some of the reviewers’ shared concerns.


1. Clarity of method description:


In response to the reviewers’ suggestion  to provide a more detailed explanation of our method ,we have refined our method and description, and have re-run our experiments, now averaging our results over 10 runs instead of 5 for all experiments.

We have added Figures 1 and 2 to aid the explanation of the Prune and Regrow Paradigm: Figure 1 represents the process that occurs in the Prune and Regrow Paradigm, from L1 pruning a base model to re-initialising pruned weights and biases ready for regrowth - the strength of the method lies in re-initialising the pruned values instead of just removing the mask.

Figure 2 is an example (for models trained on SpeechCommands) of how the optimal prune value (the point closest to Utopia in the figure) is selected based on the cosine similarity of the vectorised pruned model compared to the base model as pruning is conducted between 0-100%. The optimal value minimum point from [1,1] is then used for CS unlearning as the prune value, and for POP, we show how if we take the maximum distance from polar pruning [0,-1], we can get a post-optimal prune that prunes more of the network. Once we identify these values, we use the Prune and Regrow process shown in Figure 1.

The Figures and the updated Prune and Regrow section description help to explain the method and the benefits for offering dynamic sparsity unlearning, demonstrating its novelty.

2. Use of Pre-trained models: HuBERT, Wav2Vec2.0, Whisper,Pre-trained AST:


Pre-trained models are undeniably valuable for enhancing performance.
However, in the context of machine unlearning, they can make the interpretation of results harder and lead to incorrect conclusions. In a pre-trained setting, determining whether the pre-trained data influences the effectiveness  of an unlearning method is difficult, as the feature overlap between the pre-training data and the data to be unlearned is hard to quantify.

To be able to perform a comparison of an unlearning method in a fair way the vast majority of previous work ([1],[2],[3],[4],[5],[6],[7],[8],[9],[10],[11]) compares the efficacy of an unlearning method to that of a retrained model (Naive Method) trained solely on the remain set. This allows a comparison of the expected generalisation capacity of the Naive model on the forget set, as a model trained on the remain set  will have predictive capacity over some examples in the Item Removal case (as shown by our work and many others). Additionally, the generalisation gap is also a valuable insight for understanding the effectiveness of an unlearning method and highlights the differences between various methods. Some methods are effective when the generalisation gap is small, as seen in AudioMNIST, but not effective when the generalisation gap is more pronounced (SpeechCommands, UrbanSounds8K and CIFAR10).

As we are conducting a foundational analysis of machine unlearning in audio, we have aligned with the existing literature above by utilizing the Naive baseline on models trained only on the data of interest. Additionally, we have reduced the complexity of the training setup to remove explicit regularisation (weight decay, dropout and augmentation) to reduce the number of interacting factors when evaluating the unlearning methods. As a result, our models' accuracy does not represent the state of the art, but our vanilla setup creates a stable environment wherein machine unlearning can be properly evaluated for the audio domain.

We have not included any of the requested pre-trained models in our updated version as we want to remain true to the sound experimental setup laid out by previous literature such that a fair evaluation of unlearning in audio is conducted. We agree that a separate study in this direction could be helpful - but it would require a new formalisation for machine unlearning on pre-trained models.

---

> ### Author Response · Authors · 2024-11-27
> **Part 2:**
>
> 3. Complexity of datasets + New audio dataset:
>
> As this represents one of the first comprehensive studies on unlearning in audio, we establish an essential foundation for future research to expand upon by exploring larger models and datasets.
>
> The most popular datasets to explore machine unlearning in the literature are small datasets such as MNIST, SHVN and CIFAR10 to evaluate unlearning according to these machine unlearning surveys [12],[13],[14]). As a result, our study uses AudioMNIST due to its conceptual resemblance to MNIST used in previous work, and SpeechCommnads V2, a widely used audio dataset.
>
> In response to the reviewers' request for evaluation on additional datasets, we have now also included results on UrbanSounds8K (event sound classification dataset), including Item and Class removal, scaling of unlearning request, and loss analysis. We have added this to the paper's main body and detail training modifications where necessary. We find that the Prune and Regrow Paradigm provides the best overall performance for UA disparity for Item Removal - in line with the results of SpeechCommands - further demonstrating the generalisability of our approach.
>
> 4. Transfer of the Prune and Regrow Paradigm to other modalities:
>
> To answer some of the reviewers' comments regarding the transferability of the Prune and Regrow Paradigm to other modalities, we have conducted a complete machine unlearning analysis on CIFAR10. These results are provided in Appendix F; with these results, we show that the Prune and Regrow Paradigm is the best method for Item Removal and very competitive for class removal, as well as showing strong scaling capacity for Vision.
>
> While these results are not the paper's primary focus (as we aim to address the modality gap of unlearning in audio), they provide evidence to the benefits of using the Prune and Regrow methods of CS and POP as dynamic sparsity unlearning methods on other modalities.
>
> 5. Improved presentation:
>
> We have improved the presentation of results by reducing the number of figures in the main body. The radar plots are now held in the appendix and referenced where needed, with increased size and text for improved readability. For the loss distributions, we present the most explicit example of the explored architecture and put the rest in the appendix for the reader to consider. On the unlearning scaling, we present the same architecture for Item and Class removal on both SpeechCommands and UrbanSounds8K and put the rest of the architectures in the appendix.
>
> Once again, we thank the reviewers for their time, and hope that the above help to address their concerns.  Below, we also respond to the reviewers’ individual comments.

---

> ### Author Response · Authors · 2024-11-27
> **References:**
>
> [1] Cheng, J., Dasoulas, G., He, H., Agarwal, C. and Zitnik, M., 2023. Gnndelete: A general strategy for unlearning in graph neural networks. arXiv preprint arXiv:2302.13406.
>
> [2] He, Y., Meng, G., Chen, K., He, J. and Hu, X., 2021. Deepobliviate: a powerful charm for erasing data residual memory in deep neural networks. arXiv preprint arXiv:2105.06209.
>
> [3] Golatkar, A., Achille, A. and Soatto, S., 2020. Eternal sunshine of the spotless net: Selective forgetting in deep networks. In Proceedings of the IEEE/CVF Conference on Computer Vision and Pattern Recognition (pp. 9304-9312).
>
> [4] Graves, L., Nagisetty, V. and Ganesh, V., 2021, May. Amnesiac machine learning. In Proceedings of the AAAI Conference on Artificial Intelligence (Vol. 35, No. 13, pp. 11516-11524).
>
> [5] Chundawat, V.S., Tarun, A.K., Mandal, M. and Kankanhalli, M., 2023, June. Can bad teaching induce forgetting? unlearning in deep networks using an incompetent teacher. In Proceedings of the AAAI Conference on Artificial Intelligence (Vol. 37, No. 6, pp. 7210-7217).
>
> [6] Tarun, A.K., Chundawat, V.S., Mandal, M. and Kankanhalli, M., 2023. Fast yet effective machine unlearning. IEEE Transactions on Neural Networks and Learning Systems.
>
> [7] Liu, J., Ram, P., Yao, Y., Liu, G., Liu, Y., SHARMA, P. and Liu, S., 2024. Model sparsity can simplify machine unlearning. Advances in Neural Information Processing Systems, 36.
>
> [8] Liu, J., Xue, M., Lou, J., Zhang, X., Xiong, L. and Qin, Z., 2023. Muter: Machine unlearning on adversarially trained models. In Proceedings of the IEEE/CVF International Conference on Computer Vision (pp. 4892-4902).
>
> [9] Yan, H., Li, X., Guo, Z., Li, H., Li, F. and Lin, X., 2022, July. ARCANE: An Efficient Architecture for Exact Machine Unlearning. In IJCAI (Vol. 6, p. 19).
>
> [10] Oesterling, A., Ma, J., Calmon, F. and Lakkaraju, H., 2024, April. Fair machine unlearning: Data removal while mitigating disparities. In International Conference on Artificial Intelligence and Statistics (pp. 3736-3744). PMLR.
>
> [11] Thudi, A., Deza, G., Chandrasekaran, V. and Papernot, N., 2022, June. Unrolling sgd: Understanding factors influencing machine unlearning. In 2022 IEEE 7th European Symposium on Security and Privacy (EuroS&P) (pp. 303-319). IEEE.
>
> [12] Shaik, T., Tao, X., Xie, H., Li, L., Zhu, X. and Li, Q., 2024. Exploring the landscape of machine unlearning: A comprehensive survey and taxonomy. IEEE Transactions on Neural Networks and Learning Systems.
>
> [13] Nguyen, T.T., Huynh, T.T., Nguyen, P.L., Liew, A.W.C., Yin, H. and Nguyen, Q.V.H., 2022. A survey of machine unlearning. arXiv preprint arXiv:2209.02299.
>
> [14] Xu, J., Wu, Z., Wang, C. and Jia, X., 2024. Machine unlearning: Solutions and challenges. IEEE Transactions on Emerging Topics in Computational Intelligence.

---

### Meta-Review · Area_Chair_zVQA · 2024-12-18

**Metareview:**

This paper investigates the effectiveness of 5 existing machine unlearning methods in the audio domain, focusing on the datasets AudioMNIST, UrbanSound8K and SpeechCommands v2, using 3 different main classes of model architectures (VGGish, CCT, ViT), but without pre-training their models. This makes analysis easier, but almost all real-world use cases rely on pre-trained models. Authors show that current unlearning techniques work for class removal but not item removal. To address this, they introduce Cosine Similarity and Post Optimal Prune within a "Prune and Regrow" paradigm and show improved performance (83% unlearning accuracy gap) for Post Optimal Prune. Authors also show that the proposed model generalizes to other domains, since it achieves good results on CIFAR10, a standard dataset for machine unlearning.

During the rebuttal, one reviewer raised their score in light of the authors' explanation and additional experiments provided (specifically on CIFAR 10). The AC agrees that the rebuttal has made the paper stronger, and it could now be accepted, but overall opinion is split. Authors' argument for not using pre-trained models because the interpretation is more difficult means that the results are less relevant to practical models, and the paper is now also not the very first such study on audio. After reviewing the paper itself, the AC recommends that authors further update the paper (specifically, the analysis and conclusion) since with the inclusion of CIFAR10 experiments, it is now no more an audio-only paper and more discussion on CIFAR results would be helpful. The information presented in the rebuttal has not (as of now) been fully incorporated into the submitted version. This could however probably be achieved within short order.

**Additional Comments On Reviewer Discussion:**

See my comment above - authors presented significant new experimental results, which swayed one reviewer. However, the presentation of the results may have overwhelmed some reviewers, so that no further in-depth discussion did occur. Some open questions (on CIFAR10, how to treat pre-trained models) remain.

---

### Decision · Program_Chairs · 2025-01-22

Reject